# OVERLAP-WEIGHTED ORTHOGONAL META-LEARNER FOR TREATMENT EFFECT ESTIMATION OVER TIME

**Konstantin Hess[1,2,*], Dennis Frauen[1,2], Mihaela van der Schaar[3,4], Stefan Feuerriegel[1,2]**

[1]LMU Munich    [2]Munich Center for Machine Learning    [3]University of Cambridge
[4]Alan Turing Institute    [*]Corresponding author: `k.hess@lmu.de`

## ABSTRACT

Estimating heterogeneous treatment effects (HTEs) in time-varying settings is particularly challenging, as the probability of observing certain treatment sequences decreases exponentially with longer prediction horizons. Thus, the observed data contain little support for many plausible treatment sequences, which creates severe *overlap* problems. Existing meta-learners for the time-varying setting typically assume adequate treatment overlap, and thus suffer from exploding estimation variance when the overlap is low. To address this problem, we introduce a novel *overlap-weighted* orthogonal (WO) meta-learner for estimating HTEs that targets regions in the observed data with high probability of receiving the interventional treatment sequences. This offers a fully data-driven approach through which our WO-learner can counteract instabilities as in existing meta-learners and thus obtain more reliable HTE estimates. Methodologically, we develop a novel Neyman-orthogonal population risk function that minimizes the overlap-weighted oracle risk. We show that our WO-learner has the favorable property of Neyman-orthogonality, meaning that it is robust against misspecification in the nuisance functions. Further, our WO-learner is fully model-agnostic and can be applied to any machine learning model. Through extensive experiments with both transformer and LSTM backbones, we demonstrate the benefits of our novel WO-learner.

## 1 INTRODUCTION

Estimating heterogeneous treatment effects (HTEs) such as conditional average potential outcomes (CAPOs) and conditional average potential outcomes (CAPOs) (Frauen et al., 2025a) *over time* from patient trajectories is central for advancing personalized medicine (Allam et al., 2021; Battalio et al., 2021; Bica et al., 2021; Feuerriegel et al., 2024). Such estimates can, for example, guide treatment adaptation in chronic disease management or inform personalized intervention strategies in digital health.

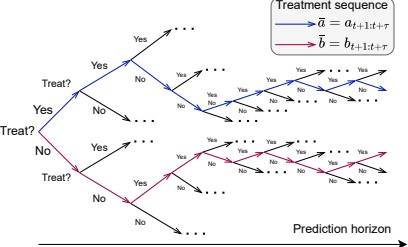

Figure 1: **Overlap problems in time-varying settings.** With longer prediction horizons, treatment sequences become more complex, and the probability of observing each treatment sequence decreases exponentially. Hence, standard meta-learners often have a poor performance due to low overlap.

Unlike standard predictive tasks, HTE estimation is inherently causal and, in time-series settings, requires adjustments for time-varying confounders (Bica et al., 2020). Without such adjustments, time-varying confounding can induce infinite-sample bias and lead to incorrect estimates. To address this, model-agnostic ***meta-learners*** (Frauen et al., 2025a) have been proposed to provide principled strategies for causal adjustments in time-varying settings.[1] Prominent examples include adaptations of the inverse propensity weighting (IPW) learner and the doubly robust (DR) learner to time-varying settings (Frauen et al., 2025a).

---

[1]The term *meta-learner* refers to general estimation strategies ("recipes") to learn causal quantities (Künzel et al., 2019; Frauen et al., 2025b), which can be instantiated with different machine learning backbones (e.g., neural networks).

However, existing meta-learners (Frauen et al., 2025a) for estimating HTEs over time suffer from *instabilities in settings with low overlap*, which renders them inapplicable to medical scenarios. Here, *overlap* refers to the probability of receiving each of the two different treatment sequences of interest; if overlap is low, standard meta-learners have severe estimation variance (Jesson et al., 2020; Melnychuk et al., 2026). This issue is especially serious in time-varying settings: with increasing prediction horizons, the probability of a treatment sequence consists of products of propensity scores (Hess & Feuerriegel, 2025), which makes the treatment overlap *decrease exponentially* (see Fig. 1). As a result, in low-overlap regimes, adjustment strategies based on inverse propensity weighting (IPW) will lead to extreme weights due to the product structure and, hence, division by values that are close to zero.

In this work, we propose a novel *overlap-**w**eighted **o**rthogonal meta-learner* (**WO**-learner) for estimating HTEs over time:

1. Our novel **WO**-learner addresses low-overlap regimes with a carefully designed population risk function that *minimizes a novel, overlap-weighted oracle risk*. Hence, by focusing on high-overlap regions in the data, our **WO**-learner avoids extreme inverse propensity weights and unreliable response function estimates. As a result, our meta-learner provides stable HTE estimates, *even* in low-overlap regimes.

2. We further ensure that our weighted population risk function is *Neyman-orthogonal* with respect to all nuisance functions. Hence, our **WO**-learner is robust to misspecification in the nuisance parameters, meaning that estimation errors in the nuisance functions do **not** propagate as first-order biases into the final HTE estimate, which is a crucial advantage over simple plug-in estimators (Hines et al., 2022; Kennedy, 2022).

3. Our **WO**-learner is *fully model-agnostic* and can be used in combination with *any* machine learning backbone, such as transformers or LSTMs. Further, we derive our **WO**-learner for estimating *conditional average treatment effects* (CATEs). On top of that, we also extend our theory to *conditional average potential outcome estimation* (CAPOs); see Section 4.

We make three key **contributions**: [2] **(1)** We introduce a novel overlap-weighted meta-learner for HTE estimation over time that minimizes the *overlap-weighted oracle risk*. **(2)** We further derive a *Neyman-orthogonal population risk* that eliminates first-order bias from the nuisance functions in the HTE estimates. **(3)** Through extensive experiments, we demonstrate that our **WO**-learner outperforms existing meta-learners, especially in settings with low overlap. In addition, we demonstrate the benefits of our meta-learner in settings where Neyman-orthogonality is crucial, such as limited sample size and complex nuisance functions.

## 2   RELATED WORK

We now review prior literature on treatment effect estimation ***over time***, namely: ① average treatment effect (ATE) estimation, ② model-based HTE estimation, and ③ meta-learners for HTE estimation. *Our WO-learner belongs to the latter category, and this is where our primary contributions are.*

① **ATE vs. HTE estimation *over time*:** The literature on estimating ATEs over time dates back to works in epidemiology and classical statistics (Robins, 1986; 1999; Robins et al., 2000). Examples are G-computation (Bang & Robins, 2005; Robins, 1999; Robins & Hernán, 2009), marginal structural models (Robins & Hernán, 2009; Robins et al., 2000) and structural nested models (Robins, 1994; Robins & Hernán, 2009), which belong to the broader class of so-called G-methods. More recently, targeted maximum likelihood has been adapted for the time-varying setting (van der Laan & Gruber, 2012; van der Laan & Rose, 2018). There is also some literature on model-based methods for estimating ATEs over time (Frauen et al., 2023; Shirakawa et al., 2024). Importantly, all of these works focus on *average* potential outcomes estimation and, therefore, *ignore patient heterogeneity*, because of which these works are not suitable for personalized medicine.

② **Limitations of model-based HTE estimation *over time*:** There has been much research on *model-based* estimation of HTEs over time (Bica et al., 2020; Hess et al., 2024; Hess & Feuerriegel, 2025; Hess et al., 2026; Li et al., 2021; Lim et al., 2018; Ma et al., 2025; Melnychuk et al., 2022; Seedat et al., 2022; Wang et al., 2025). Importantly, the focus in this literature stream is primarily

---

[2]Code is available at `https://github.com/konstantinhess/wo_learner_timeseries`.

on how to adapt the underlying neural backbone, but **not** how to find the best adjustment strategy (i.e., the learning strategy to address time-varying confounding). Further, the above model-based methods are known to be *instantiations* of different meta-learners (see Frauen et al. (2025a) for a discussion). Importantly, **none** of these model-based methods relies on either overlap-weighted or Neyman-orthogonal meta-learners. In contrast, we design an *overlap-weighted* meta-learner that is *Neyman-orthogonal* with respect to all its nuisance functions and that can be applied to ***any*** neural backbone.

③ **Meta-learners for HTE estimation *over time*:** Research on meta-learners for HTE estimators over time is still very limited, and we are aware of only a few works. Lewis & Syrgkanis (2021) developed a method that, however, relies on *parametric assumptions* on the data-generating process and is, therefore, *not* fully model agnostic.

Recently, Frauen et al. (2025a) formalized a suite of meta-learners for the time-varying setting, namely: (a) history adjustment (**HA**), (b) regression adjustment (**RA**), (c) inverse propensity weighting (**IPW**), (d) doubly-robust (**DR**), and an (e) inverse-variance-weighted (**IVW**) learner. However, they all have important *shortcomings* (Table 1): First, the **HA** is biased and does not target the correct estimand. Second, the **RA**, **IPW**, and **IVW** learners have plug-in bias and are not Neyman-orthogonal with respect to their estimated nuisance functions. Frauen et al. (2025a) refer to the IVW-learner as IVW-DR learner. However, it

| | Proper time-varying adj. ($\tau > 0$) | Neyman-orthogonal | Designed for low-overlap regimes |
|---|:---:|:---:|:---:|
| (a) **HA** | ✗ | ✗ | ✗ |
| (b) **RA** | ✓ | ✗ | ✗ |
| (c) **IPW** | ✓ | ✗ | ✗ |
| (d) **DR** | ✓ | ✓ | ✗ |
| (e) **IVW** | ✓ | ✗ | ✗ |
| (∗) **WO** (*ours*) | ✓ | ✓ | ✓ |

Table 1: **Meta-learners for HTE estimation *over time*.** Our **WO**-learner is the only method that adjusts for time-varying confounding, is Neyman-orthogonal with respect to all its nuisance functions, and avoids extreme weights in low-overlap regimes.

is **not** orthogonal with respect to its weights, and errors in the propensities propagate as first order biases to estimated inverse-variance weights, and hence, the reweighted population risk. Hence, for clarity, we refer to it as IVW-learner, as doubly-robustness does **not** hold. Third, the **IPW** and **DR** learners rely on inverse propensity weighting, which can lead to extreme weights for long treatment sequences, especially in settings with low overlap. In contrast, we develop a novel **WO**-learner that is Neyman-orthogonal, and designed to deal with low overlap.

*Why low overlap is a non-trivial challenge for existing meta-learners:* When the interventional treatment sequences have low overlap, inverse propensity scores may lead to extreme weights. Further, small errors in estimated propensity scores lead to large errors in the constructed pseudo-outcomes and, therefore, to extreme variance for the **IPW** and the **DR** learner. Importantly, this issue is even more pronounced in the time-varying setting, where inverse propensity weighting relies on *products* of propensity scores and, thereby, the treatment propensity *decreases exponentially* with longer prediction horizons (Frauen et al., 2025a; Hess et al., 2026; Lim et al., 2018). Here, propensity-score clipping is sometimes used as a heuristic that introduces *uncontrollable bias*, since the truncation level **cannot** be calibrated without access to counterfactual outcomes. In contrast, our **WO**-learner provides principled stabilization under limited overlap. Similar issues also arise for the response functions learned in the **RA** and in the **DR** learner, where low overlap leads to poorly learned, biased response surfaces, especially in high-dimensional covariate spaces. Finally, as the **IVW** learner is not Neyman-orthogonal w.r.t. its weight functions, estimation errors propagate as first-order bias through all time steps, which makes it unstable in low-overlap regimes.

**Research gap:** To the best of our knowledge, there is no meta-learner designed to counteract low-overlap regimes while being Neyman-orthogonal. As a remedy, we propose a novel *overlap-weighted*, *orthogonal* meta-learner (**WO**-learner) for HTE estimation over time.

## 3 PROBLEM FORMULATION

**Setup:** Let $t \in \mathbb{N}_0$ be the time index. Further, let $Y_t \in \mathbb{R}^{d_y}$ the outcome variable of interest (e.g., a variable indicating the health status of a patient), $X_t \in \mathbb{R}^{d_x}$ the covariates that contain relevant patient information (including static features), and $A_t \in \{0, 1\}$ the treatment variable. For any stochastic process $V_t \in \{Y_t, X_t, A_t\}$, we write $\bar{V}_t = V_{0:t} = (V_0, \ldots, V_t)$ for the history of $V_t$ up to time $t$. Then, let $\bar{H}_t = (\bar{Y}_{t-1}, \bar{X}_t, \bar{A}_{t-1})$ be the collective history observed at time step $t$, and $\bar{Z}_t = (\bar{Y}_t, \bar{X}_t, \bar{A}_t)$ the history including the final treatment and outcome. Finally, $\tau$ is the prediction

horizon such that treatment sequences are of the form $a_{t:t+\tau} \in \{0,1\}^{\tau+1}$ (see Figure 1). We further build upon the potential outcomes framework (Neyman, 1923; Rubin, 1978) for the time-varying setting (Robins, 1999; Robins & Hernán, 2009). Formally, let $Y_{t+\tau}[a_{t:t+\tau}]$ be the potential outcome that would have been observed under the *interventional* treatment sequence.

**Estimation task:** Given a history $\bar{H}_t = \bar{h}_t$ and two interventional sequences of treatments $a_{t:t+\tau} = (a_t, \ldots, a_{t+\tau})$ and $b_{t:t+\tau} = (b_t, \ldots, b_{t+\tau})$, our main objective is to estimate the CATE

$$\mu_t^{\bar{a},\bar{b}}(\bar{h}_t) = \mathbb{E}\Big[Y_{t+\tau}[a_{t:t+\tau}] - Y_{t+\tau}[b_{t:t+\tau}] \mid \bar{H}_t = \bar{h}_t\Big]. \tag{1}$$

We extend our theory to the conditional average potential outcome (CAPO), which is defined as

$$\mu_t^{\bar{a}}(\bar{h}_t) = \mathbb{E}\Big[Y_{t+\tau}[a_{t:t+\tau}] \mid \bar{H}_t = \bar{h}_t\Big]. \tag{2}$$

**Identifiability:** In order to ensure identifiability from observational data, we need to make the following assumptions that are standard in the literature (Bica et al., 2020; Frauen et al., 2025a; Hess et al., 2024; Li et al., 2021; Melnychuk et al., 2022): (i) *Consistency:* Whenever the observed treatment $A_t$ equals the interventional treatment $a_t$, the observed outcome $Y_t$ corresponds to the potential outcome $Y_t[a_t]$. (ii) *Positivity:* Given a history $\bar{H}_t = \bar{h}_t$ with $\mathbb{P}(\bar{H}_t = \bar{h}_t) > 0$, there is non-zero probability $\mathbb{P}(A_t = a_t \mid \bar{H}_t = \bar{h}_t) > 0$ of receiving any treatment $A_t = a_t$. (iii) *Sequential ignorability:* Given a history $\bar{H}_t = \bar{h}_t$, the treatment assignment $A_t$ is independent of the potential outcome $Y_{t+\tau}[a_{t:t+\tau}]$.

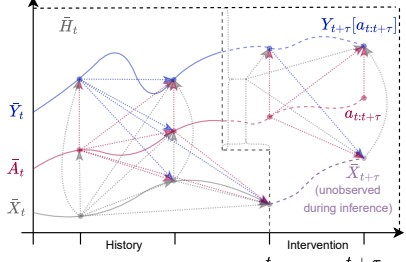

Figure 2: **Causal graph.** Shown are the observed history, the interventional treatment sequence, and the potential outcome, along with their causal connections. When intervening on future treatments, we do not observe the future covariates (in purple), which leads to *time-varying confounding*.

**Time-varying confounding:** The key difficulty in estimating HTEs over time lies in *time-varying confounding* whenever $\tau > 0$, i.e., when we are interested in a *sequence* of future treatments $a_{t:t+\tau}$ (see Fig. 2). Then, at time $t$, we intervene both on the current treatment $A_t$ **and** treatments $A_{t+\delta}$ for $0 < \delta \leq \tau$ that lie several time steps in the future. However, future covariates $X_{t+\delta}$ and outcomes $Y_{t+\delta-1}$, that are *unobserved during inference time*, will confound the treatment assignment of $A_{t+\delta}$. This induces a feedback loop that needs to be accounted for (see Fig. 2).[3] Importantly, simply conditioning on the observed history $\bar{H}_t$ (i.e., a backdoor-type history-adjustment) is *biased* in this scenario. Specifically, for $\tau > 0$,

$$\underbrace{\mathbb{E}\Big[Y_{t+\tau}[a_{t:t+\tau}] \mid \bar{H}_t = \bar{h}_t\Big]}_{\text{proper adjustment (=our method)}} \neq \underbrace{\mathbb{E}\Big[Y_{t+\tau} \mid \bar{H}_t = \bar{h}_t, A_{t:t+\tau} = a_{t:t+\tau}\Big]}_{\text{naïve history adjustment}}, \tag{3}$$

which means that methods for the right-hand side (i.e., as in the **HA**-learner) target an incorrect estimand that is different from our causal quantity of interest. Instead, proper adjustments for time-varying confounding are required, such as in the **RA**, **IPW**, or **DR** learner. However, these adjustment strategies lead to poor performance when the interventional treatment overlap is low. The inverse variance weighted adjustment in the **IVW** learner tries to circumvent this issue but suffers from first-order plug-in bias from its estimated weights that propagate through all timesteps. As a remedy, we develop our novel *weighted orthogonal (WO) meta-learner*.

**Standard nuisance functions:** To perform proper adjustments for time-varying confounding, all meta-learners rely on so-called nuisance functions; that is, functions that are not of direct interest but must be estimated accurately to enable valid estimation of the target parameter. Both existing meta-learners and, later, also our **WO**-learner rely on estimating response functions and/or propensity scores, which we define below.

**Definition 3.1** (Response functions and propensity scores). *For interventional treatment sequences $\bar{a} = a_{t:t+\tau}$ and $\bar{b} = b_{t:t+\tau}$, let the **response functions for CATE** be defined as*

$$\mu_j^{\bar{a},\bar{b}}(\bar{h}_j) = \mu_j^{\bar{a}}(\bar{h}_j) - \mu_j^{\bar{b}}(\bar{h}_j), \tag{4}$$

---

[3]In the static setting, a similar issue is known as runtime confounding, where not all confounders are observed during inference time (Coston et al., 2020).

where $\mu_j^{\bar{a}}(\bar{h}_j), \mu_j^{\bar{b}}(\bar{h}_j)$ are the **response functions for the CAPOs** with

$$\mu_{t+\tau}^{\bar{a}}\left(\bar{h}_{t+\tau}\right) = \mathbb{E}\left[Y_{t+\tau} \mid \bar{H}_{t+\tau} = \bar{h}_{t+\tau}, A_{t+\tau} = a_{t+\tau}\right] \tag{5}$$

and, recursively, for $j \in \{t, \ldots, t + \tau - 1\}$,

$$\mu_j^{\bar{a}}\left(\bar{h}_j\right) = \mathbb{E}\left[\mu_{j+1}^{\bar{a}}(\bar{H}_{j+1}) \mid \bar{H}_j = \bar{h}_j, A_j = a_j\right]. \tag{6}$$

*Further, let the **propensity scores** for $j \in \{t, \ldots, t + \tau\}$ be*

$$\pi_j^{\bar{a}}(\bar{h}_j) = \mathbb{P}(A_j = a_j \mid \bar{H}_j = \bar{h}_j). \tag{7}$$

Finally, we introduce the pseudo-outcomes of the DR learner, which are a subcomponent of our weighted population risk. Here, pseudo-outcomes are variables that are estimated from nuisance functions, for which the conditional expectation equals the target causal estimand (in our case: CATE / CAPO) and, hence, enable consistent estimation.

**Definition 3.2** (DR pseudo-outcomes (Frauen et al., 2025a)). *For interventional treatment sequences $\bar{a} = a_{t:t+\tau}$ and $\bar{b} = b_{t:t+\tau}$, let the **DR pseudo-outcomes for CATE** be*

$$\gamma_t^{\bar{a},\bar{b}}(\bar{Z}_{t+\tau}) = \gamma_t^{\bar{a}}(\bar{Z}_{t+\tau}) - \gamma_t^{\bar{b}}(\bar{Z}_{t+\tau}), \tag{8}$$

*where the corresponding **DR pseudo-outcomes for CAPO** are*

$$\gamma_t^{\bar{a}}(\bar{Z}_{t+\tau}) = \prod_{j=t}^{t+\tau} \frac{\mathbb{1}_{\{A_j=a_j\}}}{\pi_j^{\bar{a}}(\bar{H}_j)} Y_{t+\tau} + \sum_{j=t}^{t+\tau} \mu_j^{\bar{a}}\left(\bar{H}_j\right) \left(1 - \frac{\mathbb{1}_{\{A_j=a_j\}}}{\pi_j^{\bar{a}}(\bar{H}_j)}\right) \prod_{k=t}^{j-1} \frac{\mathbb{1}_{\{A_k=a_k\}}}{\pi_k^{\bar{a}}(\bar{H}_k)}. \tag{9}$$

Different from existing meta-learners, the population risk function we minimize in our **WO**-learner minimizes a *weighted, Neyman-orthogonal risk* to address low-overlap regimes.

## 4 WEIGHTED ORTHOGONAL META-LEARNER

The key idea of our **WO**-learner is to up-weight samples in the training data that have a higher probability of receiving the interventional treatment sequences. By up-weighting samples with *larger overlap* (in the case of CATE estimation) / *larger propensity* (in the case of CAPO estimation), we ensure that we target samples that are more relevant for estimating the HTE of interest (Figure 3). This requires non-trivial derivations to guarantee that our **WO**-learner (i) *correctly adjusts for time-varying confounding*, (ii) *minimizes the weighted oracle risk*, and (iii) is *Neyman-orthogonal* with respect to all its nuisance functions.

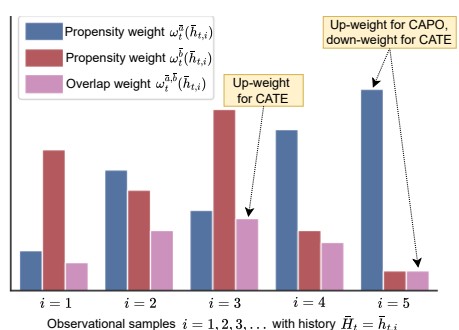

Figure 3: **Weight functions.** Our weight functions are designed to up-weight samples with large treatment overlap (CATE), or treatment propensity (CAPO). Thereby, we effectively overcome estimation variance issues in low-overlap and low-propensity regimes in a fully data-driven way.

Below, we first introduce our novel weight functions in Section 4.1. Then, we present our pseudo-outcomes for both CATE and CAPO estimation, and develop our weighted population risk; by minimizing our population risk, we adjusts for time-varying confounding while minimizing the weighted oracle risk. Finally, we ensure Neyman-orthogonality of our population risk with respect to all nuisance functions in Section 4.2.

### 4.1 WEIGHTED POPULATION RISK

We now introduce our weighted population risk for both CATE and CAPO estimation. We start by defining the weight functions, followed by our weighted population risk function. We then show that it is guaranteed to minimize the oracle risk ($\rightarrow$ Proposition 4.3) and, additionally, that it properly adjusts for time-varying confounding ($\rightarrow$ Corollary 4.4).

---

**Definition 4.1** (Weight functions). *Let the **weight functions for CATE** be the **overlap weights***

$$\omega_j^{\bar{a},\bar{b}}(\bar{h}_\ell) = \omega_j^{\bar{a}}(\bar{h}_\ell)\omega_j^{\bar{b}}(\bar{h}_\ell), \tag{10}$$

*where we define the **weight functions for CAPO** as the **propensity weights***

$$\omega_j^{\bar{a}}(\bar{h}_\ell) = \mathbb{E}\Big[\prod_{k=j}^{t+\tau} \pi_k^{\bar{a}}(\bar{H}_k) \ \Big| \ \bar{H}_\ell = \bar{h}_\ell\Big] = p(A_{j:t+\tau} = a_{j:t+\tau} \mid \bar{H}_\ell = \bar{h}_\ell) \tag{11}$$

*for $j, \ell \in \{t, \ldots, t + \tau\}$. Finally, we summarize the **set of nuisance functions for CATE** as $\eta^{\bar{a},\bar{b}} = \eta^{\bar{a}} \cup \eta^{\bar{b}}$, with the **set of nuisance functions for CAPO** $\eta^{\bar{a}} = \{\pi_j^{\bar{a}}, \mu_j^{\bar{a}}, \omega_j^{\bar{a}}\}_{j=t}^{t+\tau}$.*

---

Intuitively, our weight functions work as follows. For CATE, they act as *overlap weights*, i.e., they up-weight samples with a higher probability of receiving **both** interventional treatment sequences. Likewise, for CAPO, the weight functions correspond to *propensity weights*. That is, they seek to up-weight samples in the data that have a higher probability of receiving a single interventional treatment sequence of interest (see Fig. 3). Thereby, we effectively circumvent key issues of existing meta-learners that lead to highly unstable inverse propensity weights and response function estimates.

Next, we develop our weighted population risk that minimizes the oracle risk; specifically, our weighted risk function minimizes a weighted error term that up-weights samples that have a higher probability of receiving the interventional treatment sequences. Thereby, we counteract the issue in existing meta-learners (**IPW**, **RA**, **DR**) that suffer from poor data support when the overlap is low. For this, we first define the pseudo-outcomes for our **WO**-learner, which we will later need to satisfy Neyman-orthogonality in Section 4.2. Our pseudo-outcomes directly arise by orthogonalizing our weighted oracle risk. They inherit part of the structure of DR pseudo-outcomes, part from the orthogonalization of the overlap weights, and part of their combined structure.

---

**Definition 4.2** (**WO** pseudo-outcomes). *Let*

$$\rho_t^{\bar{a},\bar{b}}(\bar{Z}_{t+\tau}) = \rho_t^{\bar{a}}(\bar{Z}_{t+\tau})\omega_t^{\bar{b}}(\bar{H}_t) + \rho_t^{\bar{b}}(\bar{Z}_{t+\tau})\omega_t^{\bar{a}}(\bar{H}_t) - \omega_t^{\bar{a},\bar{b}}(\bar{H}_t), \tag{12}$$

$$\rho_t^{\bar{a}}(\bar{Z}_{t+\tau}) = \prod_{j=t}^{t+\tau} \pi_j^{\bar{a}}(\bar{H}_j) + \sum_{j=t}^{t+\tau} \Big(\mathbb{1}_{\{a_j=A_j\}} - \pi_j^{\bar{a}}(\bar{H}_j)\Big)\omega_{j+1}^{\bar{a}}(\bar{H}_j) \prod_{t \leq k < j} \pi_k^{\bar{a}}(\bar{H}_k). \tag{13}$$

*Then, we define our **WO pseudo-outcomes for CATE** as*

$$\xi_t^{\bar{a},\bar{b}}(\bar{Z}_{t+\tau}) = \mu_t^{\bar{a},\bar{b}}(\bar{H}_t) + \frac{\omega_t^{\bar{a},\bar{b}}(\bar{H}_t)}{\rho_t^{\bar{a},\bar{b}}(\bar{Z}_{t+\tau})}\Big(\gamma_t^{\bar{a},\bar{b}}(\bar{Z}_{t+\tau}) - \mu_t^{\bar{a},\bar{b}}(\bar{H}_t)\Big) \tag{14}$$

*and, likewise, the **WO pseudo-outcomes for CAPO** as*

$$\xi_t^{\bar{a}}(\bar{Z}_{t+\tau}) = \mu_t^{\bar{a}}(\bar{H}_t) + \frac{\omega_t^{\bar{a}}(\bar{H}_t)}{\rho_t^{\bar{a}}(\bar{Z}_{t+\tau})}\Big(\gamma_t^{\bar{a}}(\bar{Z}_{t+\tau}) - \mu_t^{\bar{a}}(\bar{H}_t)\Big). \tag{15}$$

---

We now state our first theorem, which guarantees that our *weighted population risk minimizes the weighted oracle risk*. That is, our weighted population risk assigns the appropriate weights from Definition 4.1 while correctly adjusting for time-varying confounding.

**Theorem 4.3** (Weighted population risk). *Let $\circ \in \{(\bar{a}, \bar{b}), \bar{a}\}$ for CATE and CAPO, respectively. Then, the population risk function*

$$\mathcal{L}(g; \eta^\circ) = \frac{1}{\mathbb{E}\big[\omega_t^\circ(\bar{H}_t)\big]}\mathbb{E}\Big[\rho_t^\circ(\bar{Z}_{t+\tau})\Big(\xi_t^\circ(\bar{Z}_{t+\tau}) - g(\bar{H}_t)\Big)^2\Big] \tag{16}$$

*minimizes the oracle risk*

$$\mathcal{L}^*(g;\eta^\circ) = \frac{1}{\mathbb{E}\left[\omega_t^\circ(\bar{H}_t)\right]} \mathbb{E}\left[\omega_t^\circ(\bar{H}_t)\left(\mu_t^\circ(\bar{H}_t) - g(\bar{H}_t)\right)^2\right]. \tag{17}$$

*Proof.* The proof for CAPO can be found in Supplement C.1 and for CATE in Supplement C.2. Therein, we derive several helping lemmas, including the result that $\mathbb{E}[\rho_t^\circ(\bar{Z}_{t+\tau})|\bar{H}_t] = \omega_t^\circ(\bar{H}_t)$, which we leverage to prove our main theorem. □

Finally, we show that minimizing our weighted population risk guarantees that we target the correct estimand and, therefore, that our WO-learner *adjusts for time-varying confounding*.

**Corollary 4.4** (Time-varying adjustment). *The minimizer of the weighted population risk $\mathcal{L}(g;\eta^\circ)$ adjusts for time-varying confounding.*

*Proof.* We leverage Theorem 4.3 and notice that, since $\omega_t^\circ(\bar{H}_t) > 0$ by positivity, $\mathcal{L}^*(g;\eta^\circ)$ (and, hence, $\mathcal{L}(g;\eta^\circ)$) is minimized if and only if $g = \mu_t^\circ$, which is exactly the target estimand. □

**Weighted orthogonal learning:** We summarize the training of our **WO**-learner with the *empirical weighted risk* in Algorithm 1. For this, we first learn the response functions $\hat{\mu}_j^{\bar{a}}$ and the propensity scores $\hat{\pi}_j^{\bar{a}}$, $j \in \{t, \dots, t+\tau\}$, and then, using the pull-out property of expectations, the weights via

$$\hat{\omega}_j^{\bar{a}}(\bar{h}_j) = \hat{\mathbb{E}}\left[\prod_{k=j}^{t+\tau} \hat{\pi}_k^{\bar{a}}(\bar{H}_k) \ \Big| \ \bar{H}_j = \bar{h}_j\right] = \hat{\mathbb{E}}\left[\prod_{k=j+1}^{t+\tau} \hat{\pi}_k^{\bar{a}}(\bar{H}_k) \ \Big| \ \bar{H}_j = \bar{h}_j\right]\hat{\pi}_j^{\bar{a}}(\bar{h}_j). \tag{18}$$

### 4.2 NEYMAN-ORTHOGONALITY

In the following, we show that our weighted population risk from Theorem 4.3 is Neyman-orthogonal with respect to all its nuisance functions. This is *different* from **IPW**, **RA**, and **IVW**, all of which suffer from severe plug-in bias, which means that estimation errors in their estimated nuisance functions propagate as first-order biases into their final estimated pseudo-outcomes (Kennedy, 2022). Bias propagation is *even more severe in the time-varying setting*, where multiple nuisance functions are learned on top of each other, and incorrectly estimated nuisance functions in earlier stages lead to even worse bias in nuisance functions at later stages (and, hence, the final target estimate). In contrast, our **WO**-learner is *robust against estimation errors in its nuisance functions*; ⇒ bias from nuisance function estimates only propagates as lower-order errors to the final HTE estimate. Importantly, this includes estimation errors from the weights as in Equation 18.

If we directly minimize the oracle risk $\mathcal{L}^*(g;\eta^\circ)$ as in Theorem 4.3, we do **not** achieve Neyman-orthogonality with respect to the nuisance functions $\eta^\circ$. Instead, we need our tailored population risk $\mathcal{L}(g;\eta^\circ)$.

---

**Algorithm 1 WO** learning

**Input:** Data $\mathcal{D}_n = \{\bar{Z}_{t+\tau,i}\}_{i=1}^n$, nuisance function estimators $\hat{\eta}^\circ$, parametric second-stage estimator $\hat{g}_\theta$, sample split $\lambda \in (0,1)$.

1: Perform sample split $\mathcal{D}^\eta_{\lceil(1-\lambda)n\rceil}, \mathcal{D}^g_{\lfloor\lambda n\rfloor}$

2: Learn nuisance functions $\hat{\eta}^\circ$ on $\mathcal{D}^\eta_{\lceil(1-\lambda)n\rceil}$ and evaluate on $\mathcal{D}^g_{\lfloor\lambda n\rfloor}$
3: Construct $\hat{\gamma}_t^\circ$ and $\hat{\rho}_t^\circ$ from evaluated nuisance estimators
4: Construct **WO** pseudo outcomes $\hat{\xi}_t^\circ$
5: Minimize empirical weighted risk

$$\hat{\mathcal{L}}(\hat{g}_\theta;\eta^\circ) = \frac{1}{\sum_{i=1}^{\lfloor\lambda n\rfloor} \hat{\omega}_t^\circ(\bar{H}_{t,i})} \sum_{i=1}^{\lfloor\lambda n\rfloor} \Big[\hat{\rho}_t^\circ(\bar{Z}_{t+\tau,i})$$
$$\times \left(\hat{\xi}_t^\circ(\bar{Z}_{t+\tau,i}) - \hat{g}_\theta(\bar{H}_{t,i})\right)^2\Big]$$

w.r.t. $\theta$ on $\mathcal{D}^g_{\lfloor\lambda n\rfloor}$ (e.g., with gradient descent).
6: **Return** optimized **WO**-learner $\hat{g}_\theta$

---

**Theorem 4.5** (Neyman-orthogonality). *Let $\circ \in \{(\bar{a},\bar{b}),\bar{a}\}$ for CATE and CAPO, respectively. The weighted population risk*

$$\mathcal{L}(g;\eta^\circ) = \frac{1}{\mathbb{E}\left[\omega_t^\circ(\bar{H}_t)\right]} \mathbb{E}\left[\rho_t^\circ(\bar{Z}_{t+\tau})\left(\xi_t^\circ(\bar{Z}_{t+\tau}) - g(\bar{H}_t)\right)^2\right] \tag{19}$$

*is Neyman-orthogonal with respect to all nuisance functions $\eta^\circ$.*

*Proof.* The proof for CAPO is in Supplement C.1 and for CATE in Supplement C.2. Therein, we first calculate $D_g \mathcal{L}(g; \eta^\circ)[\hat{g} - g]$, i.e., the path-wise derivative of $\mathcal{L}(g; \eta^\circ)$ w.r.t. the target parameter $g$. Then, to establish Neyman orthogonality, we check that the cross-derivative with respect to any nuisance function vanishes, i.e., that the second-order derivative $D_{h_j} D_g \mathcal{L}(g; \eta^\circ)[\hat{g} - g, \hat{h}_j^\circ - h_j^\circ] = 0$ for all $h_j \in \eta^\circ$. Intuitively, this ensures that the influence of nuisance function estimation errors enters only at second order, which makes the score function locally robust to small perturbations (i.e., estimation errors) of $\eta^\circ$. $\qquad\square$

• **Remark 1:** For a single-step-ahead prediction $\tau = 0$ (i.e., when there is **no** time-varying confounding as in the static setting), the R-learner (Nie & Wager, 2021) has the same overlap weights as our **WO**-learner for CATE. In other words, our **WO**-learner for CATE is a *non-trivial* generalization of the R-learner to the time-series setting. We show this property formally in Supplement C.3.

• **Remark 2:** Further, we show in Supplement C.4 that our overlap weights lead to *uniformly bounded variance of our pseudo-outcomes*, which enables *stable estimation even in low-overlap regimes*.

**Implementation:** All meta-learners, including our **WO**-learner, can be implemented with any state-of-the-art neural network. Analogous to Frauen et al. (2025a), our main results in Section 5 are with transformers (Vaswani et al., 2017) as neural backbones for both the nuisance function estimators and the second-stage estimators. To ensure a fair comparison, all nuisance estimators and second-stage regressions share the same transformer-based architecture (**implementation details** are in Supplement E). In our experiments, we evaluate our **WO**-learner against the entire family of meta-learners, and hence the comparison is *exhaustive*. As model-based methods are instantiations of specific meta-learners, including (or any specific model-architecture) them would distort the comparison which meta-learner is the best-performing strategy, and make the evaluation fundamentally unfair. Following Frauen et al. (2025a), we highlight that these implementations serve as an example; the optimal model architecture depends on many different factors such as sample size or data dimensionality (Curth & van der Schaar, 2021).

## 5    NUMERICAL EXPERIMENTS

In this section, we empirically evaluate our **WO**-learner against existing meta-learners. The purpose of our experiments (i) to *verify the theoretical insights* developed in Section 4; and (ii) to show that *our WO-learner consistently improves upon standard meta-learners* across a diverse set of settings with low overlap or where Neyman-orthogonality is crucial such as limited sample size and complex nuisances. Our main results are with transformer instantiations (see Supplement E for *implementation details*, *hyperparameters*, and *runtime*). Further, we provide *additional ablations with LSTM instantiations*. All experiments are repeated over five different seeds.

**Datasets:** We provide experimental results across several datasets, including synthetic, semi-synthetic, and real-world data. • We simulate four **synthetic datasets**, where we isolate different complexities for CATE estimation in the time-varying setting. • We then show that our **WO**-learner can deal with real-world covariates, and provide experiments on **semi-synthetic data** based on the MIMIC-III dataset (Johnson et al., 2016). Different from observational data, the advantages of both synthetic and semi-synthetic data are that we have access to the *ground-truth CATEs* and, thereby, can *correctly validate all meta-learners* (Poinsot et al., 2025). • We provide results on a **real-world observational dataset** in Supplement G.

• **Synthetic data:** In order to *isolate the different complexities in the time-varying setting*, we run experiments on different synthetic datasets $\mathcal{D}^* \in \{\mathcal{D}^\gamma, \mathcal{D}^\pi, \mathcal{D}^\mu, \mathcal{D}^N\}$. Therein, we show that our **WO**-learner **(1)** benefits from its overlap weights in low-overlap regimes ($\mathcal{D}^\gamma$); **(2)** has crucial advantages over propensity based-methods when the propensity score function is complex ($\mathcal{D}^\pi$); **(3)** outperforms regression adjustments when the response function is complex ($\mathcal{D}^\mu$); and **(4)** remains robust even in low-sample settings ($\mathcal{D}^N$). On a high level, **(1)** and **(2)** highlight the importance of our overlap weights, while **(2)**–**(4)** show the benefits of Neyman-orthogonality. All experiments are conducted for multi-step-ahead predictions, and all datasets have time-varying confounding. Details about the data-generating processes are in Supplement D.1.

| Overlap $\gamma$ | 0.5 | 1.0 | 1.5 | 2.0 | 2.5 | 3.0 | 3.5 | 4.0 | 4.5 | 5.0 | 5.5 | 6.0 | 6.5 |
|---|---|---|---|---|---|---|---|---|---|---|---|---|---|
| (a) **HA** | $0.17 \pm 0.03$ | $0.19 \pm 0.05$ | $0.20 \pm 0.07$ | $0.21 \pm 0.07$ | $0.23 \pm 0.05$ | $0.22 \pm 0.07$ | $0.24 \pm 0.06$ | $0.36 \pm 0.06$ | $0.22 \pm 0.04$ | $0.22 \pm 0.04$ | $0.25 \pm 0.03$ | $0.25 \pm 0.03$ | $0.25 \pm 0.03$ |
| (b) **RA** | $0.10 \pm 0.04$ | $0.11 \pm 0.04$ | $0.09 \pm 0.02$ | $0.11 \pm 0.03$ | $0.10 \pm 0.02$ | $0.10 \pm 0.02$ | $0.11 \pm 0.03$ | $0.12 \pm 0.04$ | $0.09 \pm 0.03$ | $0.11 \pm 0.03$ | $0.09 \pm 0.02$ | $0.10 \pm 0.04$ | $0.09 \pm 0.03$ |
| (c) **IPW** | $0.09 \pm 0.02$ | $0.10 \pm 0.04$ | $0.13 \pm 0.04$ | $0.10 \pm 0.04$ | $0.12 \pm 0.05$ | $0.19 \pm 0.08$ | $0.30 \pm 0.20$ | $0.70 \pm 0.76$ | $0.28 \pm 0.15$ | $0.33 \pm 0.12$ | $0.45 \pm 0.29$ | $0.67 \pm 0.32$ | $0.47 \pm 0.54$ |
| (d) **DR** | $0.06 \pm 0.01$ | $0.06 \pm 0.01$ | $0.08 \pm 0.04$ | $0.08 \pm 0.01$ | $0.10 \pm 0.04$ | $0.13 \pm 0.05$ | $0.13 \pm 0.03$ | $0.26 \pm 0.22$ | $0.17 \pm 0.07$ | $0.17 \pm 0.10$ | $0.20 \pm 0.10$ | $0.32 \pm 0.15$ | $0.20 \pm 0.10$ |
| (e) **IVW** | $0.06 \pm 0.01$ | $0.05 \pm 0.02$ | $0.07 \pm 0.04$ | $0.07 \pm 0.03$ | $0.08 \pm 0.05$ | $0.12 \pm 0.06$ | $0.11 \pm 0.03$ | $0.62 \pm 0.72$ | $0.13 \pm 0.05$ | $0.17 \pm 0.07$ | $0.08 \pm 0.02$ | $0.15 \pm 0.05$ | $0.16 \pm 0.05$ |
| ($*$) **WO** (*ours*) | $\mathbf{0.03 \pm 0.01}$ | $\mathbf{0.02 \pm 0.01}$ | $\mathbf{0.04 \pm 0.02}$ | $\mathbf{0.05 \pm 0.02}$ | $\mathbf{0.07 \pm 0.04}$ | $\mathbf{0.07 \pm 0.03}$ | $\mathbf{0.08 \pm 0.03}$ | $\mathbf{0.10 \pm 0.07}$ | $\mathbf{0.07 \pm 0.03}$ | $\mathbf{0.05 \pm 0.02}$ | $\mathbf{0.06 \pm 0.02}$ | $\mathbf{0.08 \pm 0.02}$ | $\mathbf{0.08 \pm 0.02}$ |
| Rel. improv. (%) | 54.4% | 58.4% | 39.7% | 21.0% | 6.5% | 25.9% | 27.6% | 13.6% | 23.1% | 50.2% | 26.8% | 16.2% | 13.5% |

Table 2: **Low-overlap regime** $\mathcal{D}^\gamma$: Reported are the average RMSEs $\pm$ standard deviation for CATE estimation, and the relative improvement of our **WO**-learner over the best performing baselines across different levels of overlap (larger values of $\gamma$ correspond to lower overlap). Due to our weighted population loss, our **WO**-learner is highly stable *even* when the overlap is very low.

**(1) Low-overlap regime** $\mathcal{D}^\gamma$: We control the overlap in the data generating process with an overlap parameter $\gamma$. By increasing $\gamma$, we decrease the overlap in the observed data, i.e., the probability of receiving both interventional treatment sequences.

Results: Table 2 shows that our **WO**-learner *outperforms all existing meta-learners over all levels of overlap*. This confirms the *effectiveness of our overlap weights*. We further find that the **RA**-learner performs fairly stable, as it avoids inverse propensity weights that become increasingly more unstable in low-overlap regimes. Nonetheless, our **WO**-learner consistently outperforms the best baseline with a relative improvement of up to $58.4\%$.

**(2) Complex propensity score** $\mathcal{D}^\pi$: We simulate data with a complex propensity score. We further increase the prediction horizon $\tau$, and study the performance of meta-learners that are based on the propensity score. Thereby, we gain insight into how errors propagate over time, and how the meta-learners behave under exponentially decreasing overlap for increasing prediction horizons.

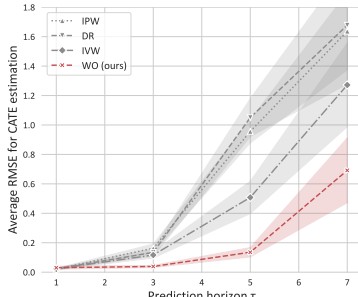

Figure 4: **Complex propensity score** $\mathcal{D}^\pi$: Reported are the average RMSEs for CATE estimation. Our **WO**-learner significantly improves upon existing meta-learners, even for large prediction horizons.

Results: In Figure 4, we can clearly see the benefits of our **WO**-learner over propensity-based baselines due to its *overlap weights*, and its *Neyman-orthogonal population risk function*. Our **WO**-learner is robust against estimation errors in the estimated *propensity scores and weight functions*, and errors do not propagate as for the **IPW** and the **IVW** learner. Further, the performance deteriorates for the **IPW** and **DR** learners is due to the exponentially decreasing overlap for increasing prediction horizons $\tau$. In contrast, our **WO**-learner remains highly stable.

**(3) Complex response function** $\mathcal{D}^\mu$: In this dataset, we simulate a complex response function, and further vary the complexity by increasing the dimension of the covariate space. The purpose of this experimental setup is to empirically validate that our **WO**-learner is Neyman-orthogonal with respect to its estimated response functions. Hence, we compare it with the **RA** learner, which solely relies on estimating the response functions and is **not** Neyman-orthogonal.

Results: Figure 5 shows the results for the complex response function. As our **WO**-learner is also *Neyman-orthogonal with respect to its estimated response functions*, we can clearly see that we outperform a simple **RA**-learner. As the response function becomes more complex with increasing dimension of the covariate space, our **WO**-learner stays largely unaffected.

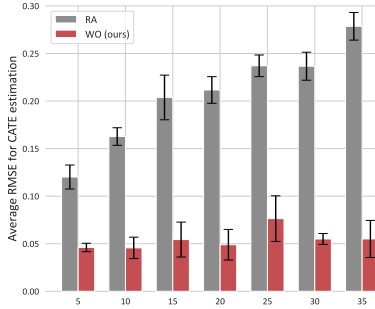

Figure 5: **Complex response function** $\mathcal{D}^\mu$: Reported are the average RMSEs for CATE estimation for increasing dimensionality of the covariate space. Our **WO**-learner clearly outperforms the non-orthogonal **RA** learner.

**(4) Low-sample setting** $\mathcal{D}^N$: We study the performance in settings with low sample size. By decreasing the number of samples for training, we demonstrate the importance of *Neyman-orthogonality with respect to all estimated nuisance functions*. That is, we show empirically that increasing errors in the nuisance function estimates due to decreasing sample size only propagate as lower-order errors to the CATE estimates of our **WO**-learner.

Results: Table 3 shows that, for decreasing sample size, our **WO**-learner maintains a stable performance, even as errors in its estimated nuisance functions increase. This is because our learner is Neyman-orthogonal with respect to all nuisance functions, and, thereby, errors do not propagate as first-order biases through all time steps up to the final CATE estimate.

• **Semi-synthetic data:** We analyze the performance of all meta-learners for increasing prediction horizons on semi-synthetic data. For this, we simulate treatments and outcomes based on real-world patient covariates of the MIMIC-III dataset with intensive care unit patients (Johnson et al., 2016). Therein, we (i) show that our **WO**-learner can easily handle the complexities of observational covariates, while we (ii) ensure that we have access to

| $n_{\text{train}}$ | 8,000 | 7,000 | 6,000 | 5,000 | 4,000 | 3,000 | 2,000 |
|---|---|---|---|---|---|---|---|
| (a) **HA** | $0.33 \pm 0.05$ | $0.38 \pm 0.05$ | $0.39 \pm 0.06$ | $0.42 \pm 0.03$ | $0.50 \pm 0.04$ | $0.49 \pm 0.04$ | $0.58 \pm 0.10$ |
| (b) **RA** | $0.15 \pm 0.02$ | $0.16 \pm 0.03$ | $0.14 \pm 0.02$ | $0.15 \pm 0.04$ | $0.23 \pm 0.05$ | $0.20 \pm 0.02$ | $0.30 \pm 0.10$ |
| (c) **IPW** | $0.13 \pm 0.04$ | $0.14 \pm 0.07$ | $0.16 \pm 0.04$ | $0.17 \pm 0.06$ | $0.24 \pm 0.09$ | $0.58 \pm 0.35$ | $0.41 \pm 0.26$ |
| (d) **DR** | $0.11 \pm 0.04$ | $0.18 \pm 0.08$ | $0.13 \pm 0.03$ | $0.17 \pm 0.03$ | $0.20 \pm 0.04$ | $0.52 \pm 0.42$ | $0.26 \pm 0.10$ |
| (e) **IVW** | $0.10 \pm 0.03$ | $0.13 \pm 0.05$ | $0.13 \pm 0.05$ | $0.13 \pm 0.05$ | $0.13 \pm 0.05$ | $0.19 \pm 0.04$ | $0.30 \pm 0.11$ |
| (∗) **WO** (*ours*) | $0.06 \pm 0.02$ | $0.04 \pm 0.01$ | $0.06 \pm 0.03$ | $0.04 \pm 0.01$ | $0.09 \pm 0.04$ | $0.13 \pm 0.06$ | $0.18 \pm 0.07$ |
| Rel. improv. (%) | 37.4% | 65.8% | 49.4% | 66.9% | 27.7% | 30.6% | 28.7% |

Table 3: **Low-sample setting** $\mathcal{D}^N$: Reported are the average RMSEs $\pm$ standard deviation for CATE estimation, and the relative improvement of our **WO**-learner with decreasing sample size. Due to Neyman-orthogonality with respect to all nuisance functions, our **WO**-learner remains stable across all sample sizes.

ground-truth values for CATE in order to properly validate our results. We provide details on the data setup in Supplement D.2. In short, the data combines **all the difficulties** from our synthetic experiments: the data has *low overlap*, a *complex propensity score*, a *complex response function*, *low sample size*, and *time-varying confounding*.

Results: Table 4 shows the results for semi-synthetic data. Our **WO**-learner is the only method that remains stable for all prediction horizons. This confirms that our learner can deal with all the aforementioned difficulties in the time-varying setting, and clearly outperforms existing meta-learners.

| Prediction horizon | 2 | 3 | 4 |
|---|---|---|---|
| (b) **RA** | $0.12 \pm 0.01$ | $0.27 \pm 0.04$ | $0.40 \pm 0.00$ |
| (c) **IPW** | $0.66 \pm 0.50$ | $0.94 \pm 0.62$ | $5.23 \pm 6.55$ |
| (d) **DR** | $0.04 \pm 0.02$ | $0.20 \pm 0.11$ | $1.57 \pm 1.79$ |
| (e) **IVW** | $0.04 \pm 0.02$ | $31.27 \pm 44.02$ | $879.80 \pm 1243.54$ |
| (∗) **WO** (*ours*) | $\mathbf{0.03 \pm 0.01}$ | $\mathbf{0.15 \pm 0.01}$ | $\mathbf{0.17 \pm 0.07}$ |
| Rel. improv. (%) | 23.1% | 22.4% | 28.5% |

Table 4: **Semi-synthetic data**: Reported are the average RMSEs for CATE estimation. With all complexities in the time-varying setting combined, our **WO**-learner is the only meta-learner with consistent performance over all prediction horizons.

• **Real-world data:** We report results for real-world outcome estimation in Supplement G.

• **Ablations:** We repeat the experiment with decreasing overlap on the dataset $\mathcal{D}^\gamma$. Here, instantiate all meta-learners with simple LSTMs (Hochreiter & Schmidhuber, 1997). That is, we substitute the transformer instantiations for both the nuisance function and the second-stage estimator from the main experiments in Section 5 with LSTM architectures. As meta-learners can be instantiated with *any* neural backbone, this shows that our main results are **robust** for different instantiations.

| Overlap $\gamma$ | 0.5 | 1.0 | 1.5 | 2.0 | 2.5 | 3.0 | 3.5 | 4.0 | 4.5 | 5.0 | 5.5 | 6.0 | 6.5 |
|---|---|---|---|---|---|---|---|---|---|---|---|---|---|
| (a) **HA** | $0.55 \pm 0.06$ | $0.51 \pm 0.06$ | $0.54 \pm 0.10$ | $0.52 \pm 0.04$ | $0.50 \pm 0.05$ | $0.52 \pm 0.03$ | $0.51 \pm 0.04$ | $0.50 \pm 0.04$ | $0.47 \pm 0.03$ | $0.52 \pm 0.03$ | $0.52 \pm 0.01$ | $0.53 \pm 0.01$ | $0.57 \pm 0.01$ |
| (b) **RA** | $0.30 \pm 0.02$ | $0.31 \pm 0.05$ | $0.31 \pm 0.04$ | $0.31 \pm 0.03$ | $0.29 \pm 0.04$ | $0.27 \pm 0.02$ | $0.29 \pm 0.02$ | $0.28 \pm 0.02$ | $0.27 \pm 0.04$ | $0.25 \pm 0.04$ | $0.27 \pm 0.03$ | $0.28 \pm 0.03$ | $0.27 \pm 0.03$ |
| (c) **IPW** | $0.20 \pm 0.04$ | $0.12 \pm 0.06$ | $0.12 \pm 0.05$ | $0.18 \pm 0.08$ | $0.14 \pm 0.04$ | $0.24 \pm 0.07$ | $0.27 \pm 0.08$ | $0.52 \pm 0.35$ | $0.45 \pm 0.21$ | $0.35 \pm 0.12$ | $0.52 \pm 0.49$ | $0.72 \pm 0.04$ | $0.47 \pm 0.08$ |
| (d) **DR** | $\mathbf{0.05 \pm 0.03}$ | $\mathbf{0.04 \pm 0.01}$ | $0.06 \pm 0.02$ | $0.07 \pm 0.03$ | $0.09 \pm 0.04$ | $0.10 \pm 0.03$ | $0.09 \pm 0.03$ | $0.23 \pm 0.19$ | $0.13 \pm 0.06$ | $0.16 \pm 0.05$ | $0.28 \pm 0.12$ | $0.15 \pm 0.04$ | $0.18 \pm 0.08$ |
| (e) **IVW** | $0.72 \pm 1.26$ | $6.14 \pm 8.63$ | $0.21 \pm 0.22$ | $6.24 \pm 7.74$ | $3.51 \pm 6.77$ | $4.98 \pm 7.21$ | $9.79 \pm 6.22$ | $31.71 \pm 31.68$ | $138.57 \pm 165.75$ | $41.99 \pm 74.10$ | $61.49 \pm 67.81$ | $1351.68 \pm 1351.68$ | $76.66 \pm 76.66$ |
| (∗) **WO** (*ours*) | $0.07 \pm 0.01$ | $0.05 \pm 0.03$ | $0.03 \pm 0.02$ | $0.04 \pm 0.03$ | $0.04 \pm 0.01$ | $0.03 \pm 0.01$ | $0.06 \pm 0.02$ | $0.06 \pm 0.03$ | $0.08 \pm 0.03$ | $0.05 \pm 0.03$ | $0.08 \pm 0.03$ | $0.08 \pm 0.03$ | $0.08 \pm 0.03$ |
| Rel. improv. (%) | −29.9% | −7.8% | 46.0% | 37.5% | 54.8% | 68.2% | 37.0% | 74.7% | 36.0% | 66.6% | 72.6% | 47.1% | 54.3% |

Table 5: **Ablations**: Reported are the average RMSEs $\pm$ standard deviation for CATE estimation on $\mathcal{D}^\gamma$, and the relative improvement of our **WO**-learner over the best performing baselines across different levels of overlap.

Results: Table 5 shows that **WO**-learner again improves over all existing meta-learners for decreasing overlap. This is expected and in line with our theoretical findings, which are *agnostic* of the neural backbone. Hence, our novel **WO**-learner has clear advantages over existing standard meta-learners, *especially when the overlap is low* (large $\gamma$).

**Limitations:** In two edge cases, simpler meta-learners may suffice: (i) when nuisance functions are known or estimated with exceptional accuracy, in which case the DR learner's double robustness can be of advantage; and (ii) when propensities remain uniformly balanced, so that exponential overlap decay occurs at a lower rate and only becomes problematic for large $\tau$. Further, as is standard in the causal literature (Curth & van der Schaar, 2021; Frauen et al., 2025a; Melnychuk et al., 2022; Seedat et al., 2022), our method has been validated validated mainly on (semi-)synthetic data. Evaluating counterfactuals on real-world data remains an open research direction (Poinsot et al., 2025).

**Conclusion:** In this work, we present a novel weighted orthogonal meta-learner for HTE estimation over time. Therein, we introduce novel overlap weights that counteract exploding variance of existing meta-learners in low-overlap regimes, and we provide an orthogonalized population risk that is insensitive to nuisance misspecification. Our experiments confirm that our **WO**-learner presents an important step towards reliable decision-making for domains such as personalized medicine.

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

# A  DISCUSSION OF PRACTICAL CONSIDERATIONS WHEN TO USE OUR WO-LEARNER VS. OTHER META-LEARNERS

While our **WO**-learner outperforms existing meta-learners in low-overlap regimes, there are narrow scenarios in which simpler approaches may be adequate or equally effective.

**(i) When nuisance functions are known or exceptionally accurate.**  In this edge case, the classical DR learner benefits from its double robustness property: if either the outcome model or the propensity model is correctly specified, the DR estimator remains unbiased. Our WO-learner is Neyman-orthogonal but not doubly robust, so in the *rare scenario where practitioners have exceptionally accurate nuisance models*, DR may perform similarly or slightly better. However, in realistic longitudinal observational data, nuisance estimation is high-dimensional and difficult, which makes our **WO**-learner the more robust and stable choice.

**(ii) When treatment assignment exhibits uniformly strong overlap and prediction horizons are small.**  If propensities are far from the boundaries and do not degrade too quickly over time, the benefits of overlap weighting only occur for larger prediction horizons, i.e., the exponential rate at which overlap reduces is slightly lower. Hence, for small prediction horizons and very balanced overlap, this is a benign regime where simpler methods may be sufficient. Our experiments show, however, that realistic longitudinal datasets rarely exhibit low overlap, and hence the significant empirical advantage of our **WO**-learner.

Beyond these specific edge cases, we are not aware of meaningful practical conditions under which our **WO**-learner would not be favored. In high-dimensional or sparse-data regimes, our **WO**-learner remains well behaved: (i) orthogonality makes the method robust to nuisance mis-specification, and (ii) our theoretical results guarantee uniformly bounded pseudo-outcome variance even when propensities become small. This is reflected consistently in our empirical results.

## B Discussion: Identifying assumptions

Throughout our work, we rely on the identifying assumptions consistency, sequential ignorability, and positivity. These assumptions are the **standard** identifying conditions in longitudinal causal inference and are used uniformly across the literature on epidemiology and biostatistics (Robins, 1986; 1999; Robins et al., 2000; Robins & Hernán, 2009; van der Laan & Gruber, 2012; van der Laan & Rose, 2018), model-based methods (Bica et al., 2020; Hess & Feuerriegel, 2025; Li et al., 2021; Melnychuk et al., 2022; Seedat et al., 2022), and HTE meta-learners (Frauen et al., 2025a). In the following we discuss their practical meaning and why they are generally viewed as reasonable in real-world longitudinal applications.

• **Consistency:** Consistency requires that the observed outcome under the realized treatment equals the corresponding potential outcome. This is an uncontroversial assumption in causal inference. In practice, it holds whenever the treatment is clearly defined and does not vary in unmodeled ways across time or individuals. This condition is typically met in longitudinal applications such as medical dosing decisions, behavioral interventions, policy sequences, and recommendation systems, where the treatment at each step has an explicit operational definition.

• **Sequential ignorability:** Sequential ignorability assumes that, conditional on the observed covariate history $\bar{H}_t$, treatment assignment $A_t$ does not depend on unobserved factors that jointly affect future outcomes. This assumption is the *direct longitudinal analogue of standard unconfoundedness*, and is required by **all** existing approaches for estimating time-varying treatment effects. In practice, it is considered plausible because the covariate history $\bar{H}_t$ captures the clinician's or decision-maker's information set at the time of treatment. Longitudinal datasets in medicine, policy, and online recommendation systems typically record rich state variables, risk indicators, prior outcomes, and past responses, precisely to model how decisions are made. When these histories include the drivers of treatment choice (which modern datasets increasingly do), the assumption is regarded as reasonable.

• **Positivity:** Positivity requires that each treatment option has nonzero probability for the covariate histories of interest. This does not require equal or history-independent probabilities as in an RCT — only that each action occurs occasionally in practice. In practice, violations of positivity show up as near-deterministic treatment rules or extremely rare treatment paths. These issues are well-known in the causal literature and directly motivate our contribution: our **WO**-learner is designed to stabilize estimation in settings with limited but nonzero overlap by down-weighting sequences with very low overlap and up-weighting sequences with large overlap. As long as propensities do not collapse to zero, identification remains valid and our method addresses the resulting finite-sample instability.

• **Summary:** Overall, the identifying assumptions we rely on are the same assumptions used throughout the causal inference literature and are widely viewed as appropriate for longitudinal observational data. Our method operates under this well-established framework and specifically addresses the remaining practical difficulty of *limited overlap*, which is a finite-sample challenge that arises even when the underlying identifying assumptions hold.

# C PROOFS

## C.1 CONDITIONAL AVERAGE POTENTIAL OUTCOMES (CAPOS)

We split the following section into two parts: first, we derive several supporting lemmas to prove our main results ($\rightarrow$ Lemma C.1 to C.4). Then, we derive the theorem from the main paper ($\rightarrow$ Theorem C.5 and Theorem C.6).

### C.1.1 LEMMAS (CAPOS)

In order to prove our main theorems for CAPOs, we first introduce a **series of helping lemmas**.

**Lemma C.1.** *Let*

$$\gamma_t^{\bar{a}}(\bar{Z}_{t+\tau}) = \prod_{j=t}^{t+\tau} \frac{\mathbb{1}_{\{A_j=a_j\}}}{\pi_j^{\bar{a}}(\bar{H}_j)} Y_{t+\tau} + \sum_{j=t}^{t+\tau} \mu_j^{\bar{a}}\left(\bar{H}_j\right) \left(1 - \frac{\mathbb{1}_{\{A_j=a_j\}}}{\pi_j^{\bar{a}}(\bar{H}_j)}\right) \prod_{k=t}^{j-1} \frac{\mathbb{1}_{\{A_k=a_k\}}}{\pi_k^{\bar{a}}(\bar{H}_k)}, \quad (20)$$

*where*

$$\pi_j^{\bar{a}}(\bar{h}_j) = \mathbb{P}(A_j = a_j | \bar{H}_j = \bar{h}_j) \quad (21)$$

*is the propensity score, and*

$$\mu_{t+\tau}^{\bar{a}}(\bar{h}_{t+\tau}) = \mathbb{E}\left[Y_{t:t+\tau} \mid \bar{H}_{t+\tau} = \bar{h}_{t+\tau}, A_{t+\tau} = a_{t+\tau}\right] \quad (22)$$

*and*

$$\mu_j^{\bar{a}}(\bar{h}_j) = \mathbb{E}\left[\mu_{j+1}^{\bar{a}}(\bar{h}_{j+1}) \mid \bar{H}_j = \bar{h}_j, A_j = a_j\right], \quad (23)$$

*such that*

$$\mu_t^{\bar{a}}(\bar{h}_t) = \mathbb{E}\left[Y_{t:t+\tau}[a_{t:t+\tau}] \mid \bar{H}_t = \bar{h}_t\right] \quad (24)$$

*is the conditional average potential outcome. Then, $\gamma_t^{\bar{a}}(\bar{Z}_{t+\tau})$ is Neyman-orthogonal with respect to all nuisance functions $\eta^{\bar{a}} = \{\pi_j^{\bar{a}}, \mu_j^{\bar{a}}, \omega_j^{\bar{a}}\}_{j=t}^{t+\tau}$.*

*Proof.* First, $\gamma_t^{\bar{a}}(\bar{Z}_{t+\tau})$ is trivially Neyman-orthogonal with respect to $\omega_j^{\bar{a}}$ as it is independent of it. Second, we notice that $\gamma_t^{\bar{a}}(\bar{Z}_{t+\tau})$ is the uncentered efficient influence function of $\mathbb{E}\left[\mu_t^{\bar{a}}(\bar{H}_t)\right]$ (Frauen et al., 2025a; van der Laan & Gruber, 2012), which is Neyman-orthogonal with respect to all nuisance functions. $\square$

**Lemma C.2.** *Let*

$$\rho_t^{\bar{a}}(\bar{Z}_{t+\tau}) = \prod_{j=t}^{t+\tau} \pi_j^{\bar{a}}(\bar{H}_j) + \sum_{j=t}^{t+\tau} \left( \mathbb{1}_{\{a_j = A_j\}} - \pi_j^{\bar{a}}(\bar{H}_j) \right) \omega_{j+1}^{\bar{a}}(\bar{H}_j) \prod_{t \le k < j} \pi_k^{\bar{a}}(\bar{H}_k), \tag{25}$$

*where*

$$\pi_j^{\bar{a}}(\bar{h}_j) = \mathbb{P}(A_j = a_j | \bar{H}_j = \bar{h}_j) \tag{26}$$

*is the propensity score, and*

$$\omega_j^{\bar{a}}(\bar{h}_\ell) = p(A_{j:t+\tau} = a_{j:t+\tau} \mid \bar{H}_\ell = \bar{h}_\ell) = \mathbb{E}\left[ \prod_{k=j}^{t+\tau} \pi_k^{\bar{a}}(\bar{H}_k) \, \Big| \, \bar{H}_\ell = \bar{h}_\ell \right] \tag{27}$$

*is the weight function. Then, $\rho_t^{\bar{a}}(\bar{Z}_{t+\tau})$ is Neyman-orthogonal with respect to all nuisance functions $\eta^{\bar{a}} = \{\pi_j^{\bar{a}}, \mu_j^{\bar{a}}, \omega_j^{\bar{a}}\}_{j=t}^{t+\tau}$.*

*Proof.* First, $\rho_t^{\bar{a}}(\bar{Z}_{t+\tau})$ is trivially Neyman-orthogonal with respect to $\mu_j^{\bar{a}}$ as it is independent of it. Second, we show that $\rho_t^{\bar{a}}(\bar{Z}_{t+\tau})$ is the uncentered efficient influence function of $\mathbb{E}\left[\omega_t^{\bar{a}}(\bar{H}_t)\right]$, and hence, Neyman-orthogonal with respect to the nuisance functions $\{\pi_j^{\bar{a}}, \omega_j^{\bar{a}}\}_{j=t}^{t+\tau}$. For this, we make use of the chain rule for pathwise derivatives (Kennedy, 2022; Luedtke, 2024). First, we compute the efficient influence function of $\omega_t^{\bar{a}}(\bar{h}_t)$ via

$$\mathbb{IF}\left(\omega_t^{\bar{a}}(\bar{h}_t)\right) \tag{28}$$

$$= \mathbb{IF}\left(\mathbb{E}\left[\prod_{j=t}^{t+\tau} \pi_j^{\bar{a}}(\bar{H}_j) \, \Big| \, \bar{H}_t = \bar{h}_t \right]\right) \tag{29}$$

$$= \mathbb{IF}\left(\sum_{h_{t+1:t+\tau}} p(h_{t+1:t+\tau} \mid \bar{h}_t) \prod_{j=t}^{t+\tau} \pi_j^{\bar{a}}(\bar{h}_j)\right) \tag{30}$$

$$= \sum_{h_{t+1:t+\tau}} \left[ \underbrace{\mathbb{IF}\left(p(h_{t+1:t+\tau} \mid \bar{h}_t)\right) \prod_{j=t}^{t+\tau} \pi_j^{\bar{a}}(\bar{h}_j)}_{(*)} + \underbrace{p(h_{t+1:t+\tau} \mid \bar{h}_t)\mathbb{IF}\left(\prod_{j=t}^{t+\tau} \pi_j^{\bar{a}}(\bar{h}_j)\right)}_{(**)} \right]. \tag{31}$$

For $(*)$, we have that

$$\mathbb{IF}\left(p(h_{t+1:t+\tau} \mid \bar{h}_t)\right) \prod_{j=t}^{t+\tau} \pi_j^{\bar{a}}(\bar{h}_j) \tag{32}$$

$$= \mathbb{IF}\left(p(\bar{h}_{t+\tau})/p(\bar{h}_t)\right) \prod_{j=t}^{t+\tau} \pi_j^{\bar{a}}(\bar{h}_j) \tag{33}$$

$$= \frac{\mathbb{1}_{\{\bar{h}_t = \bar{H}_t\}}}{p(\bar{h}_t)} \left( \mathbb{1}_{\{h_{t+1:t+\tau} = H_{t+1:t+\tau}\}} - p(h_{t+1:t+\tau} | \bar{h}_t) \right) \prod_{j=t}^{t+\tau} \pi_j^{\bar{a}}(\bar{h}_j). \tag{34}$$

Further, we obtain $(\ast\ast)$ via

$$p(h_{t+1:t+\tau} \mid \bar{h}_t) \mathbb{IF}\Big( \prod_{j=t}^{t+\tau} \pi_j^{\bar{a}}(\bar{h}_j) \Big) \tag{35}$$

$$= p(h_{t+1:t+\tau} \mid \bar{h}_t) \sum_{j=t}^{t+\tau} \mathbb{IF}\Big( \pi_j^{\bar{a}}(\bar{h}_j) \Big) \prod_{k \neq j} \pi_k^{\bar{a}}(\bar{h}_k) \tag{36}$$

$$= p(h_{t+1:t+\tau} \mid \bar{h}_t) \sum_{j=t}^{t+\tau} \frac{\mathbb{1}_{\{\bar{h}_j = \bar{H}_j\}}}{p(\bar{h}_j)} \Big( \mathbb{1}_{\{a_j = A_j\}} - \pi_j^{\bar{a}}(\bar{h}_j) \Big) \prod_{k \neq j} \pi_k^{\bar{a}}(\bar{h}_k) \tag{37}$$

$$= p(h_{t+1:t+\tau} \mid \bar{h}_t) \sum_{j=t}^{t+\tau} \frac{\mathbb{1}_{\{\bar{h}_t = \bar{H}_t\}} \mathbb{1}_{\{h_{t+1:j} = H_{t+1:j}\}}}{p(\bar{h}_j)} \Big( \mathbb{1}_{\{a_j = A_j\}} - \pi_j^{\bar{a}}(\bar{h}_j) \Big) \prod_{k \neq j} \pi_k^{\bar{a}}(\bar{h}_k) \tag{38}$$

$$= \frac{\mathbb{1}_{\{\bar{h}_t = \bar{H}_t\}}}{p(\bar{h}_t)} \sum_{j=t}^{t+\tau} \mathbb{1}_{\{h_{t+1:j} = H_{t+1:j}\}} p(h_{j+1:t+\tau} \mid \bar{h}_j) \Big( \mathbb{1}_{\{a_j = A_j\}} - \pi_j^{\bar{a}}(\bar{h}_j) \Big) \prod_{k \neq j} \pi_k^{\bar{a}}(\bar{h}_k) \tag{39}$$

Combining both results, the efficient influence function of $\omega_t^{\bar{a}}(\bar{h}_t)$ is given by

$$\mathbb{IF}\Big( \omega_t^{\bar{a}}(\bar{h}_t) \Big) \tag{40}$$

$$= \frac{\mathbb{1}_{\{\bar{h}_t = \bar{H}_t\}}}{p(\bar{h}_t)} \sum_{h_{t+1:t+\tau}} \Bigg[ \Big( \mathbb{1}_{\{h_{t+1:t+\tau} = H_{t+1:t+\tau}\}} - p(h_{t+1:t+\tau} | \bar{h}_t) \Big) \prod_{j=t}^{t+\tau} \pi_j^{\bar{a}}(\bar{h}_j) \tag{41}$$

$$+ \sum_{j=t}^{t+\tau} \mathbb{1}_{\{h_{t+1:j} = H_{t+1:j}\}} p(h_{j+1:t+\tau} \mid \bar{h}_j) \Big( \mathbb{1}_{\{a_j = A_j\}} - \pi_j^{\bar{a}}(\bar{h}_j) \Big) \prod_{k \neq j} \pi_k^{\bar{a}}(\bar{h}_k) \Bigg] \tag{42}$$

$$= \frac{\mathbb{1}_{\{\bar{h}_t = \bar{H}_t\}}}{p(\bar{h}_t)} \sum_{h_{t+1:t+\tau}} \Bigg[ \Big( \mathbb{1}_{\{h_{t+1:t+\tau} = H_{t+1:t+\tau}\}} - p(h_{t+1:t+\tau} | \bar{h}_t) \Big) \prod_{j=t}^{t+\tau} \pi_j^{\bar{a}}(\bar{h}_j) \tag{43}$$

$$+ \sum_{j=t}^{t+\tau} \mathbb{1}_{\{h_{t+1:j} = H_{t+1:j}\}} p(h_{j+1:t+\tau} \mid \bar{h}_j) \Big( \mathbb{1}_{\{a_j = A_j\}} - \pi_j^{\bar{a}}(\bar{h}_j) \Big) \tag{44}$$

$$\times \prod_{t \leq k < j} \pi_k^{\bar{a}}(\bar{h}_k) \prod_{k > j} \pi_k^{\bar{a}}(\bar{h}_k) \Bigg] \tag{45}$$

$$= \frac{\mathbb{1}_{\{\bar{h}_t = \bar{H}_t\}}}{p(\bar{h}_t)} \Bigg[ \prod_{j=t}^{t+\tau} \pi_j^{\bar{a}}(H_{t+1:j}, \bar{h}_t) - \mathbb{E}\Big[ \prod_{j=t}^{t+\tau} \pi_j^{\bar{a}}(\bar{H}_j) \; \Big| \; \bar{H}_t = \bar{h}_t \Big] \tag{46}$$

$$+ \sum_{j=t}^{t+\tau} \Big( \mathbb{1}_{\{a_j = A_j\}} - \pi_j^{\bar{a}}(H_{t+1:j}, \bar{h}_t) \Big) \tag{47}$$

$$\times \prod_{t \leq k < j} \pi_k^{\bar{a}}(H_{t+1:k}, \bar{h}_t) \mathbb{E}\Big[ \prod_{k > j} \pi_k^{\bar{a}}(\bar{H}_k) \; \Big| \; H_{t+1:j}, \bar{H}_t = \bar{h}_t \Big] \Bigg] \tag{48}$$

$$= \frac{\mathbb{1}_{\{\bar{h}_t = \bar{H}_t\}}}{p(\bar{h}_t)} \Bigg[ - \omega_t^{\bar{a}}(\bar{h}_t) + \prod_{j=t}^{t+\tau} \pi_j^{\bar{a}}(H_{t+1:j}, \bar{h}_t) + \sum_{j=t}^{t+\tau} \Big( \mathbb{1}_{\{a_j = A_j\}} - \pi_j^{\bar{a}}(H_{t+1:j}, \bar{h}_t) \Big) \tag{49}$$

$$\times \prod_{t \leq k < j} \pi_k^{\bar{a}}(H_{t+1:k}, \bar{h}_t) \mathbb{E}\Big[ \prod_{k > j} \pi_k^{\bar{a}}(\bar{H}_k) \; \Big| \; H_{t+1:j}, \bar{H}_t = \bar{h}_t \Big] \Bigg]. \tag{50}$$

Using this result, we derive the efficient influence function of $\mathbb{E}\left[\omega_t^{\bar{a}}(\bar{H}_t)\right]$ via

$$\mathbb{IF}\left(\mathbb{E}\left[\omega_t^{\bar{a}}(\bar{H}_t)\right]\right) \tag{51}$$

$$=\sum_{\bar{h}_t} \mathbb{IF}\left(p(\bar{h}_t)\omega_t^{\bar{a}}(\bar{h}_t)\right) \tag{52}$$

$$=\sum_{\bar{h}_t}\left[\mathbb{IF}\left(p(\bar{h}_t)\right)\omega_t^{\bar{a}}(\bar{h}_t) + p(\bar{h}_t)\mathbb{IF}\left(\omega_t^{\bar{a}}(\bar{h}_t)\right)\right] \tag{53}$$

$$=\omega_t^{\bar{a}}(\bar{H}_t) - \mathbb{E}\left[\omega_t^{\bar{a}}(\bar{H}_t)\right] - \omega_t^{\bar{a}}(\bar{H}_t) \tag{54}$$

$$+ \prod_{j=t}^{t+\tau} \pi_j^{\bar{a}}(\bar{H}_j) + \sum_{j=t}^{t+\tau}\left(\mathbb{1}_{\{a_j=A_j\}} - \pi_j^{\bar{a}}(\bar{H}_j)\right)\prod_{t\le k<j}\pi_k^{\bar{a}}(\bar{H}_k)\underbrace{\mathbb{E}\left[\prod_{k>j}\pi_k^{\bar{a}}(\bar{H}_k)\ \middle|\ \bar{H}_j\right]}_{=\omega_{j+1}^{\bar{a}}(\bar{H}_j)} \tag{55}$$

$$= -\mathbb{E}\left[\omega_t^{\bar{a}}(\bar{H}_t)\right] + \rho_t^{\bar{a}}(\bar{Z}_{t+\tau}), \tag{56}$$

which concludes the proof. $\qquad\square$

**Lemma C.3.** *Let*

$$\gamma_t^{\bar{a}}(\bar{Z}_{t+\tau}) = \prod_{j=t}^{t+\tau} \frac{\mathbb{1}_{\{A_j=a_j\}}}{\pi_j^{\bar{a}}(\bar{H}_j)} Y_{t+\tau} + \sum_{j=t}^{t+\tau} \mu_j^{\bar{a}}(\bar{H}_j) \left(1 - \frac{\mathbb{1}_{\{A_j=a_j\}}}{\pi_j^{\bar{a}}(\bar{H}_j)}\right) \prod_{k=t}^{j-1} \frac{\mathbb{1}_{\{A_k=a_k\}}}{\pi_k^{\bar{a}}(\bar{H}_k)}, \quad (57)$$

*where*

$$\pi_j^{\bar{a}}(\bar{h}_j) = \mathbb{P}(A_j = a_j \mid \bar{H}_j = \bar{h}_j) \quad (58)$$

*is the propensity score, and*

$$\mu_{t+\tau}^{\bar{a}}(\bar{h}_{t+\tau}) = \mathbb{E}\left[Y_{t:t+\tau} \;\middle|\; \bar{H}_{t+\tau} = \bar{h}_{t+\tau}, A_{t+\tau} = a_{t+\tau}\right] \quad (59)$$

*and*

$$\mu_j^{\bar{a}}(\bar{h}_j) = \mathbb{E}\left[\mu_{j+1}^{\bar{a}}(\bar{h}_{j+1}) \;\middle|\; \bar{H}_j = \bar{h}_j, A_j = a_j\right], \quad (60)$$

*such that*

$$\mu_t^{\bar{a}}(\bar{h}_t) = \mathbb{E}\left[Y_{t:t+\tau}[a_{t:t+\tau}] \;\middle|\; \bar{H}_t = \bar{h}_t\right] \quad (61)$$

*is the conditional average potential outcome. Then, it holds that*

$$\mathbb{E}\left[\gamma_t^{\bar{a}}(\bar{Z}_{t+\tau}) \;\middle|\; \bar{H}_t\right] = \mu_t^{\bar{a}}(\bar{H}_t). \quad (62)$$

*Proof.*

$$\mathbb{E}\left[\gamma_t^{\bar{a}}(\bar{Z}_{t+\tau}) \;\middle|\; \bar{H}_t\right] \quad (63)$$

$$= \mathbb{E}\left[\prod_{j=t}^{t+\tau} \frac{\mathbb{1}_{\{A_j=a_j\}}}{\pi_j^{\bar{a}}(\bar{H}_j)} Y_{t+\tau} + \sum_{j=t}^{t+\tau} \mu_j^{\bar{a}}(\bar{H}_j) \left(1 - \frac{\mathbb{1}_{\{A_j=a_j\}}}{\pi_j^{\bar{a}}(\bar{H}_j)}\right) \prod_{k=t}^{j-1} \frac{\mathbb{1}_{\{A_k=a_k\}}}{\pi_k^{\bar{a}}(\bar{H}_k)} \;\middle|\; \bar{H}_t\right] \quad (64)$$

$$= \underbrace{\mathbb{E}\left[\prod_{j=t}^{t+\tau} \frac{\mathbb{1}_{\{A_j=a_j\}}}{\pi_j^{\bar{a}}(\bar{H}_j)} Y_{t+\tau} \;\middle|\; \bar{H}_t\right]}_{=\mu_t^{\bar{a}}(\bar{H}_t)} \quad (65)$$

$$+ \sum_{j=t}^{t+\tau} \mathbb{E}\left[\mathbb{E}\left[\mu_j^{\bar{a}}(\bar{H}_j) \left(1 - \frac{\mathbb{1}_{\{A_j=a_j\}}}{\pi_j^{\bar{a}}(\bar{H}_j)}\right) \prod_{k=t}^{j-1} \frac{\mathbb{1}_{\{A_k=a_k\}}}{\pi_k^{\bar{a}}(\bar{H}_k)} \;\middle|\; \bar{H}_j\right] \;\middle|\; \bar{H}_t\right] \quad (66)$$

$$= \mu_t^{\bar{a}}(\bar{H}_t) + \sum_{j=t}^{t+\tau} \mathbb{E}\left[\mu_j^{\bar{a}}(\bar{H}_j) \left(1 - \frac{\mathbb{E}[\mathbb{1}_{\{A_j=a_j\}}|\bar{H}_j]}{\pi_j^{\bar{a}}(\bar{H}_j)}\right) \prod_{k=t}^{j-1} \frac{\mathbb{1}_{\{A_k=a_k\}}}{\pi_k^{\bar{a}}(\bar{H}_k)} \;\middle|\; \bar{H}_t\right] \quad (67)$$

$$= \mu_t^{\bar{a}}(\bar{H}_t) + \sum_{j=t}^{t+\tau} \mathbb{E}\left[\mu_j^{\bar{a}}(\bar{H}_j) \left(1 - \frac{\pi_j^{\bar{a}}(\bar{H}_j)}{\pi_j^{\bar{a}}(\bar{H}_j)}\right) \prod_{k=t}^{j-1} \frac{\mathbb{1}_{\{A_k=a_k\}}}{\pi_k^{\bar{a}}(\bar{H}_k)} \;\middle|\; \bar{H}_t\right] \quad (68)$$

$$= \mu_t^{\bar{a}}(\bar{H}_t). \quad (69)$$

The DR-pseudo outcomes, which are a sub-component of our pseudo-outcomes, have been introduced in Frauen et al. (2025a); van der Laan & Gruber (2012). □

**Lemma C.4.** *Let*

$$\rho_t^{\bar{a}}(\bar{Z}_{t+\tau}) = \prod_{j=t}^{t+\tau} \pi_j^{\bar{a}}(\bar{H}_j) + \sum_{j=t}^{t+\tau} \left( \mathbb{1}_{\{a_j = A_j\}} - \pi_j^{\bar{a}}(\bar{H}_j) \right) \omega_{j+1}^{\bar{a}}(\bar{H}_j) \prod_{t \le k < j} \pi_k^{\bar{a}}(\bar{H}_k), \qquad (70)$$

*where*

$$\pi_j^{\bar{a}}(\bar{h}_j) = \mathbb{P}(A_j = a_j | \bar{H}_j = \bar{h}_j) \qquad (71)$$

*is the propensity score, and*

$$\omega_j^{\bar{a}}(\bar{h}_\ell) = p(A_{j:t+\tau} = a_{j:t+\tau} \mid \bar{H}_\ell = \bar{h}_\ell) = \mathbb{E}\Big[ \prod_{k=j}^{t+\tau} \pi_k^{\bar{a}}(\bar{H}_k) \,\Big|\, \bar{H}_\ell = \bar{h}_\ell \Big] \qquad (72)$$

*is the weight function. Then, it holds that*

$$\mathbb{E}\Big[ \rho_t^{\bar{a}}(\bar{Z}_{t+\tau}) \,\Big|\, \bar{H}_t \Big] = \omega_t^{\bar{a}}(\bar{H}_t). \qquad (73)$$

*Proof.*

$$\mathbb{E}\Big[ \rho_t^{\bar{a}}(\bar{Z}_{t+\tau}) \,\Big|\, \bar{H}_t \Big] \qquad (74)$$

$$= \mathbb{E}\Big[ \prod_{j=t}^{t+\tau} \pi_j^{\bar{a}}(\bar{H}_j) + \sum_{j=t}^{t+\tau} \left( \mathbb{1}_{\{a_j = A_j\}} - \pi_j^{\bar{a}}(\bar{H}_j) \right) \omega_{j+1}^{\bar{a}}(\bar{H}_j) \prod_{t \le k < j} \pi_k^{\bar{a}}(\bar{H}_k) \,\Big|\, \bar{H}_t \Big] \qquad (75)$$

$$= \underbrace{\mathbb{E}\Big[ \prod_{j=t}^{t+\tau} \pi_j^{\bar{a}}(\bar{H}_j) \,\Big|\, \bar{H}_t \Big]}_{=\omega_t^{\bar{a}}(\bar{H}_t)} + \mathbb{E}\Big[ \sum_{j=t}^{t+\tau} \left( \mathbb{1}_{\{a_j = A_j\}} - \pi_j^{\bar{a}}(\bar{H}_j) \right) \omega_{j+1}^{\bar{a}}(\bar{H}_j) \prod_{t \le k < j} \pi_k^{\bar{a}}(\bar{H}_k) \,\Big|\, \bar{H}_t \Big] \qquad (76)$$

$$= \omega_t^{\bar{a}}(\bar{H}_t) + \sum_{j=t}^{t+\tau} \mathbb{E}\left[ \mathbb{E}\Big[ \left( \mathbb{1}_{\{a_j = A_j\}} - \pi_j^{\bar{a}}(\bar{H}_j) \right) \omega_{j+1}^{\bar{a}}(\bar{H}_j) \prod_{t \le k < j} \pi_k^{\bar{a}}(\bar{H}_k) \,\Big|\, \bar{H}_j \Big] \,\Big|\, \bar{H}_t \right] \qquad (77)$$

$$= \omega_t^{\bar{a}}(\bar{H}_t) + \sum_{j=t}^{t+\tau} \mathbb{E}\left[ \left( \mathbb{E}\Big[ \mathbb{1}_{\{a_j = A_j\}} \,\Big|\, \bar{H}_j \Big] - \pi_j^{\bar{a}}(\bar{H}_j) \right) \omega_{j+1}^{\bar{a}}(\bar{H}_j) \prod_{t \le k < j} \pi_k^{\bar{a}}(\bar{H}_k) \,\Big|\, \bar{H}_t \right] \qquad (78)$$

$$= \omega_t^{\bar{a}}(\bar{H}_t) + \sum_{j=t}^{t+\tau} \mathbb{E}\left[ \left( \pi_j^{\bar{a}}(\bar{H}_j) - \pi_j^{\bar{a}}(\bar{H}_j) \right) \omega_{j+1}^{\bar{a}}(\bar{H}_j) \prod_{t \le k < j} \pi_k^{\bar{a}}(\bar{H}_k) \,\Big|\, \bar{H}_t \right] \qquad (79)$$

$$= \omega_t^{\bar{a}}(\bar{H}_t). \qquad (80)$$

$\square$

### C.1.2 THEOREMS (CAPOS)

We now prove the CAPO version of our theorems from the main paper. For both proofs, we leverage additional helping lemmas that we derived in Supplement C.1.1.

**Theorem C.5** (Weighted population risk (CAPO)). *Let*

$$\xi_t^{\bar{a}}(\bar{Z}_{t+\tau}) = \mu_t^{\bar{a}}(\bar{H}_t) + \frac{\omega_t^{\bar{a}}(\bar{H}_t)}{\rho_t^{\bar{a}}(\bar{Z}_{t+\tau})}\Big(\gamma_t^{\bar{a}}(\bar{Z}_{t+\tau}) - \mu_t^{\bar{a}}(\bar{H}_t)\Big), \tag{81}$$

*where*

$$\rho_t^{\bar{a}}(\bar{Z}_{t+\tau}) = \prod_{j=t}^{t+\tau} \pi_j^{\bar{a}}(\bar{H}_j) + \sum_{j=t}^{t+\tau}\Big(\mathbb{1}_{\{a_j=A_j\}} - \pi_j^{\bar{a}}(\bar{H}_j)\Big)\omega_{j+1}^{\bar{a}}(\bar{H}_j)\prod_{t\leq k<j}\pi_k^{\bar{a}}(\bar{H}_k), \tag{82}$$

*and*

$$\gamma_t^{\bar{a}}(\bar{Z}_{t+\tau}) = \prod_{j=t}^{t+\tau}\frac{\mathbb{1}_{\{A_j=a_j\}}}{\pi_j^{\bar{a}}(\bar{H}_j)}Y_{t+\tau} + \sum_{j=t}^{t+\tau}\mu_j^{\bar{a}}\left(\bar{H}_j\right)\left(1 - \frac{\mathbb{1}_{\{A_j=a_j\}}}{\pi_j^{\bar{a}}(\bar{H}_j)}\right)\prod_{k=t}^{j-1}\frac{\mathbb{1}_{\{A_k=a_k\}}}{\pi_k^{\bar{a}}(\bar{H}_k)}, \tag{83}$$

*with $\bar{Z}_{t+\tau} = (\bar{H}_{t+\tau}, A_{t+\tau}, Y_{t+\tau})$. Then, the population risk function*

$$\mathcal{L}(g; \eta^{\bar{a}}) = \frac{1}{\mathbb{E}\Big[\omega_t^{\bar{a}}(\bar{H}_t)\Big]}\mathbb{E}\left[\rho_t^{\bar{a}}(\bar{Z}_{t+\tau})\Big(\xi_t^{\bar{a}}(\bar{Z}_{t+\tau}) - g(\bar{H}_t)\Big)^2\right] \tag{84}$$

*minimizes the oracle risk*

$$\mathcal{L}^*(g; \eta^{\bar{a}}) = \frac{1}{\mathbb{E}\Big[\omega_t^{\bar{a}}(\bar{H}_t)\Big]}\mathbb{E}\left[\omega_t^{\bar{a}}(\bar{H}_t)\Big(\mu_t^{\bar{a}}(\bar{H}_t) - g(\bar{H}_t)\Big)^2\right]. \tag{85}$$

*Proof.* In order to show that $\mathcal{L}(g; \eta^{\bar{a}})$ and $\mathcal{L}^*(g; \eta^{\bar{a}})$ have the same minimizer $g$, we need to show that

$$\mathbb{E}\left[\rho_t^{\bar{a}}(\bar{Z}_{t+\tau})\Big(\xi_t^{\bar{a}}(\bar{Z}_{t+\tau}) - g(\bar{H}_t)\Big)^2\right] = \mathbb{E}\left[\omega_t^{\bar{a}}(\bar{H}_t)\Big(\mu_t^{\bar{a}}(\bar{H}_t) - g(\bar{H}_t)\Big)^2\right] + C, \tag{86}$$

where $C$ is some constant term that does **not** depend on $g$. For this, notice that

$$\mathbb{E}\left[\rho_t^{\bar{a}}(\bar{Z}_{t+\tau})\Big(\xi_t^{\bar{a}}(\bar{Z}_{t+\tau}) - g(\bar{H}_t)\Big)^2\right] \tag{87}$$

$$=\mathbb{E}\left[\rho_t^{\bar{a}}(\bar{Z}_{t+\tau})\Big(\xi_t^{\bar{a}}(\bar{Z}_{t+\tau}) - \mu_t^{\bar{a}}(\bar{H}_t) + \mu_t^{\bar{a}}(\bar{H}_t) - g(\bar{H}_t)\Big)^2\right] \tag{88}$$

$$=\underbrace{\mathbb{E}\left[\rho_t^{\bar{a}}(\bar{Z}_{t+\tau})\Big(\xi_t^{\bar{a}}(\bar{Z}_{t+\tau}) - \mu_t^{\bar{a}}(\bar{H}_t)\Big)^2\right] + 2\mathbb{E}\left[\rho_t^{\bar{a}}(\bar{Z}_{t+\tau})\Big(\xi_t^{\bar{a}}(\bar{Z}_{t+\tau}) - \mu_t^{\bar{a}}(\bar{H}_t)\Big)\mu_t^{\bar{a}}(\bar{H}_t)\right]}_{=C} \tag{89}$$

$$- 2\mathbb{E}\left[\rho_t^{\bar{a}}(\bar{Z}_{t+\tau})\Big(\xi_t^{\bar{a}}(\bar{Z}_{t+\tau}) - \mu_t^{\bar{a}}(\bar{H}_t)\Big)g(\bar{H}_t)\right] + \mathbb{E}\left[\rho_t^{\bar{a}}(\bar{Z}_{t+\tau})\Big(\mu_t^{\bar{a}}(\bar{H}_t) - g(\bar{H}_t)\Big)^2\right]. \tag{90}$$

Here, the first two terms do not depend on $g$ and are therefore constant. Next we focus on

$$\mathbb{E}\left[\rho_t^{\bar{a}}(\bar{Z}_{t+\tau})\Big(\xi_t^{\bar{a}}(\bar{Z}_{t+\tau}) - \mu_t^{\bar{a}}(\bar{H}_t)\Big)g(\bar{H}_t)\right] \tag{91}$$

$$=\mathbb{E}\left[\mathbb{E}\left[\rho_t^{\bar{a}}(\bar{Z}_{t+\tau})\Big(\xi_t^{\bar{a}}(\bar{Z}_{t+\tau}) - \mu_t^{\bar{a}}(\bar{H}_t)\Big)g(\bar{H}_t) \mid \bar{H}_t\right]\right] \tag{92}$$

$$=\mathbb{E}\left[\mathbb{E}\left[\rho_t^{\bar{a}}(\bar{Z}_{t+\tau})\Big(\mu_t^{\bar{a}}(\bar{H}_t) + \frac{\omega_t^{\bar{a}}(\bar{H}_t)}{\rho_t^{\bar{a}}(\bar{Z}_{t+\tau})}\Big(\gamma_t^{\bar{a}}(\bar{Z}_{t+\tau}) - \mu_t^{\bar{a}}(\bar{H}_t)\Big) - \mu_t^{\bar{a}}(\bar{H}_t)\Big)g(\bar{H}_t) \mid \bar{H}_t\right]\right] \tag{93}$$

$$=\mathbb{E}\left[\mathbb{E}\left[\omega_t^{\bar{a}}(\bar{H}_t)\Big(\gamma_t^{\bar{a}}(\bar{Z}_{t+\tau}) - \mu_t^{\bar{a}}(\bar{H}_t)\Big)g(\bar{H}_t) \mid \bar{H}_t\right]\right] \tag{94}$$

$$=\mathbb{E}\left[\omega_t^{\bar{a}}(\bar{H}_t)\Big(\underbrace{\mathbb{E}\left[\gamma_t^{\bar{a}}(\bar{Z}_{t+\tau}) \mid \bar{H}_t\right]}_{=\mu_t^{\bar{a}}(\bar{H}_t)} - \mu_t^{\bar{a}}(\bar{H}_t)\Big)g(\bar{H}_t)\right] \tag{95}$$

$$=0, \tag{96}$$

where the result we apply in Equation 95 follows from Lemma C.3.

Finally, we focus on the last term in Equation 90. That is,

$$\mathbb{E}\left[\rho_t^{\bar{a}}(\bar{Z}_{t+\tau})\Big(\mu_t^{\bar{a}}(\bar{H}_t) - g(\bar{H}_t)\Big)^2\right] \tag{97}$$

$$=\mathbb{E}\left[\mathbb{E}\left[\rho_t^{\bar{a}}(\bar{Z}_{t+\tau})\Big(\mu_t^{\bar{a}}(\bar{H}_t) - g(\bar{H}_t)\Big)^2 \mid \bar{H}_t\right]\right] \tag{98}$$

$$=\mathbb{E}\left[\underbrace{\mathbb{E}\left[\rho_t^{\bar{a}}(\bar{Z}_{t+\tau}) \mid \bar{H}_t\right]}_{=\omega_t^{\bar{a}}(\bar{H}_t)}\Big(\mu_t^{\bar{a}}(\bar{H}_t) - g(\bar{H}_t)\Big)^2\right] \tag{99}$$

$$=\mathbb{E}[\omega_t^{\bar{a}}(\bar{H}_t)]\mathcal{L}^*(g; \eta^{\bar{a}}), \tag{100}$$

where the result we apply in Equation 99 follows from Lemma C.4.

Hence, combining Equation 90 with Equation 96 and Equation 100, and multiplying with $1/\mathbb{E}[\omega_t^{\bar{a}}(\bar{H}_t)]$ yields

$$\mathcal{L}(g; \eta^{\bar{a}}) = \mathcal{L}^*(g; \eta^{\bar{a}}) + C, \tag{101}$$

which proves the theorem. $\square$

**Theorem C.6** (Neyman-orthogonality (CAPO)). *The weighted population risk*

$$\mathcal{L}(g;\eta^{\bar{a}}) = \frac{1}{\mathbb{E}\Big[\omega_t^{\bar{a}}(\bar{H}_t)\Big]}\mathbb{E}\Big[\rho_t^{\bar{a}}(\bar{Z}_{t+\tau})\Big(\xi_t^{\bar{a}}(\bar{Z}_{t+\tau}) - g(\bar{H}_t)\Big)^2\Big] \tag{102}$$

*is Neyman-orthogonal with respect to all nuisance functions* $\eta^{\bar{a}} = \{\pi_j^{\bar{a}}, \mu_j^{\bar{a}}, \omega_j^{\bar{a}}\}_{j=t}^{t+\tau}$.

*Proof.* In order to show Neyman-orthogonality, we first calculate the pathwise-derivative with respect to the first argument, i.e., the target parameter $g$, via

$$D_g\mathcal{L}(g;\eta^{\bar{a}})[\hat{g} - g] \tag{103}$$

$$\propto \frac{\mathrm{d}}{\mathrm{d}r}\mathbb{E}\Big[\rho_t^{\bar{a}}(\bar{Z}_{t+\tau})\Big(\xi_t^{\bar{a}}(\bar{Z}_{t+\tau}) - \Big[g(\bar{H}_t) + r\{\hat{g}(\bar{H}_t) - g(\bar{H}_t)\}\Big]\Big)^2\Big]\Big|_{r=0} \tag{104}$$

$$= -2\mathbb{E}\Big[\rho_t^{\bar{a}}(\bar{Z}_{t+\tau})\Big(\xi_t^{\bar{a}}(\bar{Z}_{t+\tau}) - \Big[g(\bar{H}_t) + r\{\hat{g}(\bar{H}_t) - g(\bar{H}_t)\}\Big]\Big)\Big(\hat{g}(\bar{H}_t) - g(\bar{H}_t)\Big)\Big]\Big|_{r=0} \tag{105}$$

$$= -2\mathbb{E}\Big[\rho_t^{\bar{a}}(\bar{Z}_{t+\tau})\Big(\xi_t^{\bar{a}}(\bar{Z}_{t+\tau}) - g(\bar{H}_t)\Big)\Big(\hat{g}(\bar{H}_t) - g(\bar{H}_t)\Big)\Big] \tag{106}$$

$$= -2\mathbb{E}\Big[\rho_t^{\bar{a}}(\bar{Z}_{t+\tau})\Big(\mu_t^{\bar{a}}(\bar{H}_t) + \frac{\omega_t^{\bar{a}}(\bar{H}_t)}{\rho_t^{\bar{a}}(\bar{Z}_{t+\tau})}\Big[\gamma_t^{\bar{a}}(\bar{Z}_{t+\tau}) - \mu_t^{\bar{a}}(\bar{H}_t)\Big] - g(\bar{H}_t)\Big)\Big(\hat{g}(\bar{H}_t) - g(\bar{H}_t)\Big)\Big] \tag{107}$$

$$= -2\mathbb{E}\Big[\Big\{\rho_t^{\bar{a}}(\bar{Z}_{t+\tau})\Big(\mu_t^{\bar{a}}(\bar{H}_t) - g(\bar{H}_t)\Big) + \omega_t^{\bar{a}}(\bar{H}_t)\Big(\gamma_t^{\bar{a}}(\bar{Z}_{t+\tau}) - \mu_t^{\bar{a}}(\bar{H}_t)\Big)\Big\}\Big(\hat{g}(\bar{H}_t) - g(\bar{H}_t)\Big)\Big]. \tag{108}$$

Next, we compute the pathwise derivative of $D_g\mathcal{L}(g;\eta^{\bar{a}})[\hat{g} - g]$ with respect to all nuisance functions $\eta^{\bar{a}} = \{\pi_j^{\bar{a}}, \mu_j^{\bar{a}}, \omega_j^{\bar{a}}\}_{j=t}^{t+\tau}$. When calculating the pathwise derivative of the functions $f_t^{\bar{a}} \in \{\mu_t^{\bar{a}}, \gamma_t^{\bar{a}}, \rho_t^{\bar{a}}, \omega_t^{\bar{a}}\}$ with respect to $g_j^{\bar{a}} \in \eta^{\bar{a}}$, we use $f_t^{\bar{a}}(\cdot; g_j^{\bar{a}})$ to make our notation more explicit to highlight which $f_t^{\bar{a}}$ depends on the nuisance $g_j^{\bar{a}}$.

First, we calculate the pathwise derivative of $D_g\mathcal{L}(g;\eta^{\bar{a}})[\hat{g}-g]$ with respect to the nuisances $\pi_j^{\bar{a}}$ for $j=t,\dots,t+\tau$ via

$$D_{\pi_j^{\bar{a}}} D_g\mathcal{L}(g;\eta^{\bar{a}})[\hat{g}-g,\hat{\pi}_j^{\bar{a}}-\pi_j^{\bar{a}}] \tag{109}$$

$$=\frac{\mathrm{d}}{\mathrm{d}r} D_g\mathcal{L}\Big(g;\{\mu_j^{\bar{a}},\omega_j^{\bar{a}}\}_{j=t}^{t+\tau}\cup\{\pi_0^{\bar{a}},\dots,\pi_j^{\bar{a}}+r(\hat{\pi}_j^{\bar{a}}-\pi_j^{\bar{a}}),\dots,\pi_{t+\tau}^{\bar{a}}\}\Big)[\hat{g}-g]\,\Big|_{r=0} \tag{110}$$

$$\propto\frac{\mathrm{d}}{\mathrm{d}r}\mathbb{E}\bigg[\Big\{\rho_t^{\bar{a}}(\bar{Z}_{t+\tau};\pi_j^{\bar{a}}+r(\hat{\pi}_j^{\bar{a}}-\pi_j^{\bar{a}}))\Big(\mu_t^{\bar{a}}(\bar{H}_t)-g(\bar{H}_t)\Big) \tag{111}$$

$$+\omega_t^{\bar{a}}(\bar{H}_t;\pi_j^{\bar{a}}+r(\hat{\pi}_j^{\bar{a}}-\pi_j^{\bar{a}}))\Big(\gamma_t^{\bar{a}}(\bar{Z}_{t+\tau};\pi_j^{\bar{a}}+r(\hat{\pi}_j^{\bar{a}}-\pi_j^{\bar{a}}))-\mu_t^{\bar{a}}(\bar{H}_t)\Big)\Big\}\Big(\hat{g}(\bar{H}_t)-g(\bar{H}_t)\Big)\bigg]\bigg|_{r=0} \tag{112}$$

$$=\mathbb{E}\bigg[\Big\{\underbrace{\frac{\mathrm{d}}{\mathrm{d}r}\rho_t^{\bar{a}}(\bar{Z}_{t+\tau};\pi_j^{\bar{a}}+r(\hat{\pi}_j^{\bar{a}}-\pi_j^{\bar{a}}))\,\Big|_{r=0}}_{=0}\Big(\mu_t^{\bar{a}}(\bar{H}_t)-g(\bar{H}_t)\Big) \tag{113}$$

$$+\frac{\mathrm{d}}{\mathrm{d}r}\omega_t^{\bar{a}}(\bar{H}_t;\pi_j^{\bar{a}}+r(\hat{\pi}_j^{\bar{a}}-\pi_j^{\bar{a}}))\,\Big|_{r=0}\Big(\gamma_t^{\bar{a}}(\bar{Z}_{t+\tau})-\mu_t^{\bar{a}}(\bar{H}_t)\Big) \tag{114}$$

$$+\omega_t^{\bar{a}}(\bar{H}_t)\underbrace{\frac{\mathrm{d}}{\mathrm{d}r}\gamma_t^{\bar{a}}(\bar{Z}_{t+\tau};\pi_j^{\bar{a}}+r(\hat{\pi}_j^{\bar{a}}-\pi_j^{\bar{a}}))\,\Big|_{r=0}}_{=0}\Big\}\Big(\hat{g}(\bar{H}_t)-g(\bar{H}_t)\Big)\bigg] \tag{115}$$

$$=\mathbb{E}\bigg[\Big\{\frac{\mathrm{d}}{\mathrm{d}r}\omega_t^{\bar{a}}(\bar{H}_t;\pi_j^{\bar{a}}+r(\hat{\pi}_j^{\bar{a}}-\pi_j^{\bar{a}}))\,\Big|_{r=0}\Big(\gamma_t^{\bar{a}}(\bar{Z}_{t+\tau})-\mu_t^{\bar{a}}(\bar{H}_t)\Big)\Big\}\Big(\hat{g}(\bar{H}_t)-g(\bar{H}_t)\Big)\bigg] \tag{116}$$

$$=\mathbb{E}\bigg[\mathbb{E}\Big[\Big\{\frac{\mathrm{d}}{\mathrm{d}r}\omega_t^{\bar{a}}(\bar{H}_t;\pi_j^{\bar{a}}+r(\hat{\pi}_j^{\bar{a}}-\pi_j^{\bar{a}}))\,\Big|_{r=0}\Big(\gamma_t^{\bar{a}}(\bar{Z}_{t+\tau})-\mu_t^{\bar{a}}(\bar{H}_t)\Big)\Big\}\Big(\hat{g}(\bar{H}_t)-g(\bar{H}_t)\Big)\,\Big|\,\bar{H}_t\Big]\bigg] \tag{117}$$

$$=\mathbb{E}\bigg[\Big\{\frac{\mathrm{d}}{\mathrm{d}r}\omega_t^{\bar{a}}(\bar{H}_t;\pi_j^{\bar{a}}+r(\hat{\pi}_j^{\bar{a}}-\pi_j^{\bar{a}}))\,\Big|_{r=0}\Big(\underbrace{\mathbb{E}\Big[\gamma_t^{\bar{a}}(\bar{Z}_{t+\tau})\,\Big|\,\bar{H}_t\Big]}_{=\mu_t^{\bar{a}}(\bar{H}_t)}-\mu_t^{\bar{a}}(\bar{H}_t)\Big)\Big\}\Big(\hat{g}(\bar{H}_t)-g(\bar{H}_t)\Big)\bigg] \tag{118}$$

$$=0, \tag{119}$$

where the result we apply in Equation 113 follows from Lemma C.2, in Equation 115 from Lemma C.1, and in Equation 118 follows from Lemma C.3.

Next, we compute the pathwise derivative of $D_g \mathcal{L}(g; \eta^{\bar{a}})[\hat{g} - g]$ with respect to the nuisances $\mu_j^{\bar{a}}$ for $j = t, \ldots, t + \tau$ via

$$D_{\mu_j^{\bar{a}}} D_g \mathcal{L}(g; \eta^{\bar{a}})[\hat{g} - g, \hat{\mu}_j^{\bar{a}} - \mu_j^{\bar{a}}] \tag{120}$$

$$= \frac{\mathrm{d}}{\mathrm{d}r} D_g \mathcal{L}\Big(g; \{\pi_j^{\bar{a}}, \omega_j^{\bar{a}}\}_{j=t}^{t+\tau} \cup \{\mu_0^{\bar{a}}, \ldots, \mu_j^{\bar{a}} + r(\hat{\mu}_j^{\bar{a}} - \mu_j^{\bar{a}}), \ldots, \mu_{t+\tau}^{\bar{a}}\}\Big)[\hat{g} - g] \Big|_{r=0} \tag{121}$$

$$\propto \frac{\mathrm{d}}{\mathrm{d}r} \mathbb{E}\bigg[ \Big\{ \rho_t^{\bar{a}}(\bar{Z}_{t+\tau}) \Big( \mu_t^{\bar{a}}(\bar{H}_t; \mu_j^{\bar{a}} + r(\hat{\mu}_j^{\bar{a}} - \mu_j^{\bar{a}})) - g(\bar{H}_t) \Big) \tag{122}$$

$$+ \omega_t^{\bar{a}}(\bar{H}_t) \Big( \gamma_t^{\bar{a}}(\bar{Z}_{t+\tau}; \mu_j^{\bar{a}} + r(\hat{\mu}_j^{\bar{a}} - \mu_j^{\bar{a}})) - \mu_t^{\bar{a}}(\bar{H}_t; \mu_j^{\bar{a}} + r(\hat{\mu}_j^{\bar{a}} - \mu_j^{\bar{a}})) \Big) \Big\} \Big( \hat{g}(\bar{H}_t) - g(\bar{H}_t) \Big) \bigg] \bigg|_{r=0} \tag{123}$$

$$= \mathbb{E}\bigg[ \Big\{ \rho_t^{\bar{a}}(\bar{Z}_{t+\tau}) \frac{\mathrm{d}}{\mathrm{d}r} \mu_t^{\bar{a}}(\bar{H}_t; \mu_j^{\bar{a}} + r(\hat{\mu}_j^{\bar{a}} - \mu_j^{\bar{a}})) \Big|_{r=0} \tag{124}$$

$$+ \omega_t^{\bar{a}}(\bar{H}_t) \Big( \underbrace{\frac{\mathrm{d}}{\mathrm{d}r} \gamma_t^{\bar{a}}(\bar{Z}_{t+\tau}; \mu_j^{\bar{a}} + r(\hat{\mu}_j^{\bar{a}} - \mu_j^{\bar{a}})) \Big|_{r=0} - \frac{\mathrm{d}}{\mathrm{d}r} \mu_t^{\bar{a}}(\bar{H}_t; \mu_j^{\bar{a}} + r(\hat{\mu}_j^{\bar{a}} - \mu_j^{\bar{a}})) \Big|_{r=0}}_{=0} \Big) \Big\} \Big( \hat{g}(\bar{H}_t) - g(\bar{H}_t) \Big) \bigg] \tag{125}$$

$$= \mathbb{E}\bigg[ \Big\{ \rho_t^{\bar{a}}(\bar{Z}_{t+\tau}) \frac{\mathrm{d}}{\mathrm{d}r} \mu_t^{\bar{a}}(\bar{H}_t; \mu_j^{\bar{a}} + r(\hat{\mu}_j^{\bar{a}} - \mu_j^{\bar{a}})) \Big|_{r=0} \tag{126}$$

$$- \omega_t^{\bar{a}}(\bar{H}_t) \Big( \frac{\mathrm{d}}{\mathrm{d}r} \mu_t^{\bar{a}}(\bar{H}_t; \mu_j^{\bar{a}} + r(\hat{\mu}_j^{\bar{a}} - \mu_j^{\bar{a}})) \Big|_{r=0} \Big) \Big\} \Big( \hat{g}(\bar{H}_t) - g(\bar{H}_t) \Big) \bigg] \tag{127}$$

$$= \mathbb{E}\bigg[ \mathbb{E}\Big[ \Big\{ \rho_t^{\bar{a}}(\bar{Z}_{t+\tau}) \frac{\mathrm{d}}{\mathrm{d}r} \mu_t^{\bar{a}}(\bar{H}_t; \mu_j^{\bar{a}} + r(\hat{\mu}_j^{\bar{a}} - \mu_j^{\bar{a}})) \Big|_{r=0} \tag{128}$$

$$- \omega_t^{\bar{a}}(\bar{H}_t) \frac{\mathrm{d}}{\mathrm{d}r} \mu_t^{\bar{a}}(\bar{H}_t; \mu_j^{\bar{a}} + r(\hat{\mu}_j^{\bar{a}} - \mu_j^{\bar{a}})) \Big|_{r=0} \Big\} \Big( \hat{g}(\bar{H}_t) - g(\bar{H}_t) \Big) \Big| \bar{H}_t \Big] \bigg] \tag{129}$$

$$= \mathbb{E}\bigg[ \underbrace{\mathbb{E}\Big[ \Big\{ \rho_t^{\bar{a}}(\bar{Z}_{t+\tau}) \Big| \bar{H}_t \Big]}_{=\omega_t^{\bar{a}}(\bar{H}_t)} \frac{\mathrm{d}}{\mathrm{d}r} \mu_t^{\bar{a}}(\bar{H}_t; \mu_j^{\bar{a}} + r(\hat{\mu}_j^{\bar{a}} - \mu_j^{\bar{a}})) \Big|_{r=0} \tag{130}$$

$$- \omega_t^{\bar{a}}(\bar{H}_t) \frac{\mathrm{d}}{\mathrm{d}r} \mu_t^{\bar{a}}(\bar{H}_t; \mu_j^{\bar{a}} + r(\hat{\mu}_j^{\bar{a}} - \mu_j^{\bar{a}})) \Big|_{r=0} \Big\} \Big( \hat{g}(\bar{H}_t) - g(\bar{H}_t) \Big) \bigg] \tag{131}$$

$$= 0, \tag{132}$$

where the result we apply in Equation 125 follows from Lemma C.1, and in Equation 130 from Lemma C.4.

Finally, we compute the pathwise derivative of $D_g \mathcal{L}(g; \eta^{\bar{a}})[\hat{g} - g]$ with respect to the nuisances $\omega_j^{\bar{a}}$ for $j = t, \ldots, t + \tau$ via

$$D_{\omega_j^{\bar{a}}} D_g \mathcal{L}(g; \eta^{\bar{a}})[\hat{g} - g, \hat{\omega}_j^{\bar{a}} - \omega_j^{\bar{a}}] \tag{133}$$

$$= \frac{\mathrm{d}}{\mathrm{d}r} D_g \mathcal{L}\Big(g; \{\pi_j^{\bar{a}}, \mu_j^{\bar{a}}\}_{j=t}^{t+\tau} \cup \{\omega_0^{\bar{a}}, \ldots, \omega_j^{\bar{a}} + r(\hat{\omega}_j^{\bar{a}} - \omega_j^{\bar{a}}), \ldots, \omega_{t+\tau}^{\bar{a}}\}\Big)[\hat{g} - g] \Big|_{r=0} \tag{134}$$

$$\propto \frac{\mathrm{d}}{\mathrm{d}r} \mathbb{E}\Bigg[\Big\{\rho_t^{\bar{a}}(\bar{Z}_{t+\tau}; \omega_j^{\bar{a}} + r(\hat{\omega}_j^{\bar{a}} - \omega_j^{\bar{a}}))\Big(\mu_t^{\bar{a}}(\bar{H}_t) - g(\bar{H}_t)\Big) \tag{135}$$

$$+ \omega_t^{\bar{a}}(\bar{H}_t; \omega_j^{\bar{a}} + r(\hat{\omega}_j^{\bar{a}} - \omega_j^{\bar{a}}))\Big(\gamma_t^{\bar{a}}(\bar{Z}_{t+\tau}) - \mu_t^{\bar{a}}(\bar{H}_t)\Big)\Big\}\Big(\hat{g}(\bar{H}_t) - g(\bar{H}_t)\Big)\Bigg]\Bigg|_{r=0} \tag{136}$$

$$= \mathbb{E}\Bigg[\Big\{\underbrace{\frac{\mathrm{d}}{\mathrm{d}r} \rho_t^{\bar{a}}(\bar{Z}_{t+\tau}; \omega_j^{\bar{a}} + r(\hat{\omega}_j^{\bar{a}} - \omega_j^{\bar{a}}))\Big|_{r=0}}_{=0}\Big(\mu_t^{\bar{a}}(\bar{H}_t) - g(\bar{H}_t)\Big) \tag{137}$$

$$+ \frac{\mathrm{d}}{\mathrm{d}r} \omega_t^{\bar{a}}(\bar{H}_t; \omega_j^{\bar{a}} + r(\hat{\omega}_j^{\bar{a}} - \omega_j^{\bar{a}}))\Big|_{r=0}\Big(\gamma_t^{\bar{a}}(\bar{Z}_{t+\tau}) - \mu_t^{\bar{a}}(\bar{H}_t)\Big)\Big\}\Big(\hat{g}(\bar{H}_t) - g(\bar{H}_t)\Big)\Bigg] \tag{138}$$

$$= \mathbb{E}\Bigg[\frac{\mathrm{d}}{\mathrm{d}r} \omega_t^{\bar{a}}(\bar{H}_t; \omega_j^{\bar{a}} + r(\hat{\omega}_j^{\bar{a}} - \omega_j^{\bar{a}}))\Big|_{r=0}\Big(\gamma_t^{\bar{a}}(\bar{Z}_{t+\tau}) - \mu_t^{\bar{a}}(\bar{H}_t)\Big)\Big(\hat{g}(\bar{H}_t) - g(\bar{H}_t)\Big)\Bigg] \tag{139}$$

$$= \mathbb{E}\Bigg[\mathbb{E}\Big[\frac{\mathrm{d}}{\mathrm{d}r} \omega_t^{\bar{a}}(\bar{H}_t; \omega_j^{\bar{a}} + r(\hat{\omega}_j^{\bar{a}} - \omega_j^{\bar{a}}))\Big|_{r=0}\Big(\gamma_t^{\bar{a}}(\bar{Z}_{t+\tau}) - \mu_t^{\bar{a}}(\bar{H}_t)\Big)\Big(\hat{g}(\bar{H}_t) - g(\bar{H}_t)\Big)\Big|\bar{H}_t\Big]\Bigg] \tag{140}$$

$$= \mathbb{E}\Bigg[\frac{\mathrm{d}}{\mathrm{d}r} \omega_t^{\bar{a}}(\bar{H}_t; \omega_j^{\bar{a}} + r(\hat{\omega}_j^{\bar{a}} - \omega_j^{\bar{a}}))\Big|_{r=0}\Big(\underbrace{\mathbb{E}\Big[\gamma_t^{\bar{a}}(\bar{Z}_{t+\tau})\Big|\bar{H}_t\Big]}_{=\mu_t^{\bar{a}}(\bar{H}_t)} - \mu_t^{\bar{a}}(\bar{H}_t)\Big)\Big(\hat{g}(\bar{H}_t) - g(\bar{H}_t)\Big)\Bigg] \tag{141}$$

$$= 0, \tag{142}$$

where the result we apply in Equation 137 follows from Lemma C.2, and in Equation 141 from Lemma C.3. $\qquad \square$

## C.2 Conditional average treatment effects (CATEs)

We split the following section into two parts: first, as for the CAPOs, we derive several supporting lemmas to prove our main results ($\rightarrow$Lemmas C.7 to C.10). Then, we derive the theorems from the main paper ($\rightarrow$Theorem C.11 and Theorem C.12).

### C.2.1 Lemmas (CATEs)

In order to prove our main theorems for CATEs, we first introduce a **series of helping lemmas**.

**Lemma C.7.** *Let*

$$\gamma_t^{\bar{a},\bar{b}}(\bar{Z}_{t+\tau}) = \gamma_t^{\bar{a}}(\bar{Z}_{t+\tau}) - \gamma_t^{\bar{b}}(\bar{Z}_{t+\tau}). \tag{143}$$

*Then, $\gamma_t^{\bar{a},\bar{b}}(\bar{Z}_{t+\tau})$ is Neyman-orthogonal with respect to all nuisance functions $\eta^{\bar{a},\bar{b}} = \eta^{\bar{a}} \cup \eta^{\bar{b}}$.*

*Proof.* The proof immediately follows from linearity of the efficient influence function and Lemma C.1. $\qquad\square$

**Lemma C.8.** *Let*

$$\rho_t^{\bar{a},\bar{b}}(\bar{Z}_{t+\tau}) = \rho_t^{\bar{a}}(\bar{Z}_{t+\tau})\omega_t^{\bar{b}}(\bar{H}_t) + \rho_t^{\bar{b}}(\bar{Z}_{t+\tau})\omega_t^{\bar{a}}(\bar{H}_t) - \omega_t^{\bar{a},\bar{b}}(\bar{H}_t). \tag{144}$$

*Then, $\rho_t^{\bar{a},\bar{b}}(\bar{Z}_{t+\tau})$ is Neyman-orthogonal with respect to all nuisance functions $\eta^{\bar{a},\bar{b}} = \eta^{\bar{a}} \cup \eta^{\bar{b}}$.*

*Proof.* As in Lemma C.2, we notice that $\rho_t^{\bar{a},\bar{b}}(\bar{Z}_{t+\tau})$ is trivially Neyman-orthogonal with respect to $\mu_j^{\bar{a}}$ and $\mu_j^{\bar{b}}$ as it does not dependent on it. Further, we show that $\rho_t^{\bar{a},\bar{b}}(\bar{Z}_{t+\tau})$ is the uncentered efficient influence function of $\mathbb{E}\left[\omega_t^{\bar{a},\bar{b}}(\bar{H}_t)\right]$, and hence, Neyman-orthogonal with respect to the nuisance functions $\{\pi_j^{\bar{a}}, \omega_j^{\bar{a}}, \pi_j^{\bar{b}}, \omega_j^{\bar{b}}\}_{j=t}^{t+\tau}$. For this, we make once again use of the chain rule for pathwise derivatives (Kennedy, 2022; Luedtke, 2024). We start with the efficient influence function of $\omega_t^{\bar{a}}(\bar{h}_t)$, which is given by

$$\mathbb{IF}\left(\omega_t^{\bar{a},\bar{b}}(\bar{h}_t)\right) \tag{145}$$

$$\mathbb{IF}\left(\omega_t^{\bar{a}}(\bar{h}_t)\omega_t^{\bar{b}}(\bar{h}_t)\right) \tag{146}$$

$$= \underbrace{\mathbb{IF}\left(\mathbb{E}\Big[\prod_{j=t}^{t+\tau}\pi_j^{\bar{a}}(\bar{H}_j) \mid \bar{H}_t = \bar{h}_t\Big]\right)\omega_t^{\bar{b}}(\bar{h}_t)}_{(*)} + \underbrace{\omega_t^{\bar{b}}(\bar{h}_t)\,\mathbb{IF}\left(\mathbb{E}\Big[\prod_{j=t}^{t+\tau}\pi_j^{\bar{b}}(\bar{H}_j) \mid \bar{H}_t = \bar{h}_t\Big]\right)}_{(**)} \tag{147}$$

For both $(*)$ and $(**)$, we can follow the derivations in Lemma C.2, which yields

$$\mathbb{IF}\left(\omega_t^{\bar{a},\bar{b}}(\bar{h}_t)\right) \tag{148}$$

$$= \frac{\mathbb{1}_{\{\bar{h}_t=\bar{H}_t\}}}{p(\bar{h}_t)}\Bigg[\Big\{ -\omega_t^{\bar{a}}(\bar{h}_t) + \prod_{j=t}^{t+\tau}\pi_j^{\bar{a}}(H_{t+1:j},\bar{h}_t) + \sum_{j=t}^{t+\tau}\Big(\mathbb{1}_{\{a_j=A_j\}} - \pi_j^{\bar{a}}(H_{t+1:j},\bar{h}_t)\Big) \tag{149}$$

$$\times \prod_{t\leq k<j}\pi_k^{\bar{a}}(H_{t+1:k},\bar{h}_t)\mathbb{E}\Big[\prod_{k>j}\pi_k^{\bar{a}}(\bar{H}_k) \mid H_{t+1:j}, \bar{H}_t = \bar{h}_t\Big]\Big\}\omega_t^{\bar{b}}(\bar{h}_t) \tag{150}$$

$$+ \omega_t^{\bar{a}}(\bar{h}_t)\Big\{ -\omega_t^{\bar{b}}(\bar{h}_t) + \prod_{j=t}^{t+\tau}\pi_j^{\bar{b}}(H_{t+1:j},\bar{h}_t) + \sum_{j=t}^{t+\tau}\Big(\mathbb{1}_{\{a_j=A_j\}} - \pi_j^{\bar{b}}(H_{t+1:j},\bar{h}_t)\Big) \tag{151}$$

$$\times \prod_{t\leq k<j}\pi_k^{\bar{b}}(H_{t+1:k},\bar{h}_t)\mathbb{E}\Big[\prod_{k>j}\pi_k^{\bar{b}}(\bar{H}_k) \mid H_{t+1:j}, \bar{H}_t = \bar{h}_t\Big]\Big\}\Bigg]. \tag{152}$$

Finally, we derive the efficient influence function of $\mathbb{E}\left[\omega_t^{\bar{a},\bar{b}}(\bar{H}_t)\right]$ via

$$\mathbb{IF}\left(\mathbb{E}\left[\omega_t^{\bar{a},\bar{b}}(\bar{H}_t)\right]\right) \tag{153}$$

$$= \sum_{\bar{h}_t} \mathbb{IF}\left(p(\bar{h}_t)\omega_t^{\bar{a},\bar{b}}(\bar{h}_t)\right) \tag{154}$$

$$= \sum_{\bar{h}_t} \left[\mathbb{IF}\left(p(\bar{h}_t)\right)\omega_t^{\bar{a},\bar{b}}(\bar{h}_t) + p(\bar{h}_t)\mathbb{IF}\left(\omega_t^{\bar{a},\bar{b}}(\bar{h}_t)\right)\right] \tag{155}$$

$$= \omega_t^{\bar{a},\bar{b}}(\bar{H}_t) - \mathbb{E}\left[\omega_t^{\bar{a},\bar{b}}(\bar{H}_t)\right] \tag{156}$$

$$+ \left\{ -\omega_t^{\bar{a}}(\bar{H}_t) + \prod_{j=t}^{t+\tau} \pi_j^{\bar{a}}(\bar{H}_j) + \sum_{j=t}^{t+\tau} \left(\mathbb{1}_{\{a_j=A_j\}} - \pi_j^{\bar{a}}(\bar{H}_j)\right) \prod_{t \leq k < j} \pi_k^{\bar{a}}(\bar{H}_k) \underbrace{\mathbb{E}\left[\prod_{k>j} \pi_k^{\bar{a}}(\bar{H}_k) \mid \bar{H}_j\right]}_{=\omega_{j+1}^{\bar{a}}(\bar{H}_j)} \right\} \omega_t^{\bar{b}}(\bar{H}_t) \tag{157}$$

$$+ \omega_t^{\bar{a}}(\bar{H}_t)\left\{ -\omega_t^{\bar{b}}(\bar{H}_t) + \prod_{j=t}^{t+\tau} \pi_j^{\bar{b}}(\bar{H}_j) + \sum_{j=t}^{t+\tau} \left(\mathbb{1}_{\{a_j=A_j\}} - \pi_j^{\bar{b}}(\bar{H}_j)\right) \prod_{t \leq k < j} \pi_k^{\bar{b}}(\bar{H}_k) \underbrace{\mathbb{E}\left[\prod_{k>j} \pi_k^{\bar{b}}(\bar{H}_k) \mid \bar{H}_j\right]}_{=\omega_{j+1}^{\bar{b}}(\bar{H}_j)} \right\} \tag{158}$$

$$= \omega_t^{\bar{a},\bar{b}}(\bar{H}_t) - \mathbb{E}\left[\omega_t^{\bar{a},\bar{b}}(\bar{H}_t)\right] + \left\{ -\omega_t^{\bar{b}}(\bar{H}_t) + \rho_t^{\bar{b}}(\bar{Z}_{t+\tau})\right\}\omega_t^{\bar{b}}(\bar{H}_t) + \omega_t^{\bar{a}}(\bar{H}_t)\left\{ -\omega_t^{\bar{b}}(\bar{H}_t) + \rho_t^{\bar{b}}(\bar{Z}_{t+\tau})\right\} \tag{159}$$

$$= \omega_t^{\bar{a},\bar{b}}(\bar{H}_t) - \mathbb{E}\left[\omega_t^{\bar{a},\bar{b}}(\bar{H}_t)\right] - 2\omega_t^{\bar{a},\bar{b}}(\bar{H}_t) + \omega_t^{\bar{b}}(\bar{H}_t)\rho_t^{\bar{a}}(\bar{Z}_{t+\tau}) + \omega_t^{\bar{a}}(\bar{H}_t)\rho_t^{\bar{b}}(\bar{Z}_{t+\tau}) \tag{160}$$

$$= -\mathbb{E}\left[\omega_t^{\bar{a},\bar{b}}(\bar{H}_t)\right] + \rho_t^{\bar{a},\bar{b}}(\bar{Z}_{t+\tau}). \tag{161}$$

$$\square$$

**Lemma C.9.** *Let*

$$\gamma_t^{\bar{a},\bar{b}}(\bar{Z}_{t+\tau}) = \gamma_t^{\bar{a}}(\bar{Z}_{t+\tau}) - \gamma_t^{\bar{b}}(\bar{Z}_{t+\tau}), \tag{162}$$

*for two treatment sequences $a_{t:t+\tau}$, $b_{t:t+\tau}$, and let*

$$\mu_t^{\bar{a},\bar{b}}(\bar{h}_t) = \mathbb{E}\left[Y_{t:t+\tau}[a_{t:t+\tau}] - Y_{t:t+\tau}[b_{t:t+\tau}] \,\Big|\, \bar{H}_t = \bar{h}_t\right] \tag{163}$$

*be the conditional average treatment effect. Then, it holds that*

$$\mathbb{E}\left[\gamma_t^{\bar{a},\bar{b}}(\bar{Z}_{t+\tau}) \,\Big|\, \bar{H}_t\right] = \mu_t^{\bar{a},\bar{b}}(\bar{H}_t). \tag{164}$$

*Proof.* The proof immediately follows from linearity of expectations and Lemma C.3. $\square$

**Lemma C.10.** *Let*

$$\rho_t^{\bar{a},\bar{b}}(\bar{Z}_{t+\tau}) = \rho_t^{\bar{a}}(\bar{Z}_{t+\tau})\omega_t^{\bar{b}}(\bar{H}_t) + \rho_t^{\bar{b}}(\bar{Z}_{t+\tau})\omega_t^{\bar{a}}(\bar{H}_t) - \omega_t^{\bar{a},\bar{b}}(\bar{H}_t) \tag{165}$$

*for two for two treatment sequences $a_{t:t+\tau}$, $b_{t:t+\tau}$, and let*

$$\omega_j^{\bar{a},\bar{b}}(\bar{h}_\ell) = \omega_j^{\bar{a}}(\bar{h}_\ell) + \omega_j^{\bar{b}}(\bar{h}_\ell) \tag{166}$$

*be the weight function. Then, it holds that*

$$\mathbb{E}\left[\rho_t^{\bar{a},\bar{b}}(\bar{Z}_{t+\tau}) \,\Big|\, \bar{H}_t\right] = \omega_t^{\bar{a},\bar{b}}(\bar{H}_t). \tag{167}$$

*Proof.* The proof follows from Lemma C.4 via

$$\mathbb{E}\left[\rho_t^{\bar{a},\bar{b}}(\bar{Z}_{t+\tau}) \,\Big|\, \bar{H}_t\right] \tag{168}$$

$$= \mathbb{E}\left[\rho_t^{\bar{a}}(\bar{Z}_{t+\tau})\omega_t^{\bar{b}}(\bar{H}_t) + \rho_t^{\bar{b}}(\bar{Z}_{t+\tau})\omega_t^{\bar{a}}(\bar{H}_t) - \omega_t^{\bar{a}\bar{b}}(\bar{H}_t)\right] \tag{169}$$

$$= \mathbb{E}\left[\rho_t^{\bar{a}}(\bar{Z}_{t+\tau}) \,\Big|\, \bar{H}_t\right]\omega_t^{\bar{b}}(\bar{H}_t) + \mathbb{E}\left[\rho_t^{\bar{b}}(\bar{Z}_{t+\tau}) \,\Big|\, \bar{H}_t\right]\omega_t^{\bar{a}}(\bar{H}_t) - \omega_t^{\bar{a}\bar{b}}(\bar{H}_t) \tag{170}$$

$$= \omega_t^{\bar{a}}(\bar{H}_t)\omega_t^{\bar{b}}(\bar{H}_t) + \omega_t^{\bar{a}}(\bar{H}_t)\omega_t^{\bar{b}}(\bar{H}_t) - \omega_t^{\bar{a},\bar{b}}(\bar{H}_t) \tag{171}$$

$$= \omega_t^{\bar{a},\bar{b}}(\bar{H}_t). \tag{172}$$

$\square$

### C.2.2 THEOREMS (CATEs)

Finally, we can prove the CATE version of our theorems from the main paper. For both proofs, we leverage additional helping lemmas that we derived in Supplement C.2.1.

**Theorem C.11** (Weighted population risk (CATE)). *Let*

$$\xi_t^{\bar{a},\bar{b}}(\bar{Z}_{t+\tau}) = \mu_t^{\bar{a},\bar{b}}(\bar{H}_t) + \frac{\omega_t^{\bar{a},\bar{b}}(\bar{H}_t)}{\rho_t^{\bar{a},\bar{b}}(\bar{Z}_{t+\tau})}\Big(\gamma_t^{\bar{a},\bar{b}}(\bar{Z}_{t+\tau}) - \mu_t^{\bar{a},\bar{b}}(\bar{H}_t)\Big), \tag{173}$$

*where*

$$\rho_t^{\bar{a},\bar{b}}(\bar{Z}_{t+\tau}) = \rho_t^{\bar{a}}(\bar{Z}_{t+\tau}) + \rho_t^{\bar{b}}(\bar{Z}_{t+\tau}) \tag{174}$$

*and*

$$\gamma_t^{\bar{a},\bar{b}}(\bar{Z}_{t+\tau}) = \gamma_t^{\bar{a}}(\bar{Z}_{t+\tau}) - \gamma_t^{\bar{b}}(\bar{Z}_{t+\tau}) \tag{175}$$

*with $\bar{Z}_{t+\tau} = (\bar{H}_{t+\tau}, A_{t+\tau}, Y_{t+\tau})$. Then, the population risk function*

$$\mathcal{L}(g; \eta^{\bar{a},\bar{b}}) = \frac{1}{\mathbb{E}\Big[\omega_t^{\bar{a},\bar{b}}(\bar{H}_t)\Big]}\mathbb{E}\bigg[\rho_t^{\bar{a},\bar{b}}(\bar{Z}_{t+\tau})\Big(\xi_t^{\bar{a},\bar{b}}(\bar{Z}_{t+\tau}) - g(\bar{H}_t)\Big)^2\bigg] \tag{176}$$

*minimizes the oracle risk*

$$\mathcal{L}^*(g; \eta^{\bar{a},\bar{b}}) = \frac{1}{\mathbb{E}\Big[\omega_t^{\bar{a},\bar{b}}(\bar{H}_t)\Big]}\mathbb{E}\bigg[\omega_t^{\bar{a},\bar{b}}(\bar{H}_t)\Big(\mu_t^{\bar{a},\bar{b}}(\bar{H}_t) - g(\bar{H}_t)\Big)^2\bigg]. \tag{177}$$

*Proof.* The proof follows the exact same steps as for Theorem C.5, where we can replace Lemma C.3 with Lemma C.9, and Lemma C.4 with Lemma C.10.

For completeness, we repeat the derivations in the following:

As in Theorem C.5, we need to show that

$$\mathbb{E}\bigg[\rho_t^{\bar{a},\bar{b}}(\bar{Z}_{t+\tau})\Big(\xi_t^{\bar{a},\bar{b}}(\bar{Z}_{t+\tau}) - g(\bar{H}_t)\Big)^2\bigg] = \mathbb{E}\bigg[\omega_t^{\bar{a}}(\bar{H}_t)\Big(\mu_t^{\bar{a},\bar{b}}(\bar{H}_t) - g(\bar{H}_t)\Big)^2\bigg] + C, \tag{178}$$

where $C$ is some constant term that does **not** depend on $g$. For this, notice that

$$\mathbb{E}\bigg[\rho_t^{\bar{a},\bar{b}}(\bar{Z}_{t+\tau})\Big(\xi_t^{\bar{a},\bar{b}}(\bar{Z}_{t+\tau}) - g(\bar{H}_t)\Big)^2\bigg] \tag{179}$$

$$=\mathbb{E}\bigg[\rho_t^{\bar{a},\bar{b}}(\bar{Z}_{t+\tau})\Big(\xi_t^{\bar{a},\bar{b}}(\bar{Z}_{t+\tau}) - \mu_t^{\bar{a},\bar{b}}(\bar{H}_t) + \mu_t^{\bar{a}}(\bar{H}_t) - g(\bar{H}_t)\Big)^2\bigg] \tag{180}$$

$$=\underbrace{\mathbb{E}\bigg[\rho_t^{\bar{a},\bar{b}}(\bar{Z}_{t+\tau})\Big(\xi_t^{\bar{a},\bar{b}}(\bar{Z}_{t+\tau}) - \mu_t^{\bar{a},\bar{b}}(\bar{H}_t)\Big)^2\bigg] + 2\mathbb{E}\bigg[\rho_t^{\bar{a},\bar{b}}(\bar{Z}_{t+\tau})\Big(\xi_t^{\bar{a},\bar{b}}(\bar{Z}_{t+\tau}) - \mu_t^{\bar{a},\bar{b}}(\bar{H}_t)\Big)\mu_t^{\bar{a},\bar{b}}(\bar{H}_t)\bigg]}_{=C}$$

$$\tag{181}$$

$$- 2\mathbb{E}\bigg[\rho_t^{\bar{a},\bar{b}}(\bar{Z}_{t+\tau})\Big(\xi_t^{\bar{a},\bar{b}}(\bar{Z}_{t+\tau}) - \mu_t^{\bar{a},\bar{b}}(\bar{H}_t)\Big)g(\bar{H}_t)\bigg] + \mathbb{E}\bigg[\rho_t^{\bar{a},\bar{b}}(\bar{Z}_{t+\tau})\Big(\mu_t^{\bar{a},\bar{b}}(\bar{H}_t) - g(\bar{H}_t)\Big)^2\bigg]. \tag{182}$$

Again, the first two terms do not depend on $g$ and are therefore constant. Hence, we focus on

$$\mathbb{E}\left[\rho_t^{\bar{a},\bar{b}}(\bar{Z}_{t+\tau})\Big(\xi_t^{\bar{a},\bar{b}}(\bar{Z}_{t+\tau}) - \mu_t^{\bar{a},\bar{b}}(\bar{H}_t)\Big)g(\bar{H}_t)\right] \tag{183}$$

$$=\mathbb{E}\left[\mathbb{E}\left[\rho_t^{\bar{a},\bar{b}}(\bar{Z}_{t+\tau})\Big(\xi_t^{\bar{a},\bar{b}}(\bar{Z}_{t+\tau}) - \mu_t^{\bar{a},\bar{b}}(\bar{H}_t)\Big)g(\bar{H}_t) \,\Big|\, \bar{H}_t\right]\right] \tag{184}$$

$$=\mathbb{E}\left[\mathbb{E}\left[\rho_t^{\bar{a},\bar{b}}(\bar{Z}_{t+\tau})\Big(\mu_t^{\bar{a},\bar{b}}(\bar{H}_t) + \frac{\omega_t^{\bar{a},\bar{b}}(\bar{H}_t)}{\rho_t^{\bar{a},\bar{b}}(\bar{Z}_{t+\tau})}\Big(\gamma_t^{\bar{a},\bar{b}}(\bar{Z}_{t+\tau}) - \mu_t^{\bar{a},\bar{b}}(\bar{H}_t)\Big) - \mu_t^{\bar{a},\bar{b}}(\bar{H}_t)\Big)g(\bar{H}_t) \,\Big|\, \bar{H}_t\right]\right] \tag{185}$$

$$=\mathbb{E}\left[\mathbb{E}\left[\omega_t^{\bar{a},\bar{b}}(\bar{H}_t)\Big(\gamma_t^{\bar{a},\bar{b}}(\bar{Z}_{t+\tau}) - \mu_t^{\bar{a},\bar{b}}(\bar{H}_t)\Big)g(\bar{H}_t) \,\Big|\, \bar{H}_t\right]\right] \tag{186}$$

$$=\mathbb{E}\left[\omega_t^{\bar{a},\bar{b}}(\bar{H}_t)\Big(\underbrace{\mathbb{E}\left[\gamma_t^{\bar{a},\bar{b}}(\bar{Z}_{t+\tau}) \,\Big|\, \bar{H}_t\right]}_{=\mu_t^{\bar{a},\bar{b}}(\bar{H}_t)} - \mu_t^{\bar{a},\bar{b}}(\bar{H}_t)\Big)g(\bar{H}_t)\right] \tag{187}$$

$$=0, \tag{188}$$

where the result in Equation 187 follows from Lemma C.9.

Finally, we simplify Equation 182 via

$$\mathbb{E}\left[\rho_t^{\bar{a},\bar{b}}(\bar{Z}_{t+\tau})\Big(\mu_t^{\bar{a},\bar{b}}(\bar{H}_t) - g(\bar{H}_t)\Big)^2\right] \tag{189}$$

$$=\mathbb{E}\left[\mathbb{E}\left[\rho_t^{\bar{a},\bar{b}}(\bar{Z}_{t+\tau})\Big(\mu_t^{\bar{a},\bar{b}}(\bar{H}_t) - g(\bar{H}_t)\Big)^2 \,\Big|\, \bar{H}_t\right]\right] \tag{190}$$

$$=\mathbb{E}\left[\underbrace{\mathbb{E}\left[\rho_t^{\bar{a},\bar{b}}(\bar{Z}_{t+\tau}) \,\Big|\, \bar{H}_t\right]}_{=\omega_t^{\bar{a},\bar{b}}(\bar{H}_t)}\Big(\mu_t^{\bar{a},\bar{b}}(\bar{H}_t) - g(\bar{H}_t)\Big)^2\right] \tag{191}$$

$$=\mathbb{E}[\omega_t^{\bar{a},\bar{b}}(\bar{H}_t)]\mathcal{L}^*(g;\eta^{\bar{a},\bar{b}}), \tag{192}$$

where the result Equation 191 follows from Lemma C.10.

Hence, combining Equation 182 with Equation 188 and Equation 192, and multiplying with $1/\mathbb{E}[\omega_t^{\bar{a},\bar{b}}(\bar{H}_t)]$ yields

$$\mathcal{L}(g;\eta^{\bar{a},\bar{b}}) = \mathcal{L}^*(g;\eta^{\bar{a},\bar{b}}) + C. \tag{193}$$

$\square$

**Theorem C.12** (Neyman-orthogonality (CATE)). *The weighted population risk*

$$\mathcal{L}(g; \eta^{\bar{a},\bar{b}}) = \frac{1}{\mathbb{E}\left[\omega_t^{\bar{a},\bar{b}}(\bar{H}_t)\right]} \mathbb{E}\left[\rho_t^{\bar{a},\bar{b}}(\bar{Z}_{t+\tau})\Big(\xi_t^{\bar{a},\bar{b}}(\bar{Z}_{t+\tau}) - g(\bar{H}_t)\Big)^2\right] \tag{194}$$

*is Neyman-orthogonal with respect to all nuisance functions $\eta^{\bar{a},\bar{b}} = \eta^{\bar{a}} \cup \eta^{\bar{b}}$.*

*Proof.* The proof follows the proof for Theorem C.6, where we can replace Lemma C.1 with Lemma C.7, and Lemma C.2 with Lemma C.8.

Again, for completeness, we provide the steps below.

In order to show Neyman-orthogonality, we first calculate the pathwise-derivative with respect to the target parameter $g$ via

$$D_g \mathcal{L}(g; \eta^{\bar{a},\bar{b}})[\hat{g} - g] \tag{195}$$

$$\propto \frac{\mathrm{d}}{\mathrm{d}r} \mathbb{E}\left[\rho_t^{\bar{a},\bar{b}}(\bar{Z}_{t+\tau})\Big(\xi_t^{\bar{a},\bar{b}}(\bar{Z}_{t+\tau}) - \Big[g(\bar{H}_t) + r\{\hat{g}(\bar{H}_t) - g(\bar{H}_t)\}\Big]\Big)^2\right]\Bigg|_{r=0} \tag{196}$$

$$= -2\mathbb{E}\left[\rho_t^{\bar{a},\bar{b}}(\bar{Z}_{t+\tau})\Big(\xi_t^{\bar{a},\bar{b}}(\bar{Z}_{t+\tau}) - \Big[g(\bar{H}_t) + r\{\hat{g}(\bar{H}_t) - g(\bar{H}_t)\}\Big]\Big)\Big(\hat{g}(\bar{H}_t) - g(\bar{H}_t)\Big)\right]\Bigg|_{r=0} \tag{197}$$

$$= -2\mathbb{E}\left[\rho_t^{\bar{a},\bar{b}}(\bar{Z}_{t+\tau})\Big(\xi_t^{\bar{a},\bar{b}}(\bar{Z}_{t+\tau}) - g(\bar{H}_t)\Big)\Big(\hat{g}(\bar{H}_t) - g(\bar{H}_t)\Big)\right] \tag{198}$$

$$= -2\mathbb{E}\left[\rho_t^{\bar{a},\bar{b}}(\bar{Z}_{t+\tau})\Big(\mu_t^{\bar{a},\bar{b}}(\bar{H}_t) + \frac{\omega_t^{\bar{a},\bar{b}}(\bar{H}_t)}{\rho_t^{\bar{a},\bar{b}}(\bar{Z}_{t+\tau})}\Big[\gamma_t^{\bar{a},\bar{b}}(\bar{Z}_{t+\tau}) - \mu_t^{\bar{a},\bar{b}}(\bar{H}_t)\Big] - g(\bar{H}_t)\Big)\Big(\hat{g}(\bar{H}_t) - g(\bar{H}_t)\Big)\right] \tag{199}$$

$$= -2\mathbb{E}\left[\left\{\rho_t^{\bar{a},\bar{b}}(\bar{Z}_{t+\tau})\Big(\mu_t^{\bar{a},\bar{b}}(\bar{H}_t) - g(\bar{H}_t)\Big) + \omega_t^{\bar{a},\bar{b}}(\bar{H}_t)\Big(\gamma_t^{\bar{a},\bar{b}}(\bar{Z}_{t+\tau}) - \mu_t^{\bar{a},\bar{b}}(\bar{H}_t)\Big)\right\}\Big(\hat{g}(\bar{H}_t) - g(\bar{H}_t)\Big)\right]. \tag{200}$$

Without loss of generality, we compute the pathwise derivative of $D_g \mathcal{L}(g; \eta^{\bar{a},\bar{b}})[\hat{g} - g]$ with respect to the nuisance functions $\eta^{\bar{a}}$. The case for $\eta^{\bar{b}}$ follows completely analogously.

Again, for the pathwise derivative of the functions $f_t^{\bar{a},\bar{b}} \in \{\mu_t^{\bar{a},\bar{b}}, \gamma_t^{\bar{a},\bar{b}}, \rho_t^{\bar{a},\bar{b}}, \omega_t^{\bar{a},\bar{b}}\}$ with respect to $g_j^{\bar{a}} \in \eta^{\bar{a}}$, we use $f_t^{\bar{a},\bar{b}}(\cdot; g_j^{\bar{a}})$ to make our notation more explicit to highlight which $f_t^{\bar{a},\bar{b}}$ depends on the nuisance $g_j^{\bar{a}}$.

The pathwise derivative of $D_g\mathcal{L}(g;\eta^{\bar{a},\bar{b}})[\hat{g}-g]$ with respect to the nuisances $\pi_j^{\bar{a}}$ for $j = t,\ldots,t+\tau$ is given by

$$D_{\pi_j^{\bar{a}}}D_g\mathcal{L}(g;\eta^{\bar{a},\bar{b}})[\hat{g}-g,\hat{\pi}_j^{\bar{a}}-\pi_j^{\bar{a}}] \tag{201}$$

$$=\frac{\mathrm{d}}{\mathrm{d}r}D_g\mathcal{L}\Big(g;\{\mu_j^{\bar{a}},\mu_j^{\bar{b}},\omega_j^{\bar{a}},\omega_j^{\bar{b}},\pi_j^{\bar{b}}\}_{j=t}^{t+\tau}\cup\{\pi_0^{\bar{a}},\ldots,\pi_j^{\bar{a}}+r(\hat{\pi}_j^{\bar{a}}-\pi_j^{\bar{a}}),\ldots,\pi_{t+\tau}^{\bar{a}}\}\Big)[\hat{g}-g]\,\Big|_{r=0} \tag{202}$$

$$\propto\frac{\mathrm{d}}{\mathrm{d}r}\mathbb{E}\Bigg[\Big\{\rho_t^{\bar{a},\bar{b}}(\bar{Z}_{t+\tau};\pi_j^{\bar{a}}+r(\hat{\pi}_j^{\bar{a}}-\pi_j^{\bar{a}}))\Big(\mu_t^{\bar{a},\bar{b}}(\bar{H}_t)-g(\bar{H}_t)\Big) \tag{203}$$

$$+\omega_t^{\bar{a},\bar{b}}(\bar{H}_t;\pi_j^{\bar{a}}+r(\hat{\pi}_j^{\bar{a}}-\pi_j^{\bar{a}}))\Big(\gamma_t^{\bar{a},\bar{b}}(\bar{Z}_{t+\tau};\pi_j^{\bar{a}}+r(\hat{\pi}_j^{\bar{a}}-\pi_j^{\bar{a}}))-\mu_t^{\bar{a},\bar{b}}(\bar{H}_t)\Big)\Big\}\Big(\hat{g}(\bar{H}_t)-g(\bar{H}_t)\Big)\Bigg]\Bigg|_{r=0} \tag{204}$$

$$=\mathbb{E}\Bigg[\Big\{\underbrace{\frac{\mathrm{d}}{\mathrm{d}r}\rho_t^{\bar{a},\bar{b}}(\bar{Z}_{t+\tau};\pi_j^{\bar{a}}+r(\hat{\pi}_j^{\bar{a}}-\pi_j^{\bar{a}}))\,\Big|_{r=0}}_{=0}\Big(\mu_t^{\bar{a},\bar{b}}(\bar{H}_t)-g(\bar{H}_t)\Big) \tag{205}$$

$$+\frac{\mathrm{d}}{\mathrm{d}r}\omega_t^{\bar{a},\bar{b}}(\bar{H}_t;\pi_j^{\bar{a}}+r(\hat{\pi}_j^{\bar{a}}-\pi_j^{\bar{a}}))\,\Big|_{r=0}\Big(\gamma_t^{\bar{a},\bar{b}}(\bar{Z}_{t+\tau})-\mu_t^{\bar{a},\bar{b}}(\bar{H}_t)\Big) \tag{206}$$

$$+\omega_t^{\bar{a},\bar{b}}(\bar{H}_t)\underbrace{\frac{\mathrm{d}}{\mathrm{d}r}\gamma_t^{\bar{a},\bar{b}}(\bar{Z}_{t+\tau};\pi_j^{\bar{a}}+r(\hat{\pi}_j^{\bar{a}}-\pi_j^{\bar{a}}))\,\Big|_{r=0}}_{=0}\Big\}\Big(\hat{g}(\bar{H}_t)-g(\bar{H}_t)\Big)\Bigg] \tag{207}$$

$$=\mathbb{E}\Bigg[\Big\{\frac{\mathrm{d}}{\mathrm{d}r}\omega_t^{\bar{a},\bar{b}}(\bar{H}_t;\pi_j^{\bar{a}}+r(\hat{\pi}_j^{\bar{a}}-\pi_j^{\bar{a}}))\,\Big|_{r=0}\Big(\gamma_t^{\bar{a},\bar{b}}(\bar{Z}_{t+\tau})-\mu_t^{\bar{a},\bar{b}}(\bar{H}_t)\Big)\Big\}\Big(\hat{g}(\bar{H}_t)-g(\bar{H}_t)\Big)\Bigg] \tag{208}$$

$$=\mathbb{E}\Bigg[\mathbb{E}\Big[\Big\{\frac{\mathrm{d}}{\mathrm{d}r}\omega_t^{\bar{a},\bar{b}}(\bar{H}_t;\pi_j^{\bar{a}}+r(\hat{\pi}_j^{\bar{a}}-\pi_j^{\bar{a}}))\,\Big|_{r=0}\Big(\gamma_t^{\bar{a},\bar{b}}(\bar{Z}_{t+\tau})-\mu_t^{\bar{a},\bar{b}}(\bar{H}_t)\Big)\Big\}\Big(\hat{g}(\bar{H}_t)-g(\bar{H}_t)\Big)\,\Big|\,\bar{H}_t\Big]\Bigg] \tag{209}$$

$$=\mathbb{E}\Bigg[\Big\{\frac{\mathrm{d}}{\mathrm{d}r}\omega_t^{\bar{a},\bar{b}}(\bar{H}_t;\pi_j^{\bar{a}}+r(\hat{\pi}_j^{\bar{a}}-\pi_j^{\bar{a}}))\,\Big|_{r=0}\Big(\underbrace{\mathbb{E}\big[\gamma_t^{\bar{a},\bar{b}}(\bar{Z}_{t+\tau})\,\big|\,\bar{H}_t\big]}_{=\mu_t^{\bar{a},\bar{b}}(\bar{H}_t)}-\mu_t^{\bar{a},\bar{b}}(\bar{H}_t)\Big)\Big\}\Big(\hat{g}(\bar{H}_t)-g(\bar{H}_t)\Big)\Bigg] \tag{210}$$

$$=0, \tag{211}$$

where Equation 205 follows from Lemma C.8, in Equation 207 from Lemma C.1, and in Equation 210 follows from Lemma C.9.

The pathwise derivative of $D_g \mathcal{L}(g; \eta^{\bar{a}, \bar{b}})[\hat{g} - g]$ with respect to the nuisances $\mu_j^{\bar{a}}$ for $j = t, \ldots, t + \tau$ is given by

$$D_{\mu_j^{\bar{a}}} D_g \mathcal{L}(g; \eta^{\bar{a}, \bar{b}})[\hat{g} - g, \hat{\mu}_j^{\bar{a}} - \mu_j^{\bar{a}}] \tag{212}$$

$$= \frac{\mathrm{d}}{\mathrm{d}r} D_g \mathcal{L}\Big(g; \{\pi_j^{\bar{a}}, \pi_j^{\bar{b}}, \omega_j^{\bar{a}}, \omega_j^{\bar{b}}, \mu_j^{\bar{b}}\}_{j=t}^{t+\tau} \cup \{\mu_0^{\bar{a}}, \ldots, \mu_j^{\bar{a}} + r(\hat{\mu}_j^{\bar{a}} - \mu_j^{\bar{a}}), \ldots, \mu_{t+\tau}^{\bar{a}}\}\Big)[\hat{g} - g]\,\Big|_{r=0} \tag{213}$$

$$\propto \frac{\mathrm{d}}{\mathrm{d}r} \mathbb{E}\Bigg[\Big\{\rho_t^{\bar{a}, \bar{b}}(\bar{Z}_{t+\tau})\Big(\mu_t^{\bar{a}, \bar{b}}(\bar{H}_t; \mu_j^{\bar{a}} + r(\hat{\mu}_j^{\bar{a}} - \mu_j^{\bar{a}})) - g(\bar{H}_t)\Big) \tag{214}$$

$$+ \omega_t^{\bar{a}, \bar{b}}(\bar{H}_t)\Big(\gamma_t^{\bar{a}, \bar{b}}(\bar{Z}_{t+\tau}; \mu_j^{\bar{a}} + r(\hat{\mu}_j^{\bar{a}} - \mu_j^{\bar{a}})) - \mu_t^{\bar{a}, \bar{b}}(\bar{H}_t; \mu_j^{\bar{a}} + r(\hat{\mu}_j^{\bar{a}} - \mu_j^{\bar{a}}))\Big)\Big\}\Big(\hat{g}(\bar{H}_t) - g(\bar{H}_t)\Big)\Bigg]\Bigg|_{r=0} \tag{215}$$

$$= \mathbb{E}\Bigg[\Big\{\rho_t^{\bar{a}, \bar{b}}(\bar{Z}_{t+\tau}) \frac{\mathrm{d}}{\mathrm{d}r} \mu_t^{\bar{a}, \bar{b}}(\bar{H}_t; \mu_j^{\bar{a}} + r(\hat{\mu}_j^{\bar{a}} - \mu_j^{\bar{a}}))\,\Big|_{r=0} \tag{216}$$

$$+ \omega_t^{\bar{a}, \bar{b}}(\bar{H}_t)\Big(\underbrace{\frac{\mathrm{d}}{\mathrm{d}r} \gamma_t^{\bar{a}, \bar{b}}(\bar{Z}_{t+\tau}; \mu_j^{\bar{a}} + r(\hat{\mu}_j^{\bar{a}} - \mu_j^{\bar{a}}))\,\Big|_{r=0} - \frac{\mathrm{d}}{\mathrm{d}r} \mu_t^{\bar{a}, \bar{b}}(\bar{H}_t; \mu_j^{\bar{a}} + r(\hat{\mu}_j^{\bar{a}} - \mu_j^{\bar{a}}))\,\Big|_{r=0}}_{=0}\Big)\Big\}\Big(\hat{g}(\bar{H}_t) - g(\bar{H}_t)\Big)\Bigg] \tag{217}$$

$$= \mathbb{E}\Bigg[\Big\{\rho_t^{\bar{a}, \bar{b}}(\bar{Z}_{t+\tau}) \frac{\mathrm{d}}{\mathrm{d}r} \mu_t^{\bar{a}, \bar{b}}(\bar{H}_t; \mu_j^{\bar{a}} + r(\hat{\mu}_j^{\bar{a}} - \mu_j^{\bar{a}}))\,\Big|_{r=0} \tag{218}$$

$$- \omega_t^{\bar{a}, \bar{b}}(\bar{H}_t)\Big(\frac{\mathrm{d}}{\mathrm{d}r} \mu_t^{\bar{a}, \bar{b}}(\bar{H}_t; \mu_j^{\bar{a}} + r(\hat{\mu}_j^{\bar{a}} - \mu_j^{\bar{a}}))\,\Big|_{r=0}\Big)\Big\}\Big(\hat{g}(\bar{H}_t) - g(\bar{H}_t)\Big)\Bigg] \tag{219}$$

$$= \mathbb{E}\Bigg[\mathbb{E}\Big[\Big\{\rho_t^{\bar{a}, \bar{b}}(\bar{Z}_{t+\tau}) \frac{\mathrm{d}}{\mathrm{d}r} \mu_t^{\bar{a}, \bar{b}}(\bar{H}_t; \mu_j^{\bar{a}} + r(\hat{\mu}_j^{\bar{a}} - \mu_j^{\bar{a}}))\,\Big|_{r=0} \tag{220}$$

$$- \omega_t^{\bar{a}, \bar{b}}(\bar{H}_t) \frac{\mathrm{d}}{\mathrm{d}r} \mu_t^{\bar{a}, \bar{b}}(\bar{H}_t; \mu_j^{\bar{a}} + r(\hat{\mu}_j^{\bar{a}} - \mu_j^{\bar{a}}))\,\Big|_{r=0}\Big\}\Big(\hat{g}(\bar{H}_t) - g(\bar{H}_t)\Big)\,\Big|\,\bar{H}_t\Big]\Bigg] \tag{221}$$

$$= \mathbb{E}\Bigg[\underbrace{\mathbb{E}\Big[\Big\{\rho_t^{\bar{a}, \bar{b}}(\bar{Z}_{t+\tau})\,\Big|\,\bar{H}_t\Big]}_{=\omega_t^{\bar{a}, \bar{b}}(\bar{H}_t)} \frac{\mathrm{d}}{\mathrm{d}r} \mu_t^{\bar{a}, \bar{b}}(\bar{H}_t; \mu_j^{\bar{a}} + r(\hat{\mu}_j^{\bar{a}} - \mu_j^{\bar{a}}))\,\Big|_{r=0} \tag{222}$$

$$- \omega_t^{\bar{a}, \bar{b}}(\bar{H}_t) \frac{\mathrm{d}}{\mathrm{d}r} \mu_t^{\bar{a}, \bar{b}}(\bar{H}_t; \mu_j^{\bar{a}} + r(\hat{\mu}_j^{\bar{a}} - \mu_j^{\bar{a}}))\,\Big|_{r=0}\Big\}\Big(\hat{g}(\bar{H}_t) - g(\bar{H}_t)\Big)\Bigg] \tag{223}$$

$$= 0, \tag{224}$$

where Equation 217 follows from Lemma C.7, and Equation 222 from Lemma C.10.

Finally, the pathwise derivative of $D_g\mathcal{L}(g;\eta^{\bar{a},\bar{b}})[\hat{g}-g]$ with respect to the nuisances $\omega_j^{\bar{a}}$ for $j = t,\dots,t+\tau$ is given by

$$D_{\omega_j^{\bar{a}}} D_g\mathcal{L}(g;\eta^{\bar{a},\bar{b}})[\hat{g}-g,\hat{\omega}_j^{\bar{a}}-\omega_j^{\bar{a}}] \tag{225}$$

$$=\frac{\mathrm{d}}{\mathrm{d}r} D_g\mathcal{L}\Big(g;\{\pi_j^{\bar{a}},\pi_j^{\bar{b}},\mu_j^{\bar{a}},\mu_j^{\bar{b}},\omega_j^{\bar{b}}\}_{j=t}^{t+\tau}\cup\{\omega_0^{\bar{a}},\dots,\omega_j^{\bar{a}}+r(\hat{\omega}_j^{\bar{a}}-\omega_j^{\bar{a}}),\dots,\omega_{t+\tau}^{\bar{a}}\}\Big)[\hat{g}-g]\Big|_{r=0} \tag{226}$$

$$\propto\frac{\mathrm{d}}{\mathrm{d}r}\mathbb{E}\Bigg[\Big\{\rho_t^{\bar{a},\bar{b}}(\bar{Z}_{t+\tau};\omega_j^{\bar{a}}+r(\hat{\omega}_j^{\bar{a}}-\omega_j^{\bar{a}}))\Big(\mu_t^{\bar{a},\bar{b}}(\bar{H}_t)-g(\bar{H}_t)\Big) \tag{227}$$

$$+\omega_t^{\bar{a},\bar{b}}(\bar{H}_t;\omega_j^{\bar{a}}+r(\hat{\omega}_j^{\bar{a}}-\omega_j^{\bar{a}}))\Big(\gamma_t^{\bar{a},\bar{b}}(\bar{Z}_{t+\tau})-\mu_t^{\bar{a},\bar{b}}(\bar{H}_t)\Big)\Big\}\Big(\hat{g}(\bar{H}_t)-g(\bar{H}_t)\Big)\Bigg]\Bigg|_{r=0} \tag{228}$$

$$=\mathbb{E}\Bigg[\Big\{\underbrace{\frac{\mathrm{d}}{\mathrm{d}r}\rho_t^{\bar{a},\bar{b}}(\bar{Z}_{t+\tau};\omega_j^{\bar{a}}+r(\hat{\omega}_j^{\bar{a}}-\omega_j^{\bar{a}}))\Big|_{r=0}}_{=0}\Big(\mu_t^{\bar{a},\bar{b}}(\bar{H}_t)-g(\bar{H}_t)\Big) \tag{229}$$

$$+\frac{\mathrm{d}}{\mathrm{d}r}\omega_t^{\bar{a},\bar{b}}(\bar{H}_t;\omega_j^{\bar{a}}+r(\hat{\omega}_j^{\bar{a}}-\omega_j^{\bar{a}}))\Big|_{r=0}\Big(\gamma_t^{\bar{a},\bar{b}}(\bar{Z}_{t+\tau})-\mu_t^{\bar{a},\bar{b}}(\bar{H}_t)\Big)\Big\}\Big(\hat{g}(\bar{H}_t)-g(\bar{H}_t)\Big)\Bigg] \tag{230}$$

$$=\mathbb{E}\Bigg[\frac{\mathrm{d}}{\mathrm{d}r}\omega_t^{\bar{a},\bar{b}}(\bar{H}_t;\omega_j^{\bar{a}}+r(\hat{\omega}_j^{\bar{a}}-\omega_j^{\bar{a}}))\Big|_{r=0}\Big(\gamma_t^{\bar{a},\bar{b}}(\bar{Z}_{t+\tau})-\mu_t^{\bar{a},\bar{b}}(\bar{H}_t)\Big)\Big(\hat{g}(\bar{H}_t)-g(\bar{H}_t)\Big)\Bigg] \tag{231}$$

$$=\mathbb{E}\Bigg[\mathbb{E}\Big[\frac{\mathrm{d}}{\mathrm{d}r}\omega_t^{\bar{a},\bar{b}}(\bar{H}_t;\omega_j^{\bar{a}}+r(\hat{\omega}_j^{\bar{a}}-\omega_j^{\bar{a}}))\Big|_{r=0}\Big(\gamma_t^{\bar{a},\bar{b}}(\bar{Z}_{t+\tau})-\mu_t^{\bar{a},\bar{b}}(\bar{H}_t)\Big)\Big(\hat{g}(\bar{H}_t)-g(\bar{H}_t)\Big)\Big|\bar{H}_t\Big]\Bigg] \tag{232}$$

$$=\mathbb{E}\Bigg[\frac{\mathrm{d}}{\mathrm{d}r}\omega_t^{\bar{a},\bar{b}}(\bar{H}_t;\omega_j^{\bar{a}}+r(\hat{\omega}_j^{\bar{a}}-\omega_j^{\bar{a}}))\Big|_{r=0}\Big(\underbrace{\mathbb{E}\Big[\gamma_t^{\bar{a},\bar{b}}(\bar{Z}_{t+\tau})\Big|\bar{H}_t\Big]}_{=\mu_t^{\bar{a},\bar{b}}(\bar{H}_t)}-\mu_t^{\bar{a},\bar{b}}(\bar{H}_t)\Big)\Big(\hat{g}(\bar{H}_t)-g(\bar{H}_t)\Big)\Bigg] \tag{233}$$

$$=0, \tag{234}$$

where Equation 229 follows from Lemma C.8, and Equation 233 from Lemma C.9.

$\square$

### C.3 GENERALIZING THE R-LEARNER

**Remark:** *For a single-step-ahead prediction $\tau = 0$ (i.e., when there is **no** time-varying confounding as in the static setting), the R-learner has the same overlap weights as our **WO**-learner for CATE.*

*Proof.* We show how our weighted population risk function reduces for $\tau = 0$ and leverage previous findings on the identity of the R-learner (Nie & Wager, 2021).

Notice that under our identifiability assumptions, for a single-step ahead prediction $\tau = 0$, conditioning on the observed history (i.e., a backdoor-adjustment) is sufficient to adjust for all confounders, as *there is no time-varying confounding*.

Hence, for $\tau = 0$, we can treat the observed history $\bar{H}_t$ as a fixed set of covariates (typically denoted as $X$), the treatment variable $A_t \in \{0, 1\}$ as well as the intervention $\bar{a} = a_t \in \{0, 1\}$ as a binary treatment (denoted as $A$ and $a$, respectively), and the outcome $Y_t$ as the instantaneous outcome (denoted as $Y$). Finally, $\bar{Z}_t$ summarizes all variables $(\bar{H}_t, A_t, Y_t)$, which corresponds to $(X, A, Y)$ in the static setting (see Figure 6).

Let $\tau = 0$. Then, the pseudo-outcomes and weights simplify as

$$\rho_t^{\bar{a}}(\bar{Z}_t) \tag{235}$$

$$= \prod_{j=t}^{t+\tau} \pi_j^{\bar{a}}(\bar{H}_j) + \sum_{j=t}^{t+\tau} \left( \mathbb{1}_{\{a_j = A_j\}} - \pi_j^{\bar{a}}(\bar{H}_j) \right) \omega_{j+1}^{\bar{a}}(\bar{H}_j) \prod_{t \le k < j} \pi_k^{\bar{a}}(\bar{H}_k) \tag{236}$$

$$= \pi_t^{\bar{a}}(\bar{H}_t) + \left( \mathbb{1}_{\{a_t = A_t\}} - \pi_t^{\bar{a}}(\bar{H}_t) \right) \omega_{t+1}^{\bar{a}}(\bar{H}_t) \underbrace{\prod_{t \le k < t} \pi_k^{\bar{a}}(\bar{H}_k)}_{=0} \tag{237}$$

$$\underbrace{\phantom{= \pi_t^{\bar{a}}(\bar{H}_t) + \left( \mathbb{1}_{\{a_t = A_t\}} - \pi_t^{\bar{a}}(\bar{H}_t) \right) \omega_{t+1}^{\bar{a}}(\bar{H}_t)}}_{=0}$$

$$= \pi_t^{\bar{a}}(\bar{H}_t), \tag{238}$$

and

$$\omega_t^{\bar{a}}(\bar{H}_t) = \mathbb{E}\left[ \pi_t^{\bar{a}}(\bar{H}_t) \,\Big|\, \bar{H}_t \right] = \pi_t^{\bar{a}}(\bar{H}_t), \tag{239}$$

such that

$$\rho_t^{\bar{a},\bar{b}}(\bar{Z}_t) \tag{240}$$

$$= \rho_t^{\bar{a}}(\bar{H}_t)\omega_t^{\bar{b}}(\bar{H}_t) + \rho_t^{\bar{b}}(\bar{H}_t)\omega_t^{\bar{a}}(\bar{H}_t) - \omega_t^{\bar{a}}(\bar{H}_t)\omega_t^{\bar{b}}(\bar{H}_t) \tag{241}$$

$$= \pi_t^{\bar{a}}(\bar{H}_t)\pi_t^{\bar{b}}(\bar{H}_t) + \pi_t^{\bar{a}}(\bar{H}_t)\pi_t^{\bar{b}}(\bar{H}_t) - \omega_t^{\bar{a}}(\bar{H}_t)\omega_t^{\bar{b}}(\bar{H}_t) \tag{242}$$

$$= \omega_t^{\bar{a}}(\bar{H}_t)\omega_t^{\bar{b}}(\bar{H}_t) \tag{243}$$

$$= \pi_t^{\bar{a}}(\bar{H}_t)\pi_t^{\bar{b}}(\bar{H}_t) \tag{244}$$

$$= \pi_t^{\bar{a}}(\bar{H}_t)(1 - \pi_t^{\bar{a}}(\bar{H}_t)) \tag{245}$$

and finally

$$\xi_t^{\bar{a},\bar{b}}(\bar{Z}_t) \tag{246}$$

$$= \mu_t^{\bar{a},\bar{b}}(\bar{H}_t) + \frac{\omega_t^{\bar{a},\bar{b}}(\bar{H}_t)}{\rho_t^{\bar{a},\bar{b}}(\bar{Z}_t)} \left( \gamma_t^{\bar{a},\bar{b}}(\bar{Z}_t) - \mu_t^{\bar{a},\bar{b}}(\bar{H}_t) \right) \tag{247}$$

$$= \mu_t^{\bar{a},\bar{b}}(\bar{H}_t) + \frac{\pi_t^{\bar{a}}(\bar{H}_t)(1 - \pi_t^{\bar{a}}(\bar{H}_t))}{\pi_t^{\bar{a}}(\bar{H}_t)(1 - \pi_t^{\bar{a}}(\bar{H}_t))} \left( \gamma_t^{\bar{a},\bar{b}}(\bar{Z}_t) - \mu_t^{\bar{a},\bar{b}}(\bar{H}_t) \right) \tag{248}$$

$$= \gamma_t^{\bar{a},\bar{b}}(\bar{Z}_t) \tag{249}$$

$$= \gamma_t^{\bar{a},1-\bar{a}}(\bar{Z}_t), \tag{250}$$

where $\gamma_t^{\bar{a},1-\bar{a}}(\bar{Z}_t)$ simplify to the DR pseudo-outcomes for CATE in the static setting (Curth & van der Schaar, 2021; Frauen et al., 2025a). As shown by Morzywolek et al. (2023) and highlighted by

other works (Chernozhukov et al., 2024; Fisher, 2024), the R-learner is an *overlap-weighted DR learner* and, hence, minimizes the loss

$$\mathcal{L}(g; \eta^a) = \frac{1}{\mathbb{E}\left[\pi^a(X)(1 - \pi^a(X))\right]} \mathbb{E}\left[\pi^a(X)(1 - \pi^a(X))\left(\gamma^{a,1-a}(X, A, Y) - g\right)^2\right]. \quad (251)$$

Since for $\tau = 0$, we only need one set of nuisances for CATE, i.e., $\eta^{\bar{a}} = \eta^{\bar{b}} = \eta^{\bar{a},\bar{b}}$, and it follows that Equation 251 exactly mirrors

$$\mathcal{L}(g; \eta^{\bar{a}}) = \frac{1}{\mathbb{E}\left[\pi_t^{\bar{a}}(\bar{H}_t)(1 - \pi_t^{\bar{a}}(\bar{H}_t))\right]} \mathbb{E}\left[\pi_t^{\bar{a}}(\bar{H}_t)(1 - \pi_t^{\bar{a}}(\bar{H}_t))\left(\gamma_t^{\bar{a},1-\bar{a}}(\bar{Z}_t) - g\right)^2\right] \quad (252)$$

in our time-varying notation. $\qquad \square$

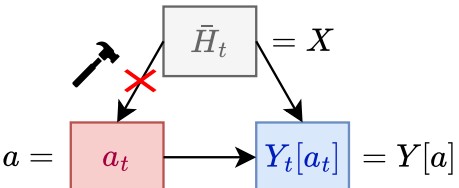

Figure 6: **One-step ahead prediction.** For a one-step ahead prediction $\tau = 0$, there is no time-varying confounding. Hence, we can treat the observed history $\bar{H}_t$ as a fixed set of covariates $X$, and the single intervention and instantaneous outcomes as in the static setting. Our **WO**-learner for CATE then simplifies to the R-learner in the static setting.

### C.4 Uniformly bounded variance in low-overlap regimes

In the following, we show how our overlap-weights stabilize the variance in low-overlap regimes. For this, we simplify notation and focus on the $\tau = 0$ case. Longer prediction horizons $\tau > 0$ follow **completely analogously**.

For our proof, we first show that the DR pseudo-outcome has conditional variance of order $1/\omega_t^{1,0}(\bar{H}_t)$, while our **WO**-learner uses pseudo-outcomes with *uniformly bounded* variance, that are **independent** of overlap. Thereby, we demonstrate how our overlap weights stabilize estimation in low-overlap regimes.

Throughout, we assume bounded conditional variances of the potential outcomes:

$$0 < \sigma_{\min}^2 \ \leq \ \mathrm{Var}(Y_t[a_t] \mid \bar{H}_t) \ \leq \ \sigma_{\max}^2 < \infty, \qquad a_t \in \{0, 1\}. \tag{253}$$

#### C.4.1 Variance of the DR Pseudo-Outcome

Recall that the DR pseudo-outcome for CATE where $\tau = 0$ is

$$\gamma_t^{1,0}(\bar{Z}_t) = \mu_t^1(\bar{H}_t) - \mu_t^0(\bar{H}_t) + \frac{A_t}{\pi_t^1(\bar{H}_t)}\big(Y_t - \mu_t^1(\bar{H}_t)\big) - \frac{1 - A_t}{\pi_t^0(\bar{H}_t)}\big(Y_t - \mu_t^0(\bar{H}_t)\big). \tag{254}$$

**Lemma C.13** (DR variance inflation under low overlap). *Let the potential outcome satisfy bounded conditional variance as in Equation 253. Then, the conditional variance satisfies*

$$\sigma_{\min}^2 \cdot \frac{1}{\omega_t^{1,0}(\bar{H}_t)} \ \leq \ \mathrm{Var}\big(\gamma_t^{1,0}(\bar{Z}_t) \mid \bar{H}_t\big) \ \leq \ \sigma_{\max}^2 \cdot \frac{1}{\omega_t^{1,0}(\bar{H}_t)} \qquad a.s.$$

*In particular, as overlap $\omega_t 1, 0(\bar{H}_t) \to 0$, the **conditional variance of the DR pseudo-outcome diverges** at rate $1/\omega_t^{1,0}(\bar{H}_t)$.*

*Proof.* First, using Equation 254 and conditioning on $\bar{H}_t$,

$$\mathbb{E}[\gamma_t^{1,0}(\bar{Z}_t) \mid \bar{H}_t] \tag{255}$$
$$= \mu_t^1(\bar{H}_t) - \mu_t^0(\bar{H}_t) + \mathbb{E}\left[\frac{A_t}{\pi_t^1(\bar{H}_t)}(Y_t - \mu_t^1(\bar{H}_t)) \mid \bar{H}_t\right] - \mathbb{E}\left[\frac{1 - A_t}{\pi_t^0(\bar{H}_t)}(Y_t - \mu_t^0(\bar{H}_t)) \mid \bar{H}_t\right]. \tag{256}$$

The two expectations vanish because

$$\mathbb{E}[Y_t - \mu_t^1(\bar{H}_t) \mid \bar{H}_t, A_t = 1] = 0 \tag{257}$$

and

$$\mathbb{E}[Y_t - \mu_t^0(\bar{H}_t) \mid \bar{H}_t, A_t = 0] = 0, \tag{258}$$

such that

$$\mathbb{E}[\gamma(\bar{Z}_t) \mid \bar{H}_t] = \mu_t^{1,0}(\bar{H}_t). \tag{259}$$

Now, let the correction term be

$$U_t := \frac{A_t}{\pi_t^1(\bar{H}_t)}(Y_t - \mu_t^1(\bar{H}_t)) - \frac{1 - A_t}{\pi_t^0(\bar{H}_t)}(Y_t - \mu_t^0(\bar{H}_t)), \tag{260}$$

so that

$$\gamma_t^{1,0}(\bar{Z}_t) = \mu_t^{1,0}(\bar{H}_t) + U_t \tag{261}$$

and

$$\text{Var}(\gamma_t^{1,0}(\bar{Z}_t) \mid \bar{H}_t) = \text{Var}(U_t \mid \bar{H}_t). \tag{262}$$

Next, we condition further on $A_t$, which yields

$$\text{Var}(U_t \mid \bar{H}_t) = \mathbb{E}[\,\text{Var}(U_t \mid \bar{H}_t, A_t) \mid \bar{H}_t]. \tag{263}$$

If $A_t = 1$, then

$$U_t = (Y_t - \mu_t^1(\bar{H}_t))/\pi_t^1(\bar{H}_t) \tag{264}$$

and hence

$$\text{Var}(U_t \mid \bar{H}_t, A_t = 1) = \sigma_1^2(\bar{H}_t)/\pi_t^1(\bar{H}_t)^2. \tag{265}$$

If $A_t = 0$, then

$$\text{Var}(U_t \mid \bar{H}_t, A_t = 0) = \sigma_0^2(\bar{H}_t)/\pi_t^0(\bar{H}_t)^2. \tag{266}$$

Using $\mathbb{P}(A_t = 1 \mid \bar{H}_t) = \pi_t^1(\bar{H}_t)$,

$$\text{Var}(U_t \mid \bar{H}_t) = \frac{\sigma_1^2(\bar{H}_t)}{\pi_t^1(\bar{H}_t)} + \frac{\sigma_0^2(\bar{H}_t)}{\pi_t^0(\bar{H}_t)}. \tag{267}$$

Finally,

$$\frac{1}{\pi_t^1(\bar{H}_t)} + \frac{1}{\pi_t^0(\bar{H}_t)} = \frac{\pi_t^1(\bar{H}_t) + \pi_t^0(\bar{H}_t)}{\pi_t^1(\bar{H}_t)\pi_t^0(\bar{H}_t)} = \frac{1}{\pi_t^1(\bar{H}_t)\pi_t^0(\bar{H}_t)} = \frac{1}{\omega(\bar{H}_t)}, \tag{268}$$

and applying the bounds Equation 253 yields the claim. $\qquad\square$

### C.4.2 STABILIZED VARIANCE WITH OUR OVERLAP-WEIGHTS

Recall that for $\tau = 0$, our **WO**-learner minimizes by Theorem 4.3 the weighted oracle risk

$$\mathcal{L}(g; \eta^{1,0}) = \frac{1}{\mathbb{E}[\omega_t^{1,0}(\bar{H}_t)]}\mathbb{E}\big[\omega_t^{1,0}(\bar{H}_t)\,(\gamma_t(\bar{Z}_t) - g(\bar{H}_t))^2\big]. \tag{269}$$

In the following, we define the transformed pseudo-outcome and prediction as

$$\tilde{\gamma}_t^{1,0}(\bar{Z}_t) := \sqrt{\omega_t^{1,0}(\bar{H}_t)}\,\gamma_t^{1,0}(\bar{Z}_t), \qquad \tilde{g}(\bar{H}_t) := \sqrt{\omega_t^{1,0}(\bar{H}_t)}\,g(\bar{H}_t). \tag{270}$$

**Lemma C.14** (Equivalent transformed risk). *For any g,*

$$\mathcal{L}(g; \eta^{1,0}) = \frac{1}{\mathbb{E}[\omega_t^{1,0}(\bar{H}_t)]}\mathbb{E}\Big[\big(\tilde{\gamma}_t^{1,0}(\bar{Z}_t) - \tilde{g}(\bar{H}_t)\big)^2\Big]. \tag{271}$$

*Proof.* We simply factor $\sqrt{\omega_t^{1,0}(\bar{H}_t)}$ inside the square:

$$\omega_t^{1,0}(\bar{H}_t)\big(\gamma_t^{1,0}(\bar{Z}_t) - g(\bar{H}_t)\big)^2 = \big(\sqrt{\omega_t^{1,0}(\bar{H}_t)}\,\gamma_t^{1,0}(\bar{Z}_t) - \sqrt{\omega_t^{1,0}(\bar{H}_t)}\,g(\bar{H}_t)\big)^2. \tag{272}$$

$\qquad\square$

In the following, we analyze the conditional variance of $\tilde{\gamma}_t^{1,0}(\bar{Z}_t)$.

**Lemma C.15** (Bounded variance using overlap weights). *Under the assumptions of Lemma C.13,*

$$\text{Var}\Big(\tilde{\gamma}_t^{1,0}(\bar{Z}_t) \mid \bar{H}_t\Big) \in [\sigma_{\min}^2, \sigma_{\max}^2] \qquad a.s. \tag{273}$$

*In particular, the conditional variance of $\tilde{\gamma}_t^{1,0}(\bar{Z}_t)$ is **uniformly bounded** and **independent of** $\omega_t^{1,0}(\bar{H}_t)$.*

*Proof.* First, we again write $\gamma_t^{1,0}(\bar{Z}_t) = \mu_t^{1,0}(\bar{H}_t) + U_t$ with $\mathbb{E}[U_t \mid \bar{H}_t] = 0$. Lemma C.13 shows that

$$\mathrm{Var}(U_t \mid \bar{H}_t) \in \left[ \sigma_{\min}^2 \cdot \frac{1}{\omega_t^{1,0}(\bar{H}_t)}, \ \sigma_{\max}^2 \cdot \frac{1}{\omega_t^{1,0}(\bar{H}_t)} \right]. \tag{274}$$

Since $\tilde{\gamma}_t^{1,0}(\bar{Z}_t) = \sqrt{\omega_t^{1,0}(\bar{H}_t)}\gamma_t^{1,0}(\bar{Z}_t) = \tilde{\mu}_t^{1,0}(\bar{H}_t) + \tilde{U}_t$ with $\tilde{\mu}_t^{1,0} = \sqrt{\omega_t^{1,0}(\bar{H}_t)}\mu_t^{1,0}$ and $\tilde{U}_t = \sqrt{\omega_t^{1,0}(\bar{H}_t)}\, U_t$, it follows that

$$\mathrm{Var}(\tilde{\gamma}_t^{1,0}(\bar{Z}_t) \mid \bar{H}_t) = \mathrm{Var}(\tilde{U}_t \mid \bar{H}_t) = \omega_t^{1,0}(\bar{H}_t)\,\mathrm{Var}(U_t \mid \bar{H}_t). \tag{275}$$

Substituting the bounds yields

$$\sigma_{\min}^2 \ \leq \ \mathrm{Var}(\tilde{\gamma}_t^{1,0}(\bar{Z}_t) \mid \bar{H}_t) \ \leq \ \sigma_{\max}^2, \tag{276}$$

as claimed. $\qquad\square$

# D    DETAILS ON THE DATA-GENERATING PROCESSES

## D.1    SYNTHETIC DATA GENERATION

We now describe the data-generating processes for the synthetic datasets $\mathcal{D}^\gamma$ *(low-overlap regime)*, $\mathcal{D}^\pi$ *(complex propensity)*, $\mathcal{D}^\mu$ *(complex response function)*, and $\mathcal{D}^N$ *(low-sample setting)*. All of them have the following general structure:

As in Frauen et al. (2025a), for each $* \in \{\gamma, \pi, \mu, N\}$, we first simulate an initial confounder $X_0 \sim \mathcal{N}(0,1)$. Then, for time steps $t = 1, \ldots, T^*$, we generate $d_x^*$-dimensional time-varying confounders via

$$X_t = (X_{t,1}, \ldots, X_{t,d_x^*}) = f_x^*(Y_{t-1}, A_{t-1}, \bar{H}_{t-1}) + \varepsilon_x \tag{277}$$

and time-varying treatments via

$$A_t \sim \sigma\left(f_a^*(\bar{H}_t)\right), \tag{278}$$

where $\sigma(\cdot)$ is the sigmoid function. The outcomes are then simulated via

$$Y_t = f_y^*(A_t, \bar{H}_t) + \epsilon_y. \tag{279}$$

with $\varepsilon_y \sim \mathcal{N}(0, 0.3^2)$. For each dataset, we simulate $n^*$ samples for training and 1000 samples for testing. For the test set, we always generate the ground-truth CATE of a $\tau^*$-step ***always treat*** against a $\tau^*$-step ***never treat*** intervention.

We provide the specific configurations of $f_x^*, f_a^*, f_y^*, \tau^*, d_x^*, T^*$ and $n^*$ below:

**(1) Low-overlap regime $\mathcal{D}^\gamma$:** For $\mathcal{D}^\gamma$, we set $\tau^\gamma = 1$, $d_x^\gamma = 1$, $T^\gamma = 5$, and $n^\gamma = 4000$. The covariates are generated via $f_x^\gamma(Y_{t-1}, A_{t-1}, \bar{H}_{t-1}) = 0.5X_{t-1}$, the treatments via $f_a^\gamma(\bar{H}_t) = \gamma(0.5X_t + 0.5Y_{t-1} - 0.5(A_{t-1} - 0.5))$, where $\gamma$ controls the overlap strength, and the outcomes via $f_y^\gamma(A_t, \bar{H}_t) = 0.5\exp(-X_t^2)(A_t - 0.5)$. In order to decrease the overlap, we vary the overlap parameter $\gamma \in \{0.5, 1.0, 1.5, \ldots, 6.5\}$.

**(2) Complex treatment $\mathcal{D}^\pi$:** For $\mathcal{D}^\pi$, we set $d_x^\pi = 1$, $T^\pi = 15$, and $n^\pi = 4000$. The covariates are generated via $f_x^\pi(Y_{t-1}, A_{t-1}, \bar{H}_{t-1}) = 0.5X_{t-1}$, the treatments via $f_a^\pi(\bar{H}_t) = \sin(0.5X_t + 0.5Y_{t-1} - 0.5(A_{t-1} - 0.5))$, and the outcomes via $f_y^\pi(A_t, \bar{H}_t) = 0.5\exp(-X_t^2)(A_t - 0.5)$. In order to increase the complexity of the treatment propensity, we increase the prediction horizon $\tau^\pi \in \{1, 3, 5, 7\}$.

**(3) Complex response $\mathcal{D}^\mu$:** For $\mathcal{D}^\mu$, we set $\tau^\mu = 1$, $T^\mu = 15$, and $n^\mu = 4000$. The covariates are generated via $f_x^\pi(Y_{t-1}, A_{t-1}, \bar{H}_{t-1}) = 0.5X_{t-1}$, the treatments via $f_a^\pi(\bar{H}_t) = 0.5\sum_{p=1}^{d_x^\mu} X_{t,p}/d_x^\mu + 0.5Y_{t-1} - 0.5(A_{t-1} - 0.5)$, and the outcomes via $f_y^\pi(A_t, \bar{H}_t) = \exp(0.5(A_t - 0.5)\sum_{p=1}^{d_x^\mu}\cos(X_{t-1,p})\cos(\cos(X_{t-1,p}))/d_x^\mu$. In order to increase the complexity of the response function, we increase the dimensionality $d_x$ of the time-varying confounders $X_t = (X_{t,1}, \ldots, X_{t,d_x^\mu})$ via $d_x^\mu \in \{5, 10, 15, 20, 25, 30, 35\}$.

**(4) Low-sample setting $\mathcal{D}^N$:** For $\mathcal{D}^N$, we set $\tau^N = 1$, $d_x^N = 5$, and $T^N = 5$. The covariates are generated via $f_x^N(Y_{t-1}, A_{t-1}, \bar{H}_{t-1}) = 0.5X_{t-1}$, the treatments via $f_a^N(\bar{H}_t) = 3.5(0.5\sum_{p=1}^{d_x^\mu} X_{t,p}/d_x^N + 0.5Y_{t-1} - 0.5(A_{t-1} - 0.5))$, and the outcomes via $f_y^N(A_t, \bar{H}_t) = 0.5\exp(-(\sum_{p=1}^{d_x^N}\cos(X_{t,p})/d_x^N)^2)(A_t - 0.5)$. We vary the sample size for training the nuisance functions and second-stage estimators via $n^N \in \{8000, 7000, 6000, 5000, 4000, 3000, 2000\}$.

## D.2 SEMI-SYNTHETIC DATA GENERATION

For our semi-synthetic experiments, we employ the MIMIC-III (Wang et al., 2020) extract based on the MIMIC-III dataset (Johnson et al., 2016). Here, we use time-varying real-world covariates *heart rate, red blood cell count, sodium, mean blood pressure, systemic vascular resistance, glucose, chloride urine, glascow coma scale total, hematocrit, positive end-expiratory pressure set,* and *respiratory rate.* All measurements are aggregated at hourly levels. Further, we include *gender* and *age* as a static covariates. We summarize all covariates as $X_t = (X_{t,1}, \ldots, X_{t,d_x})$. Then, we simulate treatments $A_t$ and outcomes $Y_t$ based on these covariates.

Our data generating process is designed to have time-varying confounding, has a complex propensity score, and a complex response function. Specifically, we simulate the treatments via

$$A_t \sim \sigma\Big( f_a(\bar{H}_t) + \varepsilon_a \Big), \tag{280}$$

with $\varepsilon_a \sim \mathcal{N}(0, 0.2^2)$, $f_a(A_t, \bar{H}_t) = \sin(Y_{t-1})) - A_{t-1} \sum_{p=1}^{d_x} \sin(X_{t,p})/d_x$, and the outcomes via

$$Y_t = f_y(A_t, \bar{H}_t) + \varepsilon_y \tag{281}$$

with $f_y(A_t, \bar{H}_t) = Y_{t-1} + 2(A_t - 0.5) \exp\Big( 2(A_t - 0.5) \sin(Y_{t-1}) \sum_{p=1}^{d_x} \cos(X_{t,p})/(td_x) \Big)$ and $\varepsilon_y \sim \mathcal{N}(0, 0.1^2)$. We include trajectories of length $T = 20$, and simulate $n_{train} = 1500$ samples for training. For evaluation of CATE, we again compare a $\tau$-step *always treat* against a $\tau$-step *never treat* treatment intervention sequence.

# E    IMPLEMENTATION DETAILS

• **Implementation details:** We report implementation details for our transformer instantiation in Section 5. Here, we closely follow the setup by Frauen et al. (2025a) (see **Table 6**):

- All nuisance functions and second-stage estimators can be written as regression models that take the history $\bar{H}_t$ as input and learn some $\delta$-step-ahead outcome $\tilde{Y}_{t+\delta}$ (e.g., for the **HA**-learner, $\tilde{Y}_{t+\delta} = Y_{t+\delta}$).

- Hence, we parametrize each regression model as $g_\theta(\bar{h}_t) = g_\theta^2(g_\theta^1(\bar{h}_t))$, where $g_\theta^1$ is a representation function (in our main experiments: a standard transformer), and $g_\theta^2$ a read-out function (a standard multi-layer perceptron).

- As in (Frauen et al., 2025a), we learn the propensity scores $\pi_{t+j}^{\bar{a}}$ in a joint model, whereas we learn the response functions $\mu_{t+j}^{\bar{a}}$ and the weight functions $\omega_{t+j}^{\bar{a}}$ in separate models.

- *Representation function $g_\theta^1(\cdot)$:* For our main experiments in Section 5, we use an encoder transformer (Vaswani et al., 2017) with a single transformer block and a causal mask to avoid look-ahead bias, as well as non-trainable positional encodings. The transformer block has a self-attention mechanism with $d_{\text{att}}$ attention heads and a hidden state dimension $d_{\text{hid}}$, followed by a feed-forward network with hidden layer size $d_{\text{ff}}$. The self-attention mechanism and the feed-forward network use residual connections, followed by dropout layers with dropout probability $0.1$, and post-normalization for regularization.

- *Read-out function $g_\theta^2(\cdot)$:* We use a simple multilayer-perceptron with one hidden layer of size $d_{\text{mlp}}$, ReLU nonlinearities, and either a linear (regression) or softmax (classification) output activation.

We summarize all parameterizations in **Table 6**. To ensure a fair comparison, all nuisance models and second-stage estimators share, where appropriate, the exact same architecture and parametrization.

• **Runtime:** For each transformer-based learner, training took approximately $1.5$ minutes with $n_{\text{train}} = 4000$ samples and an AMD Ryzen 7 Pro CPU and 32GB of RAM. The runtime was comparable for our WO-learner and the existing meta-learners.

| Estimator | Hyperparameter | Configuration | HA | RA | IPW | DR | IVW | WO (*ours*) |
|---|---|---|---|---|---|---|---|---|
| Second-stage function | $d_{\text{att}}$
$d_{\text{hid}}$
$d_{\text{ff}}$
$d_{\text{mlp}}$
Learning rate
Number of epochs
Batch size | 3
30
20
20
0.001
100
64 | ✓ | ✓ | ✓ | ✓ | ✓ | ✓ |
| Response functions | $d_{\text{att}}$
$d_{\text{hid}}$
$d_{\text{ff}}$
$d_{\text{mlp}}$
Learning rate
Number of epochs
Batch size | 3
30
20
20
0.001
100
64 | ✗ | ✓ | ✗ | ✓ | ✓ | ✓ |
| Propensity score | $d_{\text{att}}$
$d_{\text{hid}}$
$d_{\text{ff}}$
$d_{\text{mlp}}$
Learning rate
Number of epochs
Batch size | 3
30
20
20
0.001
100
64 | ✗ | ✗ | ✓ | ✓ | ✓ | ✓ |
| IV weight functions | $d_{\text{att}}$
$d_{\text{hid}}$
$d_{\text{ff}}$
$d_{\text{mlp}}$
Learning rate
Number of epochs
Batch size | 3
30
20
20
0.001
100
64 | ✗ | ✗ | ✗ | ✗ | ✓ | ✗ |
| Overlap weight functions | $d_{\text{att}}$
$d_{\text{hid}}$
$d_{\text{ff}}$
$d_{\text{mlp}}$
Learning rate
Number of epochs
Batch size | 3
30
20
20
0.001
100
64 | ✗ | ✗ | ✗ | ✗ | ✗ | ✓ |

Table 6: **Hyperparameters** of our transformer instantiations for the nuisance function estimators and second-stage estimators. To ensure a fair comparison, the nuisance functions for all meta-learners share the exact same parametrization.

## F    SENSITIVITY ANALYSIS FOR NUISANCE MISSPECIFICATION

In Section 4 we show that our population risk is Neyman-orthogonal with respect to the esitmated nuisance functions. This means that that errors in estimated nuisances should enter the final CATE estimator only at higher order. In the following, we empirically validate this property. Here, we conduct a controlled sensitivity analysis in which we corrupt the estimated response-function nuisance $\mu_t^{\bar{a},\bar{b}}$ by a fixed bias and measure how this corruption propagates to the final-stage estimate of CATE.

For this, we use our data generating process $\mathcal{D}^\gamma$ and fix $\gamma = 3.5$ (for details, see Supplement D.1). We fist estimate the response-function nuisance $\widehat{\mu}_t^{\bar{a},\bar{b}}$, and then construct perturbed versions

$$\widehat{\mu}_{t,\delta}^{\bar{a},\bar{b}}(\cdot) \;=\; \widehat{\mu}_t^{\bar{a},\bar{b}}(\cdot) \;+\; \delta, \tag{282}$$

where $\delta \in \{0.0, 0.1, 0.2, \ldots, 1.0\}$ is a constant additive bias. Therein, we mimic systematic misspecification of the response function. For each value of $\delta$, we retrain the final-stage CATE model using the corrupted nuisance while keeping all other nuisance components fixed. The results are reported in Figure 7.

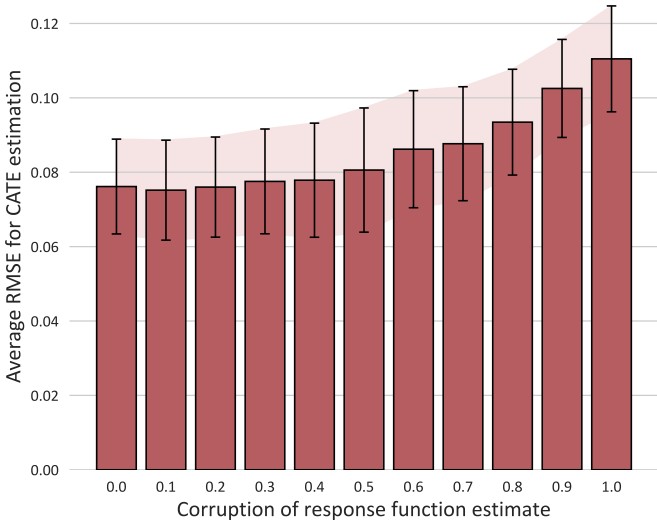

Figure 7: Sensitivity of CATE estimation with our **WO**-learner to corruption of the response-function nuisance. Reported are the average RMSE of the final CATE estimates against the injected bias.

The results show that even strong corruption of the response-function nuisance leads to only relatively low errors in the final CATE estimates. The error grows slowly and approximately linearly. This is consistent with the *higher-order* influence from our Neyman-orthogonality result (Theorem 4.5): the population risk is insensitive to first-order perturbations of nuisances such as $\mu_t^{\bar{a},\bar{b}}$, and errors enter only through second-order terms. In other words, the final estimator remains stable even under sizeable misspecification of the response-function nuisance.

# G    REAL-WORLD OUTCOME PREDICTION

In this section, we report the performance of our **WO**-learner on *factual* outcome prediction using real-world data. This experiment is *not* designed to evaluate causal validity; instead, it serves as a standard *sanity check* to confirm that the model components used in our meta-learner behave sensibly on real outcome trajectories. Such auxiliary diagnostics are commonly included in longitudinal causal studies, even though they do **not** speak to counterfactual accuracy.

In Table 7, we report the performance of the propensity-based meta-learners and the **HA** baseline performance. For this, we use the MIMIC-III dataset (Johnson et al., 2016; Wang et al., 2020). The outcome variable of interest is diastolic blood pressure, and we consider mechanical ventilation as the treatment variable. We further use 19 time-varying covariates such as cholesterol, respiratory rate, heart rate, and sodium, and further include gender as a static covariate for predicting the factual outcome. All measurements are aggregated at hourly levels.

We emphasize that *factual outcome prediction is **not** the task our method is designed for*, nor is it the target of any time-varying meta-learner that adjusts for causal, time-varying confounding. **Predicting observed outcomes requires no adjustment for future treatment sequences** and can be solved by a standard regression on the observed history. Accordingly, the **HA** baseline performs best, exactly as expected: for factual prediction, a simple history adjustment is the statistically optimal approach.

The purpose of this experiment is different: to verify that the **WO** learner, which is primarily designed to estimate *counterfactual* quantities, still produces stable and reasonable predictions when applied to real-world outcome trajectories. It is simply a confirmation that the nuisance components, final stage estimates, and representation layers behave sensibly on real data.

Unlike other propensity-based meta-learners, which exhibit instability or variance inflation even in this predictive setting, the **WO**-learner remains consistently well-behaved across all horizons. This is fully aligned with our theoretical analysis: the weighting mechanism stabilizes estimation in the presence of small propensities, which affects not only causal adjustment but also any learning task involving propensity-based pseudo-outcomes.

| Prediction horizon $\tau$ | **HA** | **IPW** | **DR** | **IVW** | **WO** (ours) |
|:---:|:---:|:---:|:---:|:---:|:---:|
| 1 | $0.708 \pm 0.022$ | $5.627 \pm 8.498$ | $1.601 \pm 1.038$ | $53.200 \pm 98.958$ | $1.077 \pm 0.025$ |
| 2 | $0.774 \pm 0.023$ | $3.419 \pm 4.767$ | $1.255 \pm 0.305$ | $37.408 \pm 36.641$ | $1.079 \pm 0.022$ |
| 3 | $0.822 \pm 0.024$ | $3.729 \pm 5.094$ | $3.978 \pm 5.208$ | $21.151 \pm 40.098$ | $1.087 \pm 0.027$ |
| 4 | $0.869 \pm 0.024$ | $1.426 \pm 0.674$ | $5.556 \pm 8.939$ | $22.128 \pm 32.211$ | $1.099 \pm 0.006$ |

Table 7: Reported are the RMSEs for factual outcome prediction. We emphasize that *factual outcome prediction is **not** the task our WO-learner is tailored for*. Yet, different from the other propensity-based methods, it has very robust performance over all prediction horizons. In line with our theoretical considerations, we conclude that the **HA** learner has the best performance since regressing on the observed history – a simple history-adjustment – is sufficient for factual outcome prediction.

This experiment is **not** intended as evidence of causal validity on real data. *Counterfactual treatment effects over time cannot be validated on observational data* (Pearl, 2009; Poinsot et al., 2025), as at least one of the required potential outcomes is never observed. Therefore, the correct and widely accepted evaluation strategy for time-varying CATE/CAPO estimation consists of synthetic and semi-synthetic experiments where ground-truth effects are known, which is the protocol we follow in our main results in Section 5.

