# OpenReview forum: "Overlap-weighted orthogonal meta-learner for treatment effect estimation over time"
_ICLR.cc/2026/Conference — ICLR 2026 Poster_

### Official Review · Reviewer_hGFP · 2025-10-19

**Soundness:** 3
**Presentation:** 3
**Contribution:** 2
**Rating:** 6
**Confidence:** 5

**Summary:**

This paper tackles the challenge of estimating HTEs in time-varying settings, where treatment sequence overlap decreases with longer horizons. The authors propose a Weighted Orthogonal (WO) meta-learner, which introduces an overlap-weighted, Neyman-orthogonal population risk function to focus learning on regions with sufficient treatment overlap. This design stabilizes estimation variance and enhances robustness to nuisance model misspecification. The method is model-agnostic and demonstrated with transformer and LSTM backbones, showing improved reliability and accuracy over existing time-varying meta-learners.

**Strengths:**

* **Problem Significance:** The paper addresses a highly relevant and challenging problem at the intersection of causal inference and time-series analysis. Estimating HTEs over time is crucial for personalized medicine, and the issue of exponentially decreasing overlap is a core, practical barrier that the paper directly confronts.
* **Methodological Novelty:** The proposed WO-learner is methodologically novel and sound. The key idea of designing a population risk function that explicitly minimizes an *overlap-weighted* oracle risk, while simultaneously enforcing Neyman-orthogonality, is the primary technical contribution.
* **Theoretical Guarantees:** The method is supported by strong theoretical guarantees. The proof that the weighted population risk is Neyman-orthogonal with respect to all nuisance functions (including the newly defined weight functions) is a non-trivial and important result.
* **Experimental Design:** The paper presents a broad set of experiments, including four synthetic datasets designed to isolate specific challenges (low overlap, complex nuisance functions, small sample size) and a semi-synthetic experiment based on real-world MIMIC-III covariates.

**Weaknesses:**

* **Marginal Absolute Improvement:** While the paper reports large *relative* improvements the *absolute* improvements in RMSE are often in the $10^{-2}$ magnitude. The practical significance of such a small absolute gain is questionable, especially given the significant added complexity of estimating the new WO pseudo-outcomes and weight functions.
* **Flawed Real-World Validation:** The validation on real-world data is fundamentally flawed. The authors state this is a "real-world outcome estimation" task, which is a standard *prediction* (supervised learning) task, not a *causal inference* (HTE estimation) task. Demonstrating that the WO-learner is "not worse" than baselines on a predictive task provides **no evidence** of its efficacy for its stated purpose (HTE estimation). This experiment does not support the paper's core claims and raises questions about the method's applicability.
* **Omission of Key Baselines:** The paper omits a crucial and common baseline: a standard DR learner or IPW learner with **propensity score truncation (clipping)**. This is a very common heuristic to handle low-overlap regimes. It is unclear if the WO-learner's complexity is justified over this much simpler, strong baseline.
* **Lack of Intuition for Key Formulas:** The derivation of the key components, particularly the WO pseudo-outcomes $\xi_{t}^{\circ}$ and the weighting term $\rho_{t}^{\circ}$ in Definition 4.2, lacks intuition. They appear to be complex formulas derived to make the orthogonality proof work, rather than being motivated by a clear, first-principles-based explanation. This makes the method difficult to understand and build upon.
* **Unclear Practicality of New Nuisance Function:** The new weight function $\omega_{j}^{\overline{a}}$ (Eq 11) is itself a complex expectation of products of future propensity scores. The practical challenges and stability of estimating this new, non-trivial nuisance function (especially for long horizons $\tau$) are not sufficiently discussed.
* **Missing Discussion of Failure Cases:** The paper does not adequately discuss the potential failure cases or limitations of the WO-learner. When would this method *not* be preferred? How does it behave in very high-dimensional or extremely sparse data settings? A "Limitations" section is missing.
* **Overall Impression and Presentation:** Overall, while the problem is valid and the method is technically correct, the contribution feels incremental. The solution is an orthodox application of established causal inference principles (weighting, orthogonality) to the time-series setting, but the practical gains appear marginal, and the real-world validation is not fit-for-purpose. The work lacks a "surprising" element. Furthermore, the paper's presentation is overly dense, and it notably lacks a "Conclusion" section, which weakens the paper's summary and impact.

**Questions:**

**CATE vs. CAPO Weighting:** Why is the overlap weight for CATE (Eq 10) defined as the *product* of the individual propensity weights ($\omega^{\overline{a}}\omega^{\overline{b}}$)? Why not estimate the CAPO for each arm separately (using the CAPO propensity weight $\omega^{\overline{a}}$) and then simply subtract them? What is the theoretical motivation for this specific product-based weighting for CATE?

---

> ### Author Response · Authors · 2025-11-19
>
> Thank you very much for your positive review of our paper! We improved our paper as a result of your comment, and highlighted all changes in **blue color** in our revised **rebuttal PDF**.
>
>
> ### Responses to Weaknesses:
>
> **>W1: Absolute vs relative improvement**
>
> Thank you. Upon reading your comment, we think there may be a misunderstanding around the role of scaling in synthetic evaluation. Scaling the entire DGP by any constant factor (e.g., multiplying all outcomes by 100) would scale every absolute RMSE and every absolute improvement by the same factor, without changing anything about the difficulty of the estimation problem. Hence, **absolute** RMSE improvements in synthetic settings are therefore **arbitrary and not meaningful** on their own. For this reason, *relative* improvement is preferred and standard for our type of task. In other words, only relative RMSE improvements can accurately capture the difficulty of the tasks and the performance difference between methods.
>
> Across all synthetic and semi-synthetic experiments, **our WO-learner achieves substantial improvements over the existing meta-learners (RA/IPW/DR/IVW), often reducing error by 20–70%**. These gains are **consistent** across horizons, low-overlap regimes, and varying nuisance complexities. Given that (i) RMSE is inherently scale-dependent, (ii) absolute magnitudes in synthetic setups are arbitrary, and (iii) the WO-learner consistently outperforms all baselines, the improvements we report are practically significant. The modest additional computation required for our WO learner is justified by the substantial stability and accuracy gains it delivers.
>
> **>W2: Real-world validation**
>
> Thank you. Your objection precisely reinforces the point we tried to make for the experiments in the supplements you are referring to: factual prediction is *not* a causal evaluation task, and we never present it as such. We apologize if our paper gives a wrong impression. The supplemental experiment is included **only as a sanity check** to verify that our model behaves correctly on real longitudinal data and does not exhibit trivial pathologies. It is explicitly *not* used to support any causal claims.
>
> Importantly, all of the evaluations in the main paper use synthetic and semi-synthetic experiments where ground truth is known (e.g., similar Frauen et al. (2025) and the broader longitudinal meta-learner literature). Our results throughout the main paper follow this standard, valid protocol.
>
> **Action:** We now clarify the above explicitly in our **revised Supplement H**. Throughout our manuscript, we made sure that we spell out clearly that real-world verification of counterfactual estimands is inherently impossible and that semi-synthetic evaluation is therefore the standard and only viable evaluation protocol.

---

> ### Author Response · Authors · 2025-11-19
>
> **> W3: Propensity score clipping**
>
> Thank you. Propensity-score clipping is indeed a commonly used *heuristic* for stabilizing propensity score based estimators, but it is **not** a principled baseline for the problem we study. Truncation directly introduces an irreducible bias: by construction, clipping replaces the true propensities with biased, incorrect values, so it cannot be considered a proper causal estimator of CATE/CAPO. This is well understood in the causal inference literature: clipping trades variance reduction for estimand distortion, which makes it inappropriate to compare against methods that preserve exact identification.
>
> More importantly, clipping is rather an implementation hack applicable to all meta-learners and does not address the statistical problem our WO-learner solves. The WO-learner constructs a *variance-stabilized pseudo-outcome* whose conditional variance remains bounded even under limited overlap while **preserving the correct causal estimand** (see our **new Theorem in our new Supplement C.4**). **In contrast, clipping does not accomplish this**: it simply caps extreme propensity ratios, and hence leaves the pseudo-outcome biased whenever truncation occurs. In fact, clipping can be applied on top of **any** meta-learner, including our own WO-learner, because it is **not** a causal adjustment strategy but a generic preprocessing hack.
>
> Crucially, there is ***no principled way to validate how much clipping is needed***: the appropriate truncation level cannot be inferred from the data, and the bias introduced by clipping is unmeasurable in observational causal inference, since **the counterfactual outcomes needed to calibrate this tradeoff are never observed**. For this reason, clipping introduces an **uncontrolled and unassessable bias–variance tradeoff**. We therefore do not use clipping as a hyperparameter for **any** method (neither our own nor the baselines) because it would confound the comparison by injecting arbitrary bias that is not grounded in the underlying estimand.
>
> To ensure **fair comparisons**, all baselines in our experiments share the same backbone architecture and target the same causal quantity. Including heuristic clipped variants of IPW/DR or our own learner would not constitute an apples-to-apples comparison and would obscure the methodological contribution of the WO-learner, which provides a principled, bias-free solution to instability in low-overlap regimes.
>
> **Action:** We highlight in our **revised related work Section 2** that, although propensity score clipping can reduce variance, it introduces uncontrollable bias. In contrast, our WO-learner offers a principled way to reduce estimation variance in low-overlap regimes while preserving the correct estimand. We further provide a **new Theorem in our new Supplement C.4** that shows how our weighting scheme uniformly bounds variance in low-overlap regimes while maintaining the correct estimand.
>
> **>W4: Key formulas:**
>
> Thank you. Our WO pseudo-outcomes and weights in Definition 4.2 are *not* ad-hoc expressions introduced to make the proofs work. We carefully derived them from first principles: we begin with the weighted oracle population loss (motivated by the need to stabilize estimation under limited overlap), and we then apply orthogonalization to this weighted objective to obtain a population risk whose sensitivity to nuisance estimation error is minimized. Therefore, the resulting pseudo-outcomes inherit part of the structure of classical doubly-robust scores, part from the orthogonalization of the overlap weights themselves, and part from a combined expression. This combination is the consequence of the derivation, and there is no simpler form that achieves both identification and Neyman-orthogonality under weighting.
>
> **Action:** We have clarified this intuition in our **revised Section 4**. We arranged our paper so that we present our weights first before diving into the theoretical analysis. If the reviewer has concrete suggestions for revising the presentation or notation without compromising the statistical properties of the estimator, we would be happy to incorporate them.

---

> > ### Comment · Reviewer_hGFP · 2025-11-21
> >
> > Thanks for the clear explanation about why "ps clipping" isn’t a proper baseline here. I brought it up because it’s an intuitive idea and can seem useful at first glance. But I agree with the authors that it’s not a principled approach, and choosing a specific clipping level is neither easy nor well-justified.

---

> ### Author Response · Authors · 2025-11-19
>
> **> W5: Practicality of nuisance function**
>
> Thank you. We agree that the overlap weight in Eq. (11) is a nontrivial quantity. This is inherent to the problem, because stabilizing estimation in long-horizon, low-overlap settings necessarily requires accounting for the cumulative effect of future propensities. Importantly, however, the complexity of the weight function does *not* translate into practical instability.
>
>
>
> * First, our new theoretical results (**new Theorem in our new Supplement  C.4**) show that weighting by this term yields a pseudo-outcome with **uniformly bounded variance**, even as overlap deteriorates. In other words, the weight is precisely what *stabilizes* the estimator in practice.
> * Second, the WO score is **Neyman-orthogonal with respect to estimation error in the weight function**, so inaccuracies in estimating the nuisance have only second-order impact on the final causal estimate. This orthogonality property is *exactly what prevents the weight from becoming a bottleneck*, even for long horizons.
> * *In practice, we do not observe instability tied to estimating this weight, consistent with the theory, and the WO-learner remains reliable even for multi-step horizons.*
>
> **Action:** We have added a clarification in our **revised Section 4** stating that (i) our weights crucially stabilize variance of the pseudo outcomes by bounding it uniformly (see our **new Theorem** in our **new Supplement C.4**), and (ii) orthogonality guarantees robustness to imperfect nuisance estimation.
>
> **>W6: Limitations / failure cases**
>
> Thank you. Below, we summarize practical considerations when to choose our WO-learner vs. when to choose simpler meta-learners (we also added practical recommendations to our **new Supplement A**). Thereby, we hope to support practitioners adopting causal ML in longitudinal settings and help them understand boundary conditions and navigate the literature:
>
> 1. **When high-quality nuisance functions are known or exceptionally accurate.**
>
> In this edge case, the classical DR learner benefits from its double robustness property: if *either* the outcome model or the propensity model is correctly specified, the DR estimator remains unbiased. Our WO-learner is Neyman-orthogonal but not doubly robust, so in the *rare scenario where practitioners have exceptionally accurate nuisance models*, DR may perform similarly or slightly better. *However, in realistic longitudinal observational data, nuisance estimation is high-dimensional and difficult, which makes our WO-learner the more robust and stable choice.*
>
> 2. **When overlap is uniformly large and prediction horizons are small.**
>
>    If propensities are far from the boundaries and do not degrade too quickly over time, the benefits of overlap weighting only occur for larger prediction horizons, i.e., the exponential rate at which overlap reduces is slightly lower. Hence, for small prediction horizons and very balanced overlap, this is a benign regime where simpler methods may be sufficient. *Our experiments show, however, that realistic longitudinal datasets rarely exhibit uniform overlap, and hence the significant empirical advantage of WO.*
>
> Beyond these specific edge cases, we are not aware of meaningful practical conditions under which our WO-learner would not be favored. In high-dimensional or sparse-data regimes, WO remains well behaved: (i) orthogonality makes the method robust to nuisance misspecification, and (ii) our theoretical results guarantee uniformly bounded pseudo-outcome variance even when propensities become small. This is reflected consistently in our empirical results.
>
> **Action:** In our **new Supplement A**, we have added a discussion of practical considerations when to use our WO-learner vs. other meta-learners. We have further added a **new Limitations paragraph** at the end of our main paper.

---

> ### Author Response · Authors · 2025-11-19
>
> **>W7: Overall impression**
>
> Thank you. We respectfully disagree with the characterization that our contribution is incremental. The central difficulty in estimating heterogeneous treatment effects over time is the rapid collapse of overlap as the horizon increases. Existing meta-learners are not designed to address this problem and therefore break down as sequence length grows. Our work tackles this important challenge directly by introducing (i) a **novel** weighted population objective tailored for limited-overlap regimes, and (ii) a **novel** orthogonalized population risk that is insensitive to nuisance errors, and preserves identification of the target estimand.
>
> This combination is not present in prior work and yields a* provably better-behaved pseudo-outcomes: the variance of our pseudo-outcomes is uniformly bounded even when propensities approach zero* (see our **new Theorem** in our **new Supplement C.4**). Thereby, we demonstrate the effectiveness of our proposed weighting strategy in low-overlap regimes.
>
> Practically, these improvements are far from marginal: across all experiments, the WO-learner delivers substantial error reductions (often 30–70%), and remains stable in exactly the regimes where existing meta-learners become unusable. These are the settings that matter for real-world sequential decision-making, where downstream policies depend on reliable long-horizon effect estimates.
>
> Regarding real-world evaluation, as explained earlier, we kindly refer to our answer **W2**.
>
> In summary, our paper addresses a fundamental and widely recognized failure mode of existing time-varying meta-learners. In particular, we provide a solution supported by new theory and demonstrate empirical gains in the regimes that matter for longitudinal causal inference. We therefore view the contribution as both meaningful and impactful.
>
> Finally, we appreciate the suggestion regarding our conclusion!
>
> **Action:** We further provide a **new Theorem in our new Supplement C.4** that shows how our weighting scheme uniformly bounds variance of our pseudo-outcomes in low-overlap regimes. We **extended our previous Conclusion paragraph** at the end of our main paper.

---

> > ### Comment · Reviewer_hGFP · 2025-11-21
> >
> > Sorry to comment that the contribution feels incremental, but the overall structure of the paper comes across as fairly standard, essentially an “orthogonalize everything style” approach. The analytical procedure follows quite standard patterns, and while the results are valid and technically sound, there isn’t a particularly “surprising” or novel element.

---

> ### Author Response · Authors · 2025-11-19
>
> ### Responses to Questions:
>
> Thank you for your interesting question!
>
> We agree this is an important conceptual point and clarify both (i) why a “CATE = difference of two CAPOs” strategy is **not** sufficient, and (ii) why the CATE overlap weight in Eq. (10) naturally appears as a **product** of CAPO weights in our WO-learner. We answer your questions in two steps below.
>
> **Ad (i): Why learn the CATE directly and not two CAPOs separately?**
>
> Treating CATE as “CAPO(1) – CAPO(0)” and learning the two CAPOs separately is exactly the situation that the CATE meta-learner literature warns against (Curth, 2021). When the response function is complex, each treatment-arm-specific CAPO model is a high-dimensional regression problem; taking their difference inherits the **full estimation error from *both*** treatment arms. By contrast, the treatment effect function itself can be much simpler and more stable to learn directly. This is *precisely the motivation behind CATE meta-learners in general* and is explicitly discussed in Frauen et al. (2025) and the broader causal inference literature: directly targeting the treatment effect avoids the error amplification that occurs when subtracting two large, noisy nuisance estimates. Our CATE WO-learner is designed to avoid this (like the other CATE meta learners): we construct a pseudo-outcome and loss that **directly target the CATE**, not the two CAPOs separately.
>
> **Ad (ii) Why the specific form (“product”) of our overlap weights?**
>
> Given the above, we need a **single weighted population risk** that (a) has CATE as its target, and (b) emphasizes the regions where CATE is actually identifiable and relevant, i.e., where *both* treatment sequences have sufficient support. This leads to an overlap weight that is large exactly when **both** CAPO propensities are large and small when either treatment arm is unlikely. The *product form in Eq. (10) does exactly that*: it upweights samples that are simultaneously informative for both potential paths and downweights those with poor joint overlap. Starting from this CATE-focused weighted loss and then orthogonalizing it with respect to the nuisances yields the WO pseudo-outcomes and guarantees Neyman-orthogonality. In other words, the product-based CATE overlap weight is not an arbitrary choice, but the mathematically natural consequence of (i) directly targeting CATE rather than two separate CAPOs, and (ii) concentrating the objective on the joint overlap region where the CATE is identified and can be estimated reliably.
>
> In short:
>
> 1. CATE directly is the right target (difference-of-CAPOs is statistically worse).
>
>
> 2. As we commit to *direct* CATE, we need a single weighted risk.
>
>
> 3. That risk must upweight **joint** overlap, hence the product weight.
>
>
> 4. Orthogonalizing that risk yields our pseudo-outcomes.
>
> Thank you for your thoughtful review. If you have any further questions, we are happy  to clarify them!
>
> ____
>
> Alicia Curth and Mihaela van der Schaar. Nonparametric estimation of heterogeneous treatment effects: From theory to learning algorithms. In AISTATS, 2021.

---

> > ### Comment · Reviewer_hGFP · 2025-11-21
> >
> > Overall, this is a technically solid and analytically rigorous paper that addresses the low-overlap problem in time-series HTE estimation. The authors have satisfactorily resolved my concerns in their rebuttal, so I am increasing my overall score from 6 to 8, as well as raising the contribution score from 2 to 3 and the soundness score from 3 to 4.

---

> > > ### Author Response · Authors · 2025-11-23
> > >
> > > Thank you very much for your helpful and positive review! We will incorporate all changes into the final version of our paper.

---

### Official Review · Reviewer_jn4J · 2025-10-27

**Soundness:** 3
**Presentation:** 3
**Contribution:** 3
**Rating:** 8
**Confidence:** 3

**Summary:**

This paper proposes a new framework, Overlap-Weighted Orthogonal Learning (OWL), for estimating heterogeneous treatment effects in time varying settings where covariate overlap between treatment groups decreases exponentially with time horizon. The method incorporates the a propensity/overlap weights that up-weights samples with a higher probability to having the treatment sequence of interest. Moreover, the proposed estimator is Neyman orthogonal which makes the estimator insensitive to nuisance estimation error.

**Strengths:**

- Clear motivation and the proposed estimator addresses the problem of insufficient overlap.
- The estimator is flexible and is robust to nuisance errors as well.
- Performed comprehensive experiments that addresses different settings. Results demonstrates the effectiveness of the proposed method at addressing the different issues discussed in the paper.

**Weaknesses:**

- While the paper showed that the proposed population risk is Neyman orthogonal, papers on mete-learners usually also provide error analysis and show that the error terms from the nuisance functions are higher order, which is not presented in the paper.
- The main innovation is to address the problem of limited overlap, so it would be nice to have some theorems that showcase how this estimator have better behavior (e.g. variance) in those regimes.

**Questions:**

- Do we expect the performance of the proposed estimator to deteriorate more that the DR-learner when propensity is hard to learn, as the weights also rely on the propensity estimates?

---

> ### Author Response · Authors · 2025-11-19
>
> Thank you very much for your positive review of our paper! We added the new theorem and the new experiments (see below) in **blue color** to our revised **rebuttal PDF**.
>
>
> ### Responses to Weaknesses:
>
> **>W1: Error analysis**
>
> Thank you for your suggestion! Analyzing how errors in the nuisance functions propagate to the final stage estimate is exactly the intention behind our synthetic data experiment on $\mathcal{D}^N$, where we reduce the sample size for the nuisance functions and evaluate the error on the final stage estimate. Here, *we can clearly see the benefits of Neyman-orthogonality of our learner*.
>
> Upon reading your comment, we realized that providing an additional sensitivity experiment is a great idea!
>
> **Action:** We added a **new experiment **in our **new Supplement G,** where we analyze how errors in a mis-specified nuisance function propagate to the final stage estimate.
>
> **>W2: Theoretical support for lower variance**
>
> Thank you for this suggestion! We think it is a great idea to give a theoretical justification why our overlap weights reduce variance in low-overlap regimes.
>
> **Action:** We provide a **new Theorem** in our **new Supplement C.4**. Therein, we show that for decreasing overlap, the variance of the DR pseudo-outcomes explodes, whereas the pseudo-outcomes in overlap-weighted population risk have bounded variance. Thereby,* we demonstrate the effectiveness of our proposed weighting strategy in low-overlap regimes*.
>
>
> ### Responses to Questions:
>
> **>Q1: Complex propensity scores**
>
> Thank you for your question! Generally, we expect the performance of our WO-learner to deteriorate as propensities become more complex. However, our method is Neyman-orthogonal with respect to **all** nuisances, including propensities **and** weights. Hence, errors in the estimated weights also only propagate as lower-order errors to the final stage estimate. In simple words, this should make our method highly robust even under complex propensity scores.
>
> Empirically, we found that our method remains very robust under complex propensities: this is what we report in our synthetic experiment **(2) Complex propensity score** $\mathcal{D}^\pi$. Here, we simulate a very complex treatment propensity (details in Supplement D.1.) and report the errors for increasing prediction windows. => The CATE error increases for our method, but different to the other propensity-score based baselines, at a lower rate. This confirms empirically our theory: **our overlap-weights are effective even when nuisance errors are relatively large (e.g., due to complex propensities)**.
>
> Thank you for your thoughtful review. If there is anything else that you would like us to clarify, please let us know!

---

### Official Review · Reviewer_gNhd · 2025-10-28

**Soundness:** 4
**Presentation:** 3
**Contribution:** 3
**Rating:** 8
**Confidence:** 3

**Summary:**

The paper proposes an over-weighted meta-learner to estimate HTE in a very challenging time-varying scenario with a treatment sequence scenario. It also develops a new risk esimation to reduce the effects of first order bias in the estimation. The paper provides detailed theorectical analysis. The empirical results show that the proposed method achieve the best performance compared with other adjustment method.

**Strengths:**

- Compared with other meta-learners in this field, the study proposes a novel method to address the challenges of other learners in the low overlap scenario, which is common in the time varying treatment assignment strategy.
- The overlap weighted orthogonal
- The study has conducted both theorectical and empirical analysis to demonstrate the advantages of their proposed method.
- The results demonstrate dramatical improvement of the performance in Table 2. And the performance is uniformly the best over all the sample sizes.

**Weaknesses:**

- Can you please revise the writing of the problem setup? Maybe provide some examples to illustrate what the treatment looks like. Based on my understanding of your problem, the two sequences should be a list of 1 or 0, correct? i.e. $a = [1, 0, 0, 1], b = [0, 1, 1, 1]$
- One very relevant study as mentioned in the paper is the IVW method. As stated in the IVW paper, the framework is a composition of IVW and DR learner [1]. The paper provides very brief description in line 116-125. I wonder here is the IVW refers to the same model as the IVW-DR learner in the original paper or just the IVW only.
- Another concern about the time-varying treatment strategy is that, the proposed method only considers confounders happen before $t$. But in the real-world setting, the propensity scores are usually depend on what happened during $t:t+\tau$. I know it makes the problem much harder, but it is very realistic in healthcare domain.

[1] Frauen, D., Hess, K., & Feuerriegel, S. Model-agnostic meta-learners for estimating heterogeneous treatment effects over time. In The Thirteenth International Conference on Learning Representations.

**Questions:**

- In table 4, why the IVW performance get dramatically incresed when the prediction horizon increases? In the paper [1], the performance looks normal when $\tau=2,3$.

---

> ### Author Response · Authors · 2025-11-19
>
> Thank you very much for your positive review of our paper! We improved our paper as a result of your comment, and highlighted all changes in **blue color** in our revised **rebuttal PDF**.
>
>
> ### Responses to Weaknesses:
>
> **>W1: Revision of problem setup**
>
> Thank you for your suggestion. Thank you for pointing this out. You are exactly right, the treatment sequences are vectors of 0’s and 1’s. We made a notational error in our problem setup: the time-specific treatments are supposed to be $A_t \in \{0,1\}$, and the full treatment sequences $A_{t+\tau}$ vectors of the form $(0,1,0,...1)$ etc. We apologize for this oversight.
>
> **Action:** We revised the problem setup in Section 3 of our paper. Further, we highlight Figure 1 as an example of the treatments of interest.
>
> **>W2: IVW / IVW-DR learner**
>
> Thank you for your question! Indeed, we are referring to the IVW-DR learner by Frauen et al. (2025). We deliberately referred to their method as IVW learner since their learner is **not** doubly robust. Specifically, their learner relies on estimating inverse variance weights, which rely on estimated propensity scores. While the pseudo outcomes they use are, indeed, doubly robust, their loss is **not** double robust. Error from the estimated propensity scores *propagate as first-order biases to the estimated inverse-variance weights*, and therefore as first-order biases to the loss. The key issue is that their loss is **not** orthogonalized. The authors simply multiply their weights to the loss, and perform no first-order error correction for the resulting nuisance function error propagation. Hence, their method is **not** doubly robust.
>
> **Action:** We clarify in our **revised related work Section 2** why we are referring to the IVW-DR learner as IVW learner => to spell out clearly their loss is not doubly robust.
>
> **>W3: Confounders after time t**
>
> Thank you for this question! You are right: the key issue in time-varying treatment effect estimation lies in correctly adjusting for confounding that arises **after** the initial treatment time $t$. This type of confounding is typically referred to as “**time-varying confounding**”.
>
> Importantly: not properly adjusting for this source of confounding leads to infinite-data bias, as we highlight in **Figure 2 and EQ (3).** Instead, adjustments for time-varying confounding are required, as performed by the baseline meta-learners IPW, DR, IVW, and RA (**not** the HA / history adjustment). We **also adjust for this type of confounding** in our paper: **This is exactly what we guarantee in our Corollary 4.4 (Time–varying adjustment)**.
>
> To be concrete, we show in Theorem 4.3 that our orthogonalized population risk minimizes the weighted oracle risk. Since the weighted oracle risk attains its minimum for $g = \mu_t^{a,b}$, i.e., the CATE (or similarly, the CAPO), we ensure that our orthogonalized risk is minimized if and only if $g=\mu_t^{a,b}$ (and **not** some biased history adjustment term), i.e., only for the **correctly adjusted, true** CATE.
>
>
> ### Responses to Questions:
>
> **>Q1: Performance of baseline in Table 4**
>
> Thank you for your question. In Table 4, we report the performance of all methods based on **semi-synthetic** data (details on the data-generating process are in Supplement D.2). Here, we simulate treatments and outcomes based on real-world covariates. This allows us to simulate a ground-truth CATE, which is unobserved in real-world data settings, and hence to correctly validate all methods.
>
> The authors of the inverse-variance weighted learner did **not** provide results on semi-synthetic data, only on purely synthetic data and real-world data for factual outcome prediction. Empirically, we found that the IVW-DR learner (here: IVW-learner) performed badly for very high-dimensional covariate spaces, as shown in our semi-synthetic experiments. Here, the non-orthogonal estimated inverse variance weights quickly blew up, and estimation errors propagated heavily.
>
> Thank you again for your thoughtful review. If there is anything else that you would like us to clarify, please let us know!

---

> > ### Comment · Reviewer_gNhd · 2025-11-23
> >
> > Thank you very much for your detailed response. The points are now clear to me. So I have improved my confidence score from 3 to 4.
> >
> > Overall it is a good paper with both solid theory and experiments. So I have improved the overall score to 10.

---

> > > ### Author Response · Authors · 2025-11-23
> > >
> > > Thank you very much for your helpful and positive review! We will incorporate all changes in the final version of our paper.

---

### Official Review · Reviewer_u6pF · 2025-11-01

**Soundness:** 3
**Presentation:** 2
**Contribution:** 3
**Rating:** 6
**Confidence:** 3

**Summary:**

The paper proposes an overlap-weighted orthogonal (WO) meta-learner to estimate time-varying heterogeneous treatment effects (HTEs) under low-overlap conditions. The framework aims to stabilize inverse-propensity-based estimators by re-weighting high-overlap regions.

**Strengths:**

- The paper proposes a model-agnostic framework for adjusting time-varying confounders.
- The paper provides a theoretical foundation for orthogonality.
- An implementation code is available for review.

**Weaknesses:**

- The claim that weighting by overlap improves estimator stability is weakly justified. By emphasizing high-overlap regions, the learner avoids extreme inverse-propensity weights but can ignore potentially outcome-informative regions. Is there a trade-off between propensity and outcome predictions?
- The motivation behind the study problem (treatment effect estimation over time) needs to be further clarified, particularly in relation to its real-world applicability. It is not evident whether the problem is testable, whether the authors have empirically validated it in real applications, or whether the underlying assumptions commonly hold in practical settings.
- The experimental validation is primarily based on synthetic data. Even in the semi-synthetic MIMIC-III experiment, both treatments and outcomes are simulated, so there is no demonstration of applicability to real observational data. The study does not evaluate whether estimating CATE over time provides practical benefits or is actually necessary or useful in real-world contexts. The study shows performance under a controlled scenario rather than demonstrating practical utility.
- The authors explicitly note that “factual outcome prediction is not the task our method is tailored for,” yet give no alternative validation criterion for real-world use. How would it be verified in practice? The authors should add a discussion and limitations on this point.
- The baselines are narrow, limited to RA, IPW, DR, and IVW  from [1]. The paper omits several relevant baselines for CATE over time, e.g., G-Net [2] and G-Transformer [3].

[1] Frauen, D., Hess, K., & Feuerriegel, S. Model-agnostic meta-learners for estimating heterogeneous treatment effects over time. In The Thirteenth International Conference on Learning Representations.

[2] Li, R., Shahn, Z., Li, J., Lu, M., Chakraborty, P., Sow, D., ... & Lehman, L. W. H. (2020). G-Net: a deep learning approach to G-computation for counterfactual outcome prediction under dynamic treatment regimes. arXiv preprint arXiv:2003.10551.

[3] Hess, Konstantin, Dennis Frauen, Valentyn Melnychuk, and Stefan Feuerriegel. "G-Transformer for Conditional Average Potential Outcome Estimation over Time." CoRR (2024).

**Questions:**

- The paper claims to estimate HTE but reports only CATE in the experiments. There is no discussion of heterogeneity across subgroups or covariate-dependent variation in effects. It may mislead readers about what “HTE” means in this context.
- Please see weaknesses.

---

> ### Author Response · Authors · 2025-11-19
>
> Thank you very much for your positive review of our paper! We improved our paper as a result of your comment, and highlighted all changes in **blue color** in our revised **rebuttal PDF**.
>
>
>
>
> ### Responses to Weaknesses:
>
> **>W1: Theoretical justification for overlap-weights**
>
> Thank you for your suggestion. We think it is a great idea to give a theoretical justification why our overlap weights reduce variance in low-overlap regimes. Further, we are not aware of any trade-offs between propensity and outcome predictions.
>
> **Action:** We provide a **new Theorem** in our **new Supplement C.4**. Therein, we show mathematically that, *for decreasing overlap, the variance of the DR pseudo-outcomes explodes, whereas pseudo-outcomes in our overlap-weighted population risk have uniformly bounded variance*. Thereby, we demonstrate the effectiveness of our proposed weighting strategy in low-overlap regimes.
>
> **>W2: Motivation for treatment effect estimation over time**
>
> Thank you.
>
> **Motivation behind the problem:**
>
> Treatment effect estimation over time is **one of the foundational problems in causal inference** and dates back to classical work in epidemiology and classical statistics (Bang & Robins, 2005; Robins, 1986; 1994; 1999; Robins et al., 2000; Robins & Hernán, 2009; van der Laan & Gruber, 2012; van der Laan & Rose, 2018). Whenever interventions are delivered repeatedly (e.g., in medicine, public health guidelines, education, recommendation systems) covariates evolve in response to past treatments. This introduces time-varying confounding. Our setting is the **standard formulation for sequential decision-making** in real-world systems.
>
> The ML literature has recently built on these ideas to estimate individualized effects in longitudinal data (Bica et al., 2020; Hess & Feuerriegel, 2025; Li et al., 2021; Lim et al., 2018; Melnychuk et al., 2022; Seedat et al., 2022; Wang et al., 2025). These works are neural instantiations of meta-learners, and are designed to adjust for time-varying confounding. They **all** rely on the same assumptions that we also use: consistency, sequential ignorability, and positivity. These are **not** special assumptions of our framework; they are the **standard identifying conditions across decades of longitudinal causal inference and all modern ML extensions**.
>
> Frauen et al. (2025) recently formalized meta-learners for the time-varying setting, and our work directly extends this line of research. Specifically, we address a well-known practical challenge: the collapsing overlap of treatment sequences as the prediction horizon increases. This problem is documented throughout the causal literature but is not addressed by any existing method.
>
> **Identifying assumptions:**
>
> - **Consistency:** This is the standard assumption that the observed outcome under the actual treatment history corresponds to the relevant potential outcome, which holds whenever treatments are well-defined and consistently recorded.
>
> - **Sequential Ignorability:** This assumption requires that the recorded covariate history captures the information driving treatment decisions; it is the standard identifying condition used in all longitudinal causal inference methods and is generally plausible in modern datasets with rich time-varying state information (e.g., datasets in oncology or intensive care units as ours typically capture all relevant time-varying information).
>
> - **Positivity:** This requires only that each treatment option occurs with nonzero probability along observed covariate histories. This condition is typically satisfied when the sample size is large enough and the policy is randomized. Our WO-learner directly addresses the setting of low overlap, i..e, when the propensity of at least one treatment sequence of interest is low.
>
> **Testability:**
>
> Regarding testability, individualized treatment effects (both static or time-varying) are **counterfactual** quantities and therefore **cannot** be validated directly on observational real-world data. This is a **fundamental** challenge in causal inference (Bang & Robins, 2005; Robins, 1986; 1994; 1999; Robins et al., 2000; Robins & Hernán, 2009; van der Laan & Gruber, 2012; van der Laan & Rose, 2018) and **not** specific to our method. Hence, the assumptions are typically ensured by design, that is, choosing a setting where the assumptions can be confirmed with domain knowledge. Further, the evaluation protocol we adopt ((semi-)synthetic data with known ground truth) is the **standard** and is used across all prior work. Our semi-synthetic experiments based on MIMIC-III further demonstrate that our method behaves well under realistic, high-dimensional covariates.
>
> In short, the problem formulation is broadly applicable, the assumptions are those used throughout the causal iterature, and evaluation follows the established norms.
>
> **Action:** We add a **new discussion** on the validity of the assumptions in a **new Supplement B**.

---

> ### Author Response · Authors · 2025-11-19
>
> **> W3: Experimental validation**
>
> Thank you.
>
> Evaluating time-varying CATE estimators on purely observational real-world data is challenging, because the estimand is a **counterfactual** object: for each subject, we observe only the outcome under the realized treatment sequence and **never** the outcomes under the alternative treatment histories required for CATE computation.
>
> Our experimental setup follows best practice (Lim et al., 2018; Bica et al., 2020; Seedat et al., 2022; Frauen et al., 2025) and is the **standard and only valid** approach for this problem class: (i) controlled synthetic data where the true causal effects are known, and (ii) semi-synthetic data grounded in real covariate trajectories where counterfactual outcomes are simulated solely to expose the true effect function for benchmarking. This is exactly the protocol used by every prior neural method for longitudinal CATE estimation. Evaluating the performance of any CATE learner would require access to counterfactual medical outcomes, which do not and cannot exist in observational datasets. Still, **we demonstrate the applicability of our method using real-world observational data in Supplement H**.
>
> Regarding practical utility, the need for time-varying treatment effect estimation is well-established. Nearly all real-world intervention settings (e.g., dosage adjustment, dynamic treatment regimes, policy evaluation, behavioral interventions, recommendation systems) depend on patient- or user-specific histories and are sequential decision-making problems. Ignoring temporal structure or collapsing longitudinal dynamics into a static setting is well known to yield biased or misleading conclusions. Our contribution therefore addresses a problem that is not only relevant but unavoidable in sequential decision-making. This is also confirmed by the large performance gains in our experiments.
>
> **>W4: Factual outcome prediction**
>
> Thank you.
>
> Indeed, factual outcome prediction is not the target task of our method. Instead, we seek to estimate the CATE / CAPO, which are causal, **counterfactual** quantities, that cannot be observed in real-world data and therefore cannot be evaluated on real-world data (Poinsot, 2025). This challenge is often referred to as “fundamental problem of causal inference”; see the discussion by the Turing award winner Judea pearl (Pearl, 2009) and hence performance is typically benchmarked not with traditional factual outcomes (e.g. as in supervised machine learning) but via simulations (e.g., as in the case reinforcement learning). Such simulations are the **common standard** (Lim et al., 2018; Bica et al., 2020; Melnychuk et al., 2022, Seedat et al., 2022; Frauen et al., 2025), because it allows us to evaluate the predictions of an outcome under the factual treatment vs the outcome under a counterfactual treatment.
>
> Still, we included an experiment using real-world data in Supplement H: it serves as a *sanity check* to demonstrate that the model architecture we use is capable of fitting real-world outcome trajectories when evaluated on observed (factual) data. This type of auxiliary validation is common in causal inference: although it does not evaluate counterfactual accuracy, it confirms that the nuisance components and representation layers behave sensibly on real data. Importantly, it is **not** presented as evidence of causal validity, nor as a substitute for counterfactual evaluation.
>
> **Action:** We now clarify the above explicitly in our **revised Supplement H**. We also **add a limitation paragraph** at the end of our main paper.

---

> ### Author Response · Authors · 2025-11-19
>
> **> W5: Baselines**
>
> Thank you. The baselines we include (HA, RA, IPW, DR, and IVW) are not narrow, but represent the **entire existing family** of meta-learners for time-varying treatment effect estimation. These model-agnostic learners correspond exactly to the standard adjustment strategies in longitudinal causal inference (i.e., regression adjustment, inverse propensity weighting, doubly-robust estimation). Hence, any of the methods that you cited (e.g., the G-transformer) is just a specific instantiation of the meta-learners. For example, any method built on G-computation, (e.g., G-Transformer), is simply a *model-specific instantiation* of the regression-adjustment (RA) meta-learner. Since our RA baseline uses the **same neural backbone** as all other meta-learners in our comparison, these model-based learners are already represented in our evaluation.
>
> Our goal in this paper is to **isolate the effect of the meta-learner itself**. That is, we want to **analyze the statistical adjustment strategy**, and **not** compare arbitrary neural modeling choices, or architectures.
>
> Introducing model-based baselines such as G-Net or G-Transformer would conflate architectural differences with causal identification strategies. Again, note that they are **instantiations** of the RA adjustment. Different architectures would simply make the comparison fundamentally unfair. To avoid this, all meta-learners in our experiments share the same neural backbones for the nuisances and final stage estimators, and differ **only** in their adjustment strategy. This ensures a clean comparison that **isolates the effects of the different meta-learners.**
>
> **Action:** We highlight in our **revised Section 4** that **our baselines are exhaustive**, and that **our comparison is fair**.

---

> ### Author Response · Authors · 2025-11-19
>
> ### Responses to Questions:
>
> **>Q1: HTE**
>
> Thank you. The term *heterogeneous treatment effects (HTE)* follows the terminology in the causal inference literature (e.g., Frauen et al. (2025)). Here, the term “HTE” is used as the umbrella concept for both the CATE and the CAPO. In other words, HTE refers simply to *covariate-dependent treatment effects*, i.e., the mapping $\bar{H}_t \to \mu_t^{a}(\bar{H}_t)$ or $\bar{H}_t \to \mu_t^{a,b}(\bar{H}_t)$, and not necessarily to subgroup contrasts or pre-specified partitions of the covariate space (this can be done as a pre-processing step).
>
> Under this standard definition, estimating CATE **is** estimating HTE: CATE is the function-valued quantity that captures individualized or covariate-dependent treatment effects. Similarly, CAPO represents covariate-dependent potential outcomes, which can likewise be viewed as HTE. Our experiments therefore directly evaluate the quantities that constitute HTE. On top of that, we develop our theory not only for CATE, but also for CAPO. To avoid the distinction throughout the paper, we follow Frauen et al. (2025) and **refer to both as HTE**.
>
> **Action:** We double-checked that we correctly define the term “HTE” in our paper as in Frauen et al. (2025) => see our **revised Section 1**.
>
> ____
>
> Ioana Bica, Ahmed M. Alaa, James Jordon, and Mihaela van der Schaar. Estimating counterfactual treatment outcomes over time through adversarially balanced representations. In ICLR, 2020.
>
> Dennis Frauen, Konstantin Hess, and Stefan Feuerriegel. Model-agnostic meta-learners for estimating heterogeneous treatment effects over time. In ICLR, 2025.
>
> Bryan Lim, Ahmed M. Alaa, and Mihaela van der Schaar. Forecasting treatment responses over time using recurrent marginal structural networks. In NeurIPS, 2018.
>
> Valentyn Melnychuk, Dennis Frauen, and Stefan Feuerriegel. Causal transformer for estimating counterfactual outcomes. In ICML, 2022.
>
> Judea Pearl. Causality. Cambridge University Press, New York City, 2009. ISBN 9780521895606.
>
> Audrey Poinsot, Panayiotis Panayiotou, Alessandro Leite, Nicolas Chesneau, Özgür Simsek, and Marc Schoenauer. Position: Causal machine learning requires rigorous synthetic experiments for broader adoption. In ICML, 2025.
>
> James M. Robins. A new approach to causal inference in mortality studies with a sustained exposure period: Application to control of the healthy worker survivor effect. Mathematical Modelling, 7:1393–1512, 1986.
>
> James M. Robins. Correcting for non-compliance in randomized trials using structural nested mean models. Communications in Statistics - Theory and Methods, 23(8):2379–2412, 1994.
>
> James M. Robins. Robust estimation in sequentially ignorable missing data and causal inference models. Proceedings of the American Statistical Association on Bayesian Statistical Science, pp. 6–10, 1999.
>
> James M. Robins and Miguel A. Hernan. Estimation of the causal effects of time-varying exposures. Chapman & Hall/CRC handbooks of modern statistical methods. CRC Press, Boca Raton, 2009. ISBN 9781584886587.
>
> James M. Robins, Miguel A. Hernan, and Babette Brumback. Marginal structural models and causal inference in epidemiology. Epidemiology, 11(5):550–560, 2000.
>
> Nabeel Seedat, Fergus Imrie, Alexis Bellot, Zhaozhi Qian, and Mihaela van der Schaar. Continuous-time modeling of counterfactual outcomes using neural controlled differential equations. In ICML, 2022.
>
> Mark J. van der Laan and Susan Gruber. Targeted minimum loss based estimation of causal effects of multiple time point interventions. The International Journal of Biostatistics, 8(1), 2012.
>
> Mark J. van der Laan and Sherri Rose. Targeted learning in data science. Springer, Cham, 2018. ISBN 978-3-319-65303-7.

---

> ### Author Response · Authors · 2025-11-28
>
> Dear reviewer u6pF,
>
> We thank you for your helpful and positive feedback on our submission.
>
> We hope that our rebuttal could clarify your remaining question. Otherwise, we are happy to provide further insights!
>
> If you feel comfortable with it, we would highly appreciate it if you would consider increasing your score.

---

### Author Response · Authors · 2025-11-19
**Response to all reviewers**

We thank the reviewers for the comprehensive and helpful feedback on our work.

To improve our work, we added the following improvements to our revised paper (highlighted in $\color{blue}{\text{blue}}$):



* **New theorem:** We have added a new theorem in our **new Supplement C.4**. Therein, we show that *our overlap weights uniformly bound the variance of our pseudo-outcomes in low-overlap regimes*. Thereby, we demonstrate the effectiveness of our proposed weighting strategy in low-overlap regimes.
* **New experiment:** We provide an error sensitivity analysis, where we analyze how errors in the response functions propagate to our final stage estimates (see **new Supplement  G**). Overall, we find that our WO-learner is highly robust.
* **New discussion on when to choose our WO-learner vs. simple meta-learners:** We have added a new discussion on when to choose our WO-learner vs. simple meta-learners (see our **new Supplement A**). Therein, we give practical recommendations when our WO-learner is preferred and vice versa.
* **New discussion on validity of assumptions:** We have added a thorough discussion on the validity of the standard causal identifiability assumptions (see our **new Supplement B**).
* **More discussion:** We have added a new limitation paragraph to our main paper, and have extended our conclusion at the end of the main paper.
* **Smaller improvements:** We made some **smaller improvements** to the paper. For example, we revised the definition of treatments; we introduced HTE as an umbrella term; we clearly state that and why our baselines are exhaustive and fair; we explained the IVW naming; we clarified the purpose of propensity score clipping; etc.

We are confident that, with the help of the reviews and our improvements, our work will be an important contribution to the community. Finally, we wanted to say “thank you” – we truly appreciate the help, which allowed us to revise our work and make important improvements.

---

### Meta-Review · Area_Chair_w7Zp · 2026-01-06

**Summary:**

This paper develops a method to address the exponential decay of overlap that arises as treatment sequences become long in heterogeneous treatment effect estimation under time-varying environments.

Reviewer u6pF raised concerns about the weakness of the theoretical justification, the lack of real-world motivation and verifiability, and insufficient comparison with baseline methods. In response, the authors introduced theoretical refinements and extended the formalization to address these concerns, justified the evaluation protocol, and provided counterarguments regarding the necessity and role of additional baselines. These responses successfully convinced the reviewer, leading to an increase in the reviewer’s score during the rebuttal phase (prior to the discussion period).

Reviewer gNhd questioned the clarity of the problem formulation and aspects of the methodological setup. The authors addressed these concerns by revising the problem formulation and adding clarifying explanations for several methods and concepts. These clarifications satisfied the reviewer, who increased their score during the rebuttal phase (again, prior to the discussion period).

Reviewer jn4J pointed out several shortcomings in the theoretical analysis, particularly regarding higher-order terms and behavior under low-overlap regimes. The authors responded by providing additional experiments and new theoretical results to substantiate their claims, and by explaining the role of Neyman orthogonality in controlling error propagation. Although the reviewer did not provide a further response, these issues appear to be supplementary and are unlikely to constitute major negative factors in the overall evaluation.

Reviewer hGFP expressed concerns about the limited practical significance of the reported error improvements, the appropriateness of the experimental setup and baselines, and shortcomings in presentation and exposition. In response, the authors clarified the use of more fundamental evaluation metrics, justified the choice of comparison methods, and provided clearer explanations of previously ambiguous points. Owing in part to the timely and thorough rebuttal, the reviewer explicitly reported that their concerns had been resolved during the rebuttal period.

Taken together, the paper was substantially improved through careful peer review and thoughtful rebuttals, and—fortunately—the authors were able to obtain explicit confirmation from multiple reviewers that their concerns had been satisfactorily addressed.

**Reviewer Concerns:**

See above.

**Reviewer Scores:**

See above.

---

### Decision · Program_Chairs · 2026-01-26

Accept (Poster)